# SEEA-R1: Tree-Structured Reinforcement Fine-Tuning for Self-Evolving Embodied Agents

**Wanxin Tian**[1,*], **Shijie Zhang**[1,*], **Kevin Zhang**[2,*], **Xiaowei Chi**[2], **Chun-Kai Fan**[2], **Junyu Lu**[1],
**Yulin Luo**[2], **Qiang Zhou**[1], **Yiming Zhao**[1], **Ning Liu**[1], **Siyu Lin**[2], **Zhiyuan Qin**[1],
**Xiaozhu Ju**[1,†], **Shanghang Zhang**[2,‡], **Jian Tang**[1,‡]

[1]Beijing Innovation Center of Humanoid Robotics
[2]State Key Laboratory of Multimedia Information Processing, School of Computer Science, Peking University

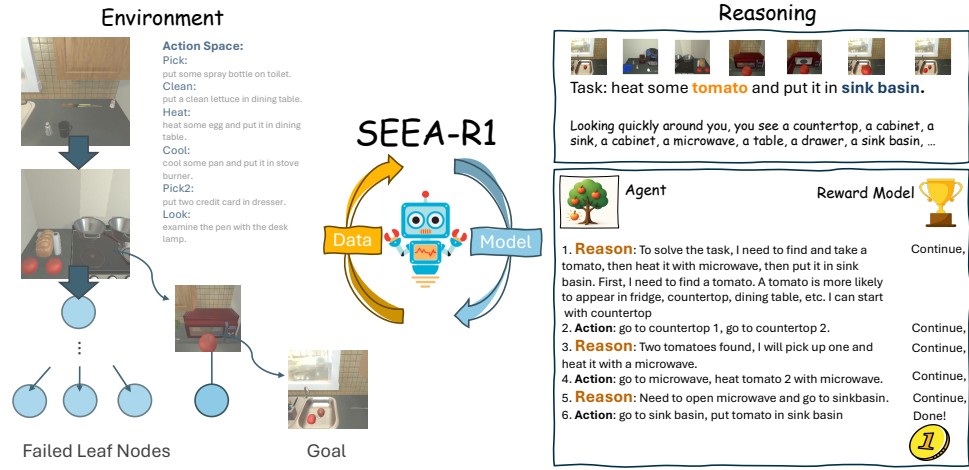

Figure 1: SEEA-R1 self-evolves by reasoning over its environment with perception-grounded planning. The agent explores task solutions using tree-based search guided by a reward model, iteratively refining actions to achieve complex goals. Given a high-level instruction, it explores, plans, and executes actions in an embodied environment.

## Abstract

Self-evolution, the ability of agents to autonomously improve their reasoning and behavior, is essential for the embodied domain with long-horizon, real-world tasks. Despite current advancements in reinforcement fine-tuning (RFT) showing strong performance in enhancing reasoning in LLMs, its potential to enable self-evolving embodied intelligence with multi-modal interactions remains largely unexplored. Specifically, reinforcement fine-tuning faces two fundamental obstacles in embodied settings: (i) the lack of accessible intermediate rewards in multi-step reasoning tasks limits effective learning signals, and (ii) reliance on hand-crafted reward functions restricts generalization to novel tasks and environments. To address these challenges, we present *S*elf-*E*volving *E*mbodied *A*gents-R1, **SEEA-R1**, the first RFT framework designed for enabling the self-evolving capabilities of embodied agents. Specifically, to convert sparse delayed rewards into denser intermediate signals that improve multi-step reasoning, we propose Tree-based Group Relative Policy Optimization (**Tree-GRPO**) integrates Monte Carlo Tree Search into GRPO. To generalize reward estimation across tasks and scenes, supporting unsupervised

---

[*]Equal Contribution, [†] Project Leader, [‡] Corresponding Authors

39th Conference on Neural Information Processing Systems (NeurIPS 2025).

adaptation and reward-driven self-evolution, we further introduce the Multi-modal Generative Reward Model (**MGRM**). To holistically evaluate the effectiveness of SEEA-R1, we evaluate on the ALFWorld benchmark, surpassing state-of-the-art methods with scores of 85.07% (textual) and 46.27% (multi-modal), outperforming prior models including GPT-4o. SEEA-R1 also achieves scores of 80.3% (textual) and 44.03% (multi-modal) without ground truth reward, surpassing all open-source baselines and highlighting its scalability as a self-evolving embodied agent. Additional experiments and qualitative analysis further support the potential of SEEA-R1 for future research in scalable embodied intelligence. Project page is at `https://seea-r1.github.io/`.

# 1 Introduction

Embodied agents commonly operate in complex, long-horizon environments that require not only low-level perception and motor control, but also high-level reasoning, planning, and decision-making capabilities. Despite recent advances in large language models (LLMs) [1, 2, 3] and multi-modal LLMs (MLLMs) [4, 5, 6, 7] have greatly improved agents' abstract reasoning and perception, extending these abilities to open-ended embodied settings remains a fundamental challenge. Since embodied agents often operate interactively over long-horizon tasks, constantly perceiving visual feedback to produce actions, existing LLMs and MLLMs fall short: LLMs often lack perceptual grounding [8], while MLLMs continue to struggle with structured multi-step planning, maintaining long-term coherence, and adapting to dynamic environments [6, 9].

To tackle general-purpose embodied intelligence, we argue that agents must acquire **self-evolution**—the capacity to self-generate training signals and refine reasoning through closed-loop learning. By continuously interacting with the environment and learning from their own experiences, self-evolving embodied agents can bridge the gap between perception and cognition, generalize to novel tasks, and develop long-horizon planning capabilities that transcend the limitations of static, supervised learning paradigms.

In this context, *reinforcement fine-tuning (RFT)* [1, 2] has emerged as a promising paradigm: By providing reward-based feedback, RFT enables agents to learn from trial-and-error trajectories and gradually refine their decision-making policies. However, bridging RFT to embodied agents introduces unique challenges that are absent in symbolic domains such as mathematics or code generation. In this work, we identify two major challenges for reinforcement fine-tuning agents in embodied environments: 1) Lack of intermediate feedback in multi-step reasoning tasks: in embodied tasks with long horizons and delayed sparse rewards, it is difficult to assign credit to intermediate decisions, making it hard for RFT to guide policy learning effectively. 2) Poor generalization of handcrafted reward functions: existing RFT pipelines typically rely on simulator-specific or task-specific reward signals that do not generalize to novel environments, harming agents' abilities to self-improve across diverse tasks.

To address these challenges, we introduce **SEEA-R1 (Self-Evolving Embodied Agents)**, the first framework to adopt RFT for training embodied agents capable of long-horizon reasoning and autonomous self-evolution. SEEA-R1 integrates two key components. 1) **Tree-GRPO** (Tree-based Group Relative Policy Optimization): we extend Group Relative Policy Optimization with Monte Carlo Tree Search (MCTS), enabling agents to explore alternative trajectories and convert sparse outcome rewards into structured, step-wise process signals. This improves credit assignment and facilitates reasoning over extended action sequences. 2) **MGRM** (Multi-modal Generative Reward Model): To eliminate the reliance on handcrafted or environment-specific reward functions, we introduce MGRM—a reward model trained from multi-modal, multi-turn trajectories. MGRM provides task-agnostic reward estimation, supporting generalization and self-improvement across diverse embodied scenarios. Moreover, SEEA-R1 utilizes a novel joint training paradigm: better MLLM reasoning enhances MGRM accuracy, while refined MGRM rewards further advance MLLM policy learning, achieving human-free, self-evolving embodied intelligence.

We evaluate **SEEA-R1** on the ALFWorld benchmark, which rigorously tests an agent's planning and reasoning capabilities by requiring it to map abstract goals to visually grounded action sequences. Our proposed method achieves state-of-the-art success rates of **85.07%** and **46.27%** on textual and multi-modal tasks, respectively, outperforming previous models including Qwen2.5-VL and GPT-4o.

To further evaluate self-evolution in a realistic yet challenging setting, we replace ground-truth rewards with MGRM-based self-supervised signals; SEEA-R1 still achieves **80.30%** and **44.03%** on textual and multi-modal tasks, respectively, surpassing all open-source baselines. These results highlight the effectiveness and scalability of SEEA-R1 as a self-evolving embodied agent.

**Our main contributions are as follows:** (1) We propose **SEEA-R1**, the first reinforcement fine-tuning framework designed to support self-evolving capabilities in embodied agents. (2) By introducing **Tree-GRPO**, we augment GRPO with MCTS to enable dense and interpretable credit assignment across multi-step trajectories, (3) We replace handcrafted reward signals with **MGRM**, a multi-modal generative reward model that rewards task completion. (4) We achieve new **state-of-the-art performance** on the ALFWorld benchmark under both supervised and self-supervised settings, demonstrating strong planning, reasoning, adaptability, and generalization. (5) To facilitate future research and applications in the embodied intelligence community, we will open-source our full framework and modular components—including our reward model MGRM, and training pipelines.

## 2 Related Works

### 2.1 Self-Evolution Learning

Recent research has explored how large language and vision-language models can self-evolve through mechanisms such as self-play, reinforcement learning, and trajectory refinement[10]. Welcome to the Era of Experience [11] highlights the importance of learning from experience and real-world interactions, emphasizing two key components: reward mechanism for environmental interaction and long-horizon perception-action task loop. SPC [12] utilizes an adversarial self-play environment to enhance reasoning, while rStar-Math [13] and Agent-Q [14] demonstrates that small models can improve using MCTS with reward modeling. Dreamsmooth [15] and R2I [16] incorporate planning and prediction modeling into reward design to address long-horizon tasks. RAGEN [17] introduces a modular system to enhance LLM reasoning capabilities in multi-turn interactive environments. Building on these foundations, our approach integrates MCTS into the self-evolution loop, enabling action rollout, trajectory filtering, and policy-reward co-refinement.

### 2.2 Planning for Embodied Agent

LLM-Planner [18], LLM+P [19], and RegionFocus [20] have demonstrated that large language and vision-language models can significantly enhance task performance. Planning is also critical for embodied agents to make sequential decisions and achieve long-term goals. Training-free approaches such as PIVOT [21] and ECoT [22] leverage the spatial reasoning capabilities of VLMs to plan effectively, leading to improved task success rates in embodied agents. In contrast, training-based methods like VIPER [9] and MPO [23] combine vision-language perception with LLM-based reasoning or meta-planning, but often depend on predefined modules or expert-designed strategies. ARMAP [24] and LS-Imagine [25] show that incorporating the planning module into the reward assignment process during training can significantly improve long-horizon task performance in web search and Minecraft environments. Wang et al. [26] combines tree search with preference optimization to improve state prediction and action selection, boosting embodied planning performance.

### 2.3 Reinforcement Fine-Tuning

DeepSeek-R1 [1] [27] shows that RL with formatting and result-only rewards can steer LLMs toward humanlike, complex chain-of-thought reasoning, boosting performance on challenging tasks. Subsequently, Search-R1 [3], TORL [28] explored extending the LLM-R1 approach to tool use, and AlphaLLM-CPL [29] utilizes MCTS for self-improving preference learning. The R1 approaches have also achieved success in the visual domain. Vision-R1 [6], Visual-RFT [4], and Video-R1 [5] improve visual grounding through reward-based reinforcement fine-tuning, while R1-OneVision [30] and Perception-R1 [7] formalize visual input into language for structured reasoning. In the embodied AI domain, robotxR1 [31] transforms visual information into textual descriptions to train LLMs, enhancing their robotic scenario decision-making capabilities. Embodied-R [32] further extends to spatial reasoning. However, these methods are not trained as multimodal models and still rely on task-specific priors. To address these limitations, we developed the first general embodied RL

framework SEEA-R1, which integrates a vision-grounded reward model with MCTS-based planning to enable self-supervised policy optimization without handcrafted rewards.

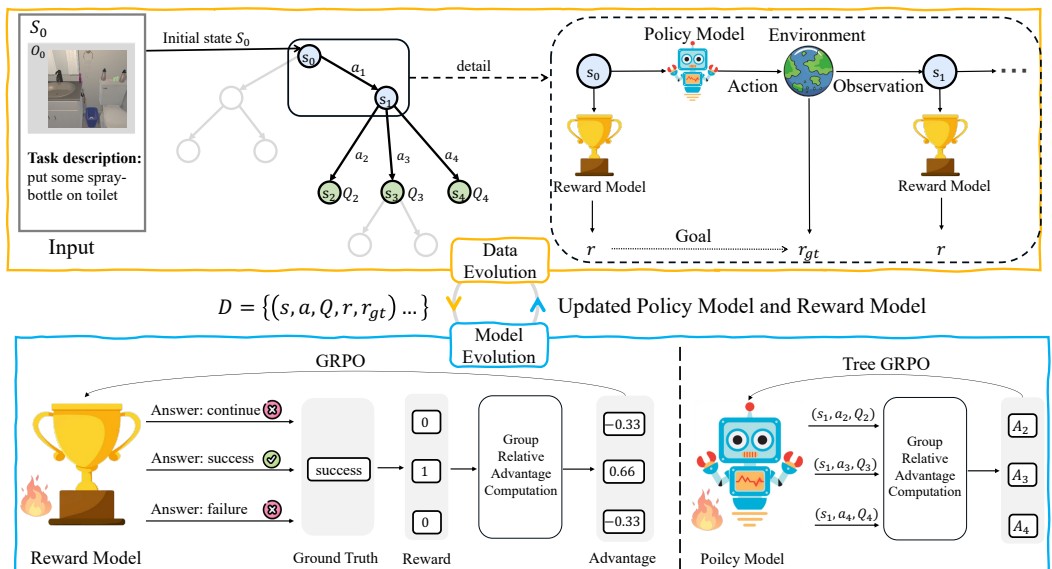

Figure 2: **SEEA-R1 framework.** The framework drives continuous improvement through an iterative loop of two core cycles as follows: **1.Data Evolution:** The Policy Model interacts with the environment via MCTS from an initial state to generate the experience dataset, containing trajectories with derived Q-values, ground truth rewards from the environment, and rewards from the current Reward Model. **2.Model Evolution:** The collected data is used to update both models: (a) **Policy Model** to predict actions and (b) **Reward Model** to predict categorical outcomes. Refined models from Model Evolution then drive the next Data Evolution iteration, enabling continuous self-evolution.

## 3 Methods

In this section, we first formulate the Embodied Agent, the Self-Evolving Embodied Agent, and Multi-round Interaction Embodied Scenario. Then, we briefly introduce the MCTS [33] algorithm. Next, we present SEEA-R1, detailing how data evolution and model evolution are implemented. Finally, we summarize the overall self-evolution loop.

### 3.1 Preliminaries

**Embodied Agent** in our methods is defined as a 4-tuple $(s_t, a_t, r_t, o_t)$ at timestep $t$. It interacts with the environment by receiving observations $o_t$ and taking actions $a_t$ in each step, with state $s_t$ maintaining the full interaction history, and reward $r_t$ reflecting task progress. Further details can be found in Appendix C. A **Self-Evolving Embodied Agent** is defined to satisfy three core conditions: automated data synthesis, iterative capability improvement, and closed-loop reasoning system [34]. **Multi-round Interaction Embodied Scenario** is modeled as a Partially Observable Markov Decision Process (POMDP) [35, 36], formalized as a 7-tuple $(\mathcal{S}, \mathcal{A}, T, R, \mathcal{O}, \Omega, \gamma)$. Here, $\mathcal{S}$ denotes the partially observable state space, $s_t \in \mathcal{S}$; $\mathcal{A}$ is the action space, $a_t \in \mathcal{A}$; $T(s_{t+1}|s_t, a_t)$ represents the transition probability; $r_t$ is the reward function, $r_t \in R$; $\mathcal{O}$ is the observation space, $o_t \in \mathcal{O}$; $\Omega(o_{t+1}|s_t, a_t)$ is the observation probability; and $\gamma$ is the discount factor.

**Monte Carlo Tree Search (MCTS) [33]** is a heuristic algorithm for decision-making in large state spaces widely used in planning, games, and embodied reasoning tasks. After initialization, MCTS iteratively performs four steps: **Selection**: Starting from the root, the agent selects child nodes with the highest UCT scores until reaching a leaf node; **Expansion**: The agent executes the selected action $a_L$ to obtain observation $o_L$, constructs a new node $s_{L+1} = \{s_L, (a_L, o_L)\}$, and expands it

with a set of candidate natural language actions $\{a_{L+1,i}\}_{i=1}^{G}$; **Simulation**: From $s_{L+1}$, multiple rollouts are simulated until termination, giving up, or reaching the depth limit, yielding trajectories $s_0 \to \cdots \to s_T^{(j)}$; **Backup**: Rollout results are used to update visit counts $N(s_t, a_t)$, cumulative rewards $R^{(j)}(s_t, a_t)$, and action values $Q(s_t, a_t)$ along each path back to the root. Further details are provided in Appendix D.

## 3.2 Self-Evolving Emodied Agent-R1

As illustrated in Figure 2, SEEA-R1 continuously improves the agent through two iterative cycles: **Data Evolution**, where the agent to interact with the environment via MCTS to generate experiential data, and **Model Evolution**, where both the Policy Model and Reward Model are updated using this data. The evolved models are then used for the subsequent Data Evolution, forming a closed self-improvement loop. The details of each phase are provided below.

### 3.2.1 Data Evolution: Experience Generation via MCTS

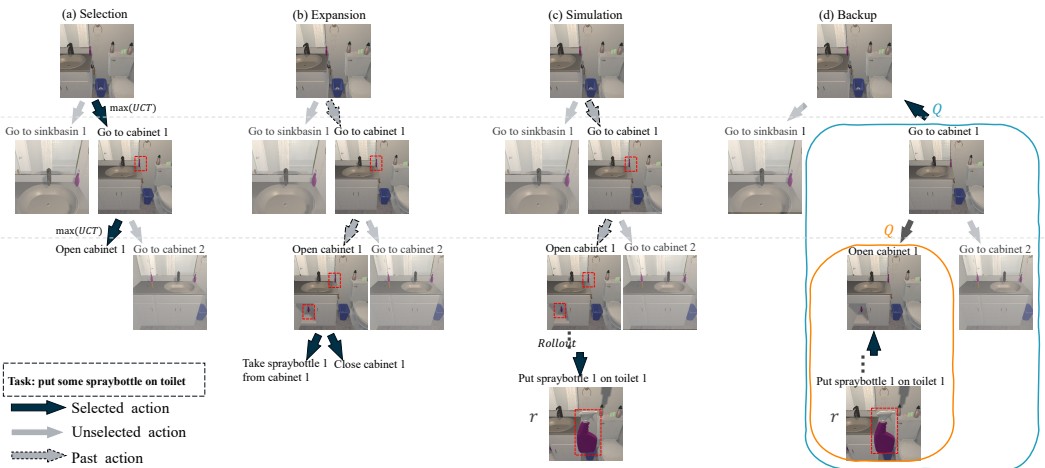

Figure 3: **Monte Carlo Tree Search (MCTS) in SEEA-R1. (a) Selection** Traverse tree via UCT (2) until reaching a leaf. **(b) Expansion** Execute action, observe result, and expand with new actions. **(c) Simulation** Roll out from new node to termination or depth limit, collecting reward $r$. **(d) Backup** Propagate rewards to update action values $Q$ using the formulation in Equation (3).

To convert sparse outcome rewards into dense process rewards for multi-step reasoning, SEEA-R1 employs Monte Carlo Tree Search (MCTS) for experience generation. MCTS estimates action-values, $Q(s_t, a_{t,i})$—the expected future reward for action $a_{t,i}$ in state $s_t$—which serve as *process rewards*. This provides immediate, step-by-step feedback from otherwise sparse environmental signals. The precision of these Q-values and therefore the quality of the rewards of the process improves as MCTS performs more simulations (Figure 3) exploring a wider range of trajectories. This progressive refinement of MCTS-generated experiential data is how SEEA-R1 *implements* 'Data Evolution', leading to increasingly reliable learning guidance for the agent.

### 3.2.2 Model Evolution: Co-refining Policy Model and Reward Model

**Multi-modal Generative Reward Model (MGRM) Training.** To improve the generalizability of the reward function, we adopt a learned reward model as an alternative to handcrafted reward. We design the reward model with several key properties: **multi-modal for perception**, **multi-round for sequential interactions**, and **high interpretability**. Specifically, our reward model is built upon a Multi-modal Large Language Model (MLLM). It takes the historical context $s_t$ as input and predicts one of three categorical outcomes: `success`, `continue`, or `failure`, whose prompt is in Appendix N.4. This allows us to leverage the DeepSeek-R1-Zero [1] training paradigm to employ the GRPO [2] for reinforcement learning , supporting two training paradigms tailored to scenarios with or without ground truth (GT) rewards.

**With GT (Supervised Paradigm)**: MGRM first undergoes *supervised fine-tuning (SFT)* on GT-aligned trajectories (e.g., ALFWorld task completion/step validity signals), minimizing cross-entropy loss between predicted outcomes and GT labels to achieve sufficient performance and eliminate initial biases. During co-training with the policy, MGRM *continues supervised training* by calibrating against GT every 500 episodes—preventing drift and keeping judgments aligned with the simulator's objective truth.

**Without GT (Self-supervised Paradigm)**: In GT-free scenarios (e.g., real-world environments), MGRM relies on Test test Reinforcement Learning (TTRL) [37] and GRPO: the policy generates K (set 10 in environments) diverse trajectories per initial state $s_0$ (via MCTS exploration), and MGRM's majority-voted predictions across these trajectories form pseudo-GT for $s_0$. MGRM is then trained via GRPO (reward = +1 for matching pseudo-GT, 0 otherwise).

**Embodied Agent Policy Update through Tree-GRPO.**  Considering the Embodied Agent as a parameterized model $\pi_\theta(a_t|s_t)$ with parameters $\theta$, which generates a set of available actions $\{a_{t,i}\}_{i=1}^G$ based on the current state $s_t$, the GRPO [2] loss can be reused for this group of actions at each node of the tree-structured experiential data:

$$\mathcal{J}(\theta) = \mathbb{E}_{s_t \sim \mathcal{S}, \{a_{t,i}\}_{i=1}^G \sim \pi_{\theta_{\text{old}}}(\mathcal{A}|s_t)}$$
$$\left[ \frac{1}{\sum_{i=1}^G |a_{t,i}|} \sum_{i=1}^G \sum_{k=1}^{|a_{t,i}|} \min\left(\rho_{t,i,k}(\theta)\hat{A}_{t,i,k}, \text{clip}\left(\rho_{t,i,k}(\theta), 1-\epsilon_{\text{low}}, 1+\epsilon_{\text{high}}\right)\hat{A}_{t,i,k}\right) \right]$$
$$- \beta\mathbb{D}_{\text{KL}}\left[\pi_\theta(\cdot|s_t)\|\pi_{\text{ref}}(\cdot|s_t)\right] \tag{1}$$

where $\rho_{t,i,k}(\theta) = \frac{\pi_\theta(a_{t,i,k}|s_t,a_{t,i,<k})}{\pi_{\theta_{\text{old}}}(a_{t,i,k}|s_t,a_{t,i,<k})}$, $\hat{A}_{t,i,k} = \frac{pr_{t,i}-mean(\{pr_{t,i}\}_{i=1}^G)}{std(\{pr_{t,i}\}_{i=1}^G)}$

Here, $a_{t,i,k}$ denotes the $k$-th token of the $i$-th action $a_{t,i}$ at timestep $t$, and $a_{t,i,<k}$ denotes the tokens preceding token $k$ in action $a_{t,i}$. $\pi_\theta$ is the current Embodied Agent, $\pi_{\theta_{old}}$ is the old Embodied Agent that collected the experiential data, and $\pi_{\text{ref}}$ is the initial Embodied Agent. $\rho_{t,i,k}(\theta)$ represents the importance sampling ratio. $\hat{A}_{t,i,k}$ is the advantage function for the $k$-th token of the $i$-th action at timestep $t$. $\epsilon_{\text{high}}$ and $\epsilon_{\text{low}}$ are from the Clip-Higher strategy of DAPO [38]. $\mathbb{D}_{\text{KL}}\left[\pi_\theta(\cdot|s_t)\|\pi_{\text{ref}}(\cdot|s_t)\right]$ is the k3 estimator of KL divergence between $\pi_\theta$ and $\pi_{\text{ref}}$. The coefficient $\beta$ controls its weight.

### 3.2.3 Iterative Self-Evolution Loop

The SEEA-R1 framework iteratively alternates between Data Evolution and Model Evolution. In each cycle, the updated Embodied Agent and MGRM from the Model Evolution phase are deployed in the subsequent Data Evolution phase. This iterative evolution process allows the SEEA-R1 agent to progressively enhance its reasoning, reward understanding, and task execution capabilities over time.

## 4 Experiments

This section presents an experimental evaluation of the effectiveness of our proposed framework on embodied tasks. The section is structured as follows. We begin by outlining the experimental setup and implementation details. Then, we demonstrate our experimental results comparisons with existing baselines, followed by an in-depth analysis and ablation studies to examine the effects of different training algorithms and our self-evolution reward design.

### 4.1 Experimental Setup

#### 4.1.1 Environment, Dataset and Evaluation

**ALFWorld.** We train and evaluate our approach on ALFWorld [39], an interactive simulation environment and a widely used benchmark for evaluating embodied agents on complex, long-horizon tasks in simulated household environments. We leverage this interactive benchmark to collect data for model training, using the rewards provided by ALFWorld as our ground truth (GT) rewards. ALFWorld provides two types of observations: textual descriptions and raw visual inputs (images),

Table 1: Comparison of **MLLM** methods on unseen tasks (∗-reported in previous work. This convention is also used for other tables.). Success rate (0-100 scale) is shown per task and overall.

| Model | Parameter Size | Optimization Policy | ALFWorld Unseen | | | | | | Avg |
|---|---|---|---|---|---|---|---|---|---|
| | | | Pick | Clean | Heat | Cool | Look | Pick2 | |
| GPT-4o [43] | - | - | 44 | 22 | **29** | **27** | 7 | 23 | **24** |
| Florence-2* [44] | 0.77B | SFT | 0 | 0 | 0 | 0 | 6 | 0 | 1 |
| Idefics-2* [45] | 8B | SFT | 4 | 0 | 0 | 0 | 6 | 0 | 2 |
| MiniGPT-4* [46] | 7B | SFT | 4 | 0 | 19 | 17 | **17** | 16 | 16 |
| InstructBLIP* [47] | 7B | SFT | **50** | 26 | 23 | 6 | 6 | 0 | 22 |
| Qwen2.5-VL [48] | 7B | SFT + DPO | 6 | 21 | 11 | 22 | 5 | 4 | 11 |
| RL4VLM* [49] | 7B | PPO | 47 | 10 | 14 | 18 | 14 | 18 | 21 |
| **SEEA-R1 (w/o GT reward)** | **7B** | **Tree-GRPO** | **44** | **60** | **73** | **58** | **26** | **26** | **44** |
| **SEEA-R1 (w/ GT reward)** | **7B** | **Tree-GRPO** | **43** | **42** | **60** | **41** | **29** | **40** | **46** |

enabling evaluation in both language-only and vision-language settings. Further details on the dataset and expert trajectory collection can be found in Appendix F and H.

**EmbodiedEval.**    To evaluate the generalization ability of SEEA-R1 beyond the training environment, we introduce EmbodiedEval [40] as an out-of-distribution benchmark. EmbodiedEval tests MLLMs as embodied agents across diverse tasks, including Attribute Question Answering (AttrQA), Spatial Question Answering (SpatialQA), Navigation, Object Interaction (Obj), and Social Interaction (Social), within 125 realistic 3D scenes. It provides a comprehensive assessment of agent capabilities in previously unseen scenarios. This setup enables us to measure generalization under significant domain shifts compared to the ALFWorld environment. We analyze the impact of different control strategies on performance, using key metrics such as overall accuracy, which reflects the percentage of fully completed tasks.

**Real-World.**    To evaluate SEEA-R1's real-world performance and analyze the sim2real gap impact on embodied planning, we conducted physical experiments using a dual-arm ARX LIFT 2 with Mobile ALOHA-style teleoperation [41]. Following RoboVQA [42]'s protocol, agents received long-horizon instructions and visual observations (front-view camera) to generate real-time actions for human-teleoperated and single-trajectory execution. We tested 3 task types (Pick & Place, Pick Two & Place, Clean) across 6 environments: apartment, reception room, tea room, family living room, children's room, and kitchen. Each environment contained 12 test cases.

**Evaluation.**    To ensure experimental reproducibility, we set the decoding temperature to zero. When generating task completion trajectories, we include a one-shot in-context example for each task. Detailed prompts are provided in Appendix N. Our primary evaluation metric is the Average Success Rate, which calculates the mean Success Rate across all test set task instances.

### 4.1.2  Implementation Details

**Monte Carlo Tree Search (MCTS)**    We uses two-level pruning to reduce complexity of MCTS: **Probabilistic Expansion** (50% chance to expand all $K$ actions per node, reducing redundancy) and **Strict Path Budget** (hard limit $L=5$ on full $K$-expansions per path, constraining search space). Each MCTS runs 30 iterations with max simulation depth 30.

**Training Setting.**    We use the Qwen2.5-VL-7B-Instruct [48] as the base model to build our embodied agents. Since experience learning in our framework consists of two stages—trajectory sampling and model training—we define each iteration as one complete cycle of these two steps. To ensure fair comparison across different training paradigms (SFT, DPO and GRPO), we use consistent sample data sizes and training hyperparameters. For each iteration, the model is trained on newly collected data for one epoch as steps is 4. The total number of iterations is not fixed and proceeds until convergence. Specifically, for Tree-GRPO, model updates are performed after collecting 512 valid samples (i.e., with non-zero advantage values). For SFT, 512 trajectories with positive advantages are used. For DPO, 512 positive-negative trajectories pairs are sampled from the same parent node in the MCST. When training the multimodal generative reward model (MGRM) via GRPO, the group size is set to 10, which equals to the vote num using TTRL). All training adopt the same hyperparameters: a cosine annealing learning rate schedule (initial LR: 1e-6, warmup ratio: 0.05), batch size of 128,

and KL divergence coefficient $\beta$ of 0.0. Experiments are conducted on 8 NVIDIA A100 80GB GPUs using the ms-swift framework [50].

## 4.2 Experimental Results

### 4.2.1 Overall Performance Comparison

Our proposed SEEA-R1 method, utilizing Qwen2.5-VL-7B-Instruct and Qwen2.5-7B-Instruct as respective base models, was evaluated against strong baselines on ALFWorld test set. As reported in Tables 1 and 7, SEEA-R1 was tested in two settings: text+visual (MLLM), with image observations, and text-only (LLM), with textual descriptions. In the more challenging MLLM setting requiring direct visual perception, SEEA-R1 achieved a 46.27% success rate, substantially outperforming GPT-4o and other leading open-source models, and demonstrated superior efficiency (23.37 vs. 25.10 average steps for GPT-4o). Furthermore, in the text-only (LLM) setting, SEEA-R1 attained an 84.3% success rate, with average completion steps (14.75) on par with GPT-4o.

### 4.2.2 Real World Results

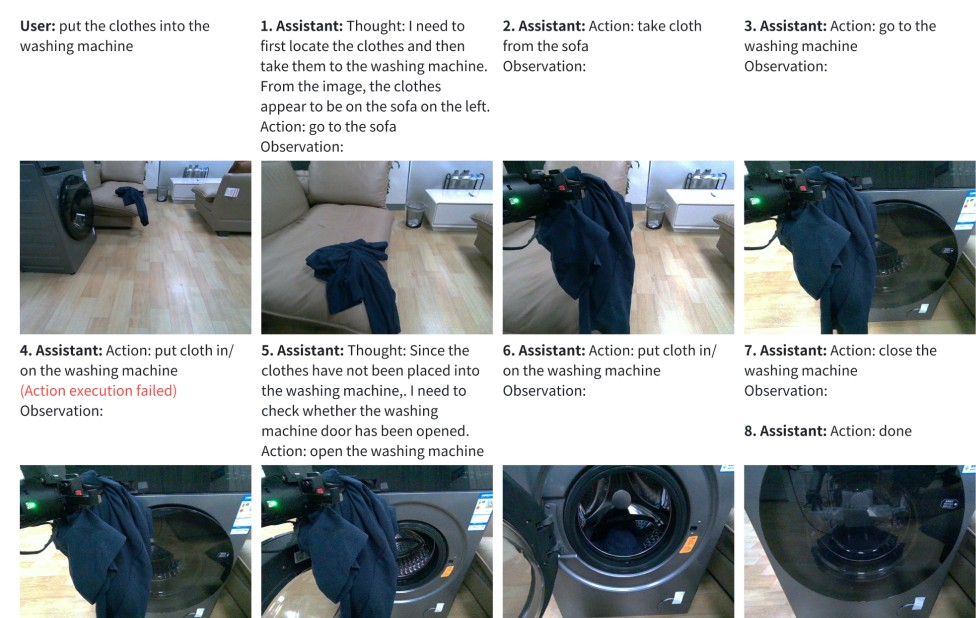

Figure 4: Visualization of SEEA-R1 executing the "put clothes into the washing machine" task in real-world settings, which demonstrates reflection-correction capability.

We evaluated SEEA-R1 via single-trajectory execution against the Qwen2.5-VL-7B-Instruct baseline in physical real-world scenarios. As shown in Figure 4, SEEA-R1 exhibits striking reflection and correction abilities: when an action fails (e.g., attempting to place clothes into a closed washing machine), it actively identifies the error, adjusts its behavior, and re-executes successfully—an ability fostered by our MCTS-based self-evolution training framework. Quantitatively, Table 2 reveals that SEEA-R1 achieves a 34.72% absolute improvement in overall success rate across all tasks and scenarios, with particularly notable gains in complex environments like kitchens (9/12 vs. 1/12) and children's rooms (10/12 vs. 4/12). This performance leap stems from the robust feature learning enabled by MCTS during training: even without simulation-specific advantages (e.g., rewinds), the model generalizes its ability to anticipate long-term consequences to real-world dynamics. These results confirm that our approach not only transfers effectively to physical robot settings but also empowers models with enhanced adaptive decision-making through strengthened reflection-correction mechanisms.

Table 2: Performance Comparison on Real-World Tasks and Scenarios

| Model | Living Room | Guest Room | Apartment | Children's Room | Kitchen | Tea Room | Overall |
|---|---|---|---|---|---|---|---|
| Qwen2.5-VL-7B-Instruct | 58.3 | 50.0 | 25.0 | 33.3 | 8.3 | 25.0 | 33.3 |
| SEEA-R1-7B (w/ GT reward) | **83.3** | **58.3** | **58.3** | **83.3** | **75.0** | **50.0** | **68.1** |

### 4.2.3 Comparative Study of Training Algorithms

To validate our proposed Tree-GRPO, we compared it against MCTS combined with DPO and SFT on ALFWorld. As shown in Figure 5, Tree-GRPO (blue line) consistently achieves superior task success rates (Figure 5a) and greater efficiency with fewer average steps (Figure 5b) than MCTS + DPO (purple line) and MCTS + SFT (red line). This demonstrates the enhanced performance of integrating GRPO within our MCTS-driven self-evolving framework for long-horizon embodied tasks.

### 4.2.4 Efficiency Analysis

Our method achieves higher success rates with lower computational cost (especially in inference token usage) via **training-stage algorithmic optimization** and **lightweight inference design**.

**Search Algorithm Optimization.** Standard MCTS has complexity $O(T \cdot D \cdot K)$ (where $T$ = training steps, $D$ = search tree depth, $K$ = actions per node). We reduce this to $O(T \cdot max(D, K))$ via MCTS pruning as mentioned in Section 4.1.2.

**Inference Efficiency.** MCTS is only used during training. At test time, SEEA-R1 uses fast ReAct-style single-path sampling (no MCTS overhead). As shown in Table 4, this achieves **3.1× higher success rate** with **38.8% fewer tokens**, confirming efficient inference.

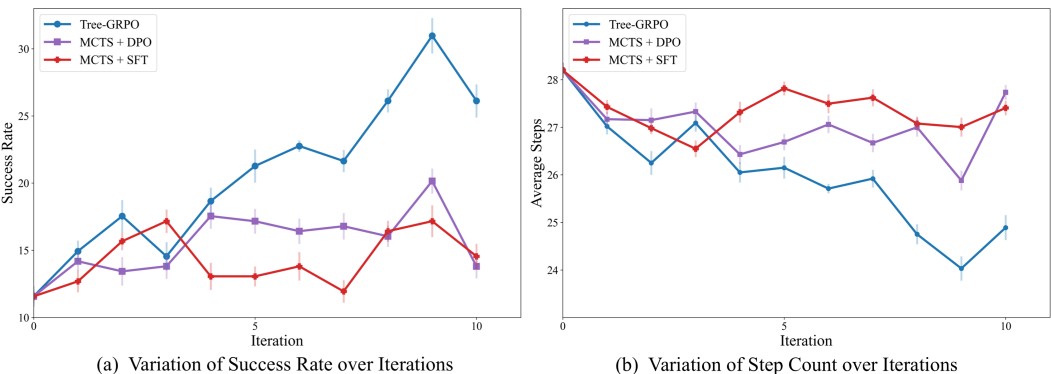

(a) Variation of Success Rate over Iterations    (b) Variation of Step Count over Iterations

Figure 5: Performance comparison of SEEA-R1 using different optimization algorithms on the multi-modal scenario of ALFWorld Benchmark over training iterations, more detailed figures are provided in Appendix G.3.

### 4.2.5 Ablation Study: Self-Evolution with Different Rewards

We conducted an ablation study to demonstrate that our SEEA-R1 can achieve sustained performance improvement by leveraging its internally trained Multi-Modal Generative Reward Model (MGRM), thereby reducing reliance on ground truth rewards from the simulator. We compared three critical configurations with results shown in Figure 6:

The key finding from Figure 6 is that SEEA-R1, when utilizing its self-trained MGRM (both in supervised and self-supervised paradigms), can effectively learn and enhance its policy. In the supervised paradigm (Figure 6(a)), the supervised MGRM enables steady performance that approaches the upper bound set by GT rewards. In the self-supervised paradigm, the self-supervised MGRM drives continuous success rate improvements, even outperforming the GT reward baseline in later iterations. This highlights the MGRM's capability to provide reliable internal reward signals, enabling robust self-evolution and continuous performance gains without persistent dependence on external GT environment rewards.

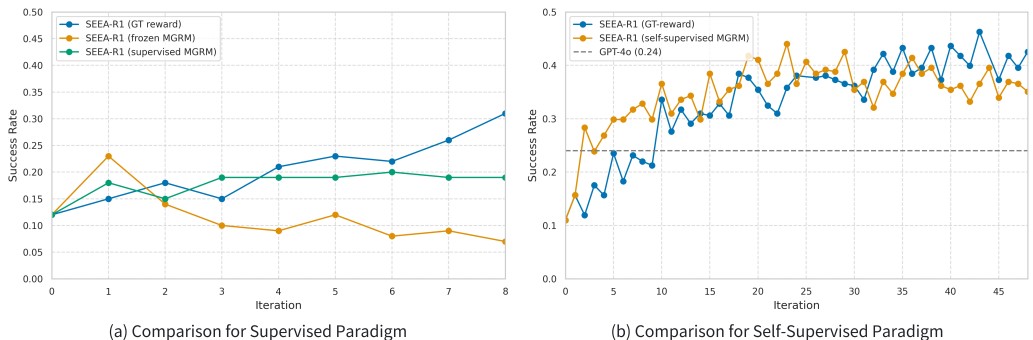

Figure 6: Comparison of success rates of SEEA-R1 under different reward and training paradigms. (a) Analyzes the supervised paradigm, contrasting performance with GT reward, frozen MGRM, and supervised MGRM. (b) Examines the self-supervised paradigm, comparing GT reward, self-supervised MGRM, and the GPT-4o baseline.

#### 4.2.6 Generalization on Out-of-Distribution Embodied Benchmark

To assess the generalization capabilities of proposed SEEA-R1, we conducted evaluations on the EmbodiedEval benchmark, an out-of-distribution (OOD) multi-modal embodied simulation test set. The success rates are reported in Table 3. Notably, our SEEA-R1 method, when fine-tuned with Tree-GRPO, demonstrates significantly improved performance after training. It achieves an overall success rate of 19.88%, which not only shows an uplift compared to the base model (Qwen2.5-VL-7B-Instruct at 18.29%) but also markedly outperforms models fine-tuned with SFT (16.77%) and DPO (16.77%). These results underscore the strong generalization performance of Tree-GRPO approach on this challenging, unseen benchmark.

Table 3: Success Rate in EmbodiedEval benchmark. Tasks include Attribute Question Answering (AttrQA), Spatial Question Answering (SpatialQA), Navigation, Object Interaction (Obj), and Social Interaction (Social).

| Model | AttrQA | Navigation | Obj | Social | SpatialQA | Overall |
|---|---|---|---|---|---|---|
| GPT-4o* [43] | 35.79 | **31.03** | 10.11 | **11.76** | **32.69** | **25.00** |
| Vision-R1-7B [5] | - | - | - | - | - | 8.54 |
| Ocean-R1 [51] | - | - | - | - | - | 15.85 |
| Qwen2.5-VL-7B* | 34.74 | 18.97 | 5.62 | 0.00 | 21.15 | 18.29 |
| Qwen-VL-Max* [52] | **37.89** | 24.14 | **24.91** | 8.82 | 17.31 | 21.04 |
| LLaVA-NEXT-72B* [53] | 23.16 | 12.07 | 7.83 | 0.00 | 5.77 | 10.67 |
| LLaVA-OneVision-72B* [53] | 26.32 | 10.34 | 1.12 | 0.00 | 19.23 | 12.80 |
| Qwen2.5-VL-7B + MCTS + SFT | 28.42 | 20.69 | 4.49 | 5.88 | 19.23 | 16.77 |
| Qwen2.5-VL-7B + MCTS + DPO | 27.37 | 17.24 | 5.62 | 5.88 | 23.08 | 16.77 |
| **SEEA-R1 (w/ GT reward)** | **30.85** | **21.05** | **4.60** | **6.25** | **32.69** | **19.88** |

## 5 Conclusion

We introduced SEEA-R1, the first framework to adapt reinforcement fine-tuning (RFT) for training self-evolving embodied agents. To address the challenges of sparse rewards and limited generalization for embodied domains, SEEA-R1 integrates Tree-GRPO, which leverages MCTS to densify reward signals, and MGRM, a multi-modal reward model that generalizes across tasks and environments. SEEA-R1 achieves new state-of-the-art performance on ALFWorld, outperforming previous models, including GPT-4o, and demonstrates strong generalization in self-learning settings. These results demonstrate the potential of SEEA-R1 as a scalable framework for embodied reasoning and autonomous learning. Further discussions on the limitations of this work are in Appendix M.

## 6   Acknowledgments

This work was supported by the National Natural Science Foundation of China (62476011).

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

# A Appendix Overview

This appendix provides detailed supplementary material to complement our main paper. It is organized as follows:

- **Appendix B: Algorithm Details**
  We outline the overall training procedure of our Self-Evolving Embodied Agent (SEEA-R1), as detailed in Algorithm 1.

- **Appendix C: Embodied Agent Formulation**
  We describe the formalization of the embodied agent's interaction with the environment as a Partially Observable Markov Decision Process (POMDP), detailing the observation, action, state, and reward components.

- **Appendix D: MCTS Implementation**
  We elaborate on the Monte Carlo Tree Search (MCTS) mechanism, a core component of our SEEA-R1, covering its initialization, selection, expansion, simulation, and backup phases.

- **Appendix E: Exploration of World Model as Environmental Feedback**
  We present our findings from exploring the use of a state-of-the-art world model (Kling) as environmental feedback in the self-evolving agent, discussing the challenges encountered such as visual hallucinations, limited instruction understanding, and generation inefficiency.

- **Appendix F: Dataset Details**
  We describe the ALFWorld dataset used in our experiments, including its structure, evaluation splits, and the rationale behind focusing on the test-unseen setting to assess out-of-distribution generalization.

- **Appendix G: Extended Experimental Results**
  This section presents comprehensive experimental results, including:
  - Evaluation on the General Multimodal Benchmark (MMBench) with a detailed breakdown of reasoning and perception capabilities.
  - Comparative performance across diverse multimodal benchmarks (MMStar, MMMU, HallusionBench, AI2D, and OCRBench).
  - An ablation study investigating the impact of sample size and batch size on training stability and success rate.
  - An algorithmic comparison demonstrating the superiority of our proposed Tree-GRPO over baselines like MCTS+DPO and MCTS+SFT.
  - Analysis of the long-term performance of Tree-GRPO over extended training iterations.
  - Investigation into the impact of iterative self-evolution on MCTS performance and generated data quality, including evaluations using Supervised Fine-Tuning (SFT).

- **Appendix H: Case Studies on ALFWorld Tasks**
  We present representative execution trajectories across various ALFWorld task categories (e.g., pick-and-place, look, clean, heat, cool) to highlight the SEEA-R1 agent's capabilities. Visualizations of policy and reward model interactions are also included.

- **Appendix I: Qualitative Analysis of Reasoning Traces**
  We analyze the evolution of reasoning capabilities by comparing the ReAct "Thought-Action-Observation" traces of the baseline (Qwen-2.5-VL-7B-Instruct) and SEEA-R1 (Iterations 5/10), focusing on the task "find two CDs and put them in the safe".

- **Appendix J: Detailed Real-World Tasks**
  We detail all of tasks for each scenario in the real-world experiment.

- **Appendix K: Reward Model Accuracy Analysis**
  We compare the task state judgment accuracy of the frozen MGRM (untrained Qwen-2.5-VL-7B-Instruct) and the GRPO-trained MGRM across "Success/Continue/Failure" states, with supporting data in Table 8.

- **Appendix L: Training Cost and Efficiency Analysis**
  We specify the experimental hardware (8 NVIDIA A100 80GB GPUs) and frameworks (MS-Swift for training, vLLM for inference), quantify SEEA-R1's time costs (sampling, policy training, total) under "with/without GT reward" configurations, and analyze efficiency trade-offs, with details in Table 10.

- **Appendix M: Limitations**
  We discuss the limitations of SEEA-R1: like current embodied AI, it fails to fully address complexities in highly dynamic and unpredictable real-world environments (a major field-wide challenge). We also propose future directions, such as scaling experiments with more diverse environments, larger sample sizes, and bigger models to enhance its adaptability to dynamic settings.

- **Appendix N: Prompts Used**
  We provide the full prompts used for both policy and reward models across different datasets (ALFWorld and EmbodiedEval), ensuring the reproducibility of our instruction-tuning pipeline.

## B  Algorithm Details

The overall training procedure of the Self-Evolving Embodied Agent is as follows (Algorithm 1).

---

**Algorithm 1:** Self-Evolving Framework Training Loop

---

1: **Initialize:** Embodied Agent $\pi_{\theta_{old}} \leftarrow \pi_{\text{ref}}$, MCTS parameters, Tree-GRPO hyperparameters
2: **for** iteration $I = 1, 2, \ldots$ **do**
3:     **// Data Evolution: Experience Generation via MCTS**
4:     Collect tree-structured experience data $\mathcal{D}_{\theta_{old}} = \emptyset$.
5:     **for** each episode **do**
6:         Generate MCTS-tree using the current agent $\pi_{\theta_{\text{old}}}$.
7:         Extract triple set $(s_t, a_{t,i}, pr_{t,i})_{i=1}^{G}$ from MCTS-tree, and add to $\mathcal{D}_{\theta_{old}}$.
8:     **end for**
9:     **// Model Evolution: Co-refining Policy Model and Reward Model**
10:    Update reward model parameters by using the GRPO algorithm using $\mathcal{D}_{\theta_{old}}$:
11:    Update agent parameters $\theta$ by optimizing the Tree-GRPO objective $\mathcal{J}(\theta)$ using $\mathcal{D}_{\theta_{old}}$:
12:    $\theta \leftarrow \text{Tree-GRPO\_Update}(\theta_{\text{old}}, \mathcal{D}_{\theta_{old}}, \mathcal{J}, \pi_{\text{ref}})$
13:    $\theta_{\text{old}} \leftarrow \theta$
14: **end for**
15: **Return:** Optimized Embodied Agent $\pi_{\theta}$

---

## C  Embodied Agent Formulation

We model the long-horizon decision-making, multi-turn interaction embodied scenario as a Partially Observable Markov Decision Process (POMDP), formalized as a 7-tuple $(\mathcal{S}, \mathcal{A}, T, R, \mathcal{O}, \Omega, \gamma)$. Here, $\mathcal{S}$ denotes the partially observable state space of the environment; $\mathcal{A}$ is the action space; $T(s_{t+1}|s_t, a_t)$ represents the conditional transition probability; $R(s_t, a_t)$ is the reward function or reward model; $\mathcal{O}$ is the observation space; $\Omega(o_{t+1}|s_t, a_t)$ is the conditional observation probability; and $\gamma$ is the discount factor. The main components of the Embodied Agent are described below:

**Embodied Agent Observation** $o_t \in \mathcal{O}$ is the observation received from the environment at timestep $t$ after the agent takes action $a_t$ in state $s_t$. In pure text-based scenarios, the observation is a natural language string, e.g., `"Observation: The drawer 1 is closed."`. In multimodal scenarios, the observation is an image representing the visual content of the environment from the current perspective.

**Embodied Agent Action** $a_t \in \mathcal{A}$ is the action taken by the agent to interact with the environment. It is a natural language string generated by the agent, following the ReAct [54] format, e.g., `"Thought: The drawer is closed, so I need to open it first to look inside.\nAction: open drawer 1"` or `"Action: open drawer 1"`. The "*Thought*" part represents the model's reasoning process, while the "*Action*" part is the specific command executed in the environment. The environment executes the action $a_t$ generated by the Embodied Agent and returns an observation $o_t$, forming an interaction loop.

**Embodied Agent State** $s_t \in \mathcal{S}$ is the partially observable environment state, derived from the agent's historical exploration. It includes the entire history of interactions from the beginning of the task up to the current timestep, formalized as $s_t = \{s_0, (a_0, o_0), (a_1, o_1), \ldots, (a_{t-1}, o_{t-1})\}$. The

complete context $s_t$ carries clues about the environment state and preserves the reasoning trajectory, enabling the Embodied Agent to maintain task context memory and perform multi-turn reasoning and actions in partially observable scenarios. In ALFWorld, state $s_t$ includes initial state $s_0$ (system prompt describing environment rules and background; user prompt describing the current task goal and providing few-shot examples) and alternating actions $a_t$ and observations $o_t$ (agent-generated thoughts and actions, and environment-returned observations).

**Embodied Agent Reward** $r_t \in R$ is the reward received by the agent at timestep $t$ for taking action $a_t$ in state $s_t$. It can be provided by a reward model or a predefined reward function.

## D  MCTS

MCTS is a crucial component of SEEA-R1, comprising four core steps: selection, expansion, simulation, and backup.

**Initialization**: The initial state $s_0$ is set as the root node of the search tree.

**Selection**: Starting from the root node, the Embodied Agent traverses down the tree. At state $s_t$, it selects the child node with the maximum Upper Confidence Bound for Trees (UCT) value:

$$a_t = \arg\max_{a_{t,i}} \left[ Q(s_t, a_{t,i}) + c \sqrt{\frac{\ln N(s_{t-1}, a_{t-1})}{1 + N(s_t, a_{t,i})}} \right] \tag{2}$$

where $c$ is the exploration constant, $Q(s_t, a_{t,i})$ is the action value, $N(s_t, a_{t,i})$ is the visit count for the action, and $a_{t,i}$ is the $i$-th available action at the current timestep $t$.

**Expansion**: After continuous selection reaches a leaf node $(s_L, a_L)$, the agent first executes this action to obtain an observation $o_L$ from the environment. Then, it creates a new non-leaf node $s_{L+1} = \{s_L, (a_L, o_L)\}$. Finally, it simultaneously samples multiple natural language text segments to expand the new non-leaf node $s_{L+1}$, obtaining a set of available actions $\{a_{L+1,i}\}_{i=1}^G$.

**Simulation**: From the new leaf node $s_{L+1}$, the Embodied Agent performs multiple rollouts until termination—either the task is completed, the agent gives up, or the maximum search depth is reached. The complete trajectory of the $j$-th rollout is formalized as $s_0 \rightarrow s_1 \rightarrow \cdots \rightarrow s_{L+1} \rightarrow s_{L+2}^{(j)} \rightarrow \cdots \rightarrow s_T^{(j)}$.

**Backup**: The results of multiple rollouts are used to update the visit counts $N(s_t, a_t)$, returns $R^{(j)}(s_t, a_t)$, and action values $Q(s_t, a_t)$ along the trajectory, starting from the terminal node and moving upwards. The update formulas are as follows:

$$N(s_t, a_t) \leftarrow N(s_t, a_t) + 1 \tag{3}$$

$$R^{(j)}(s_t, a_t) = r(s_{t+1}^{(j)}) + \gamma \, r(s_{t+2}^{(j)}) + \cdots + \gamma^{T-t-1} r(s_T^{(j)}) \tag{4}$$

$$Q(s_t, a_t) = \mathbb{E}_{a_t \sim \pi_{\theta_{\text{old}}}(\mathcal{A}|s_t)} \left[ R(s_t, a_t) \right] = \frac{\sum_{j=1}^{N(s_t, a_t)} R^{(j)}(s_t, a_t)}{N(s_t, a_t)} \tag{5}$$

### D.1  Training and Efficiency Analysis

**Training Cost.**   SEEA-R1 was trained for 36 hours on 8×A100 GPUs, a modest budget considering its performance surpasses proprietary models such as GPT-4o. For comparison, Qwen2.5 fine-tuning required 24 hours and PPO training took 48 hours under similar hardware settings.

**Algorithmic Optimization.**   Standard MCTS scales as $\mathcal{O}(b^d)$ with branching factor $b$ and depth $d$. To improve scalability, SEEA-R1 introduces a pruning strategy combining **probabilistic expansion** and a **path budget**, effectively reducing complexity to $\mathcal{O}(pLk)$. Each node expands its $k$ actions with 50% probability, and full $k$-expansions are limited to $L = 5$ per path.

**Inference Efficiency.**   MCTS is used only during training. At inference time, SEEA-R1 performs lightweight single-path (ReAct-style) reasoning without search overhead, yielding **3.1× higher success** while using **38.8% fewer tokens**.

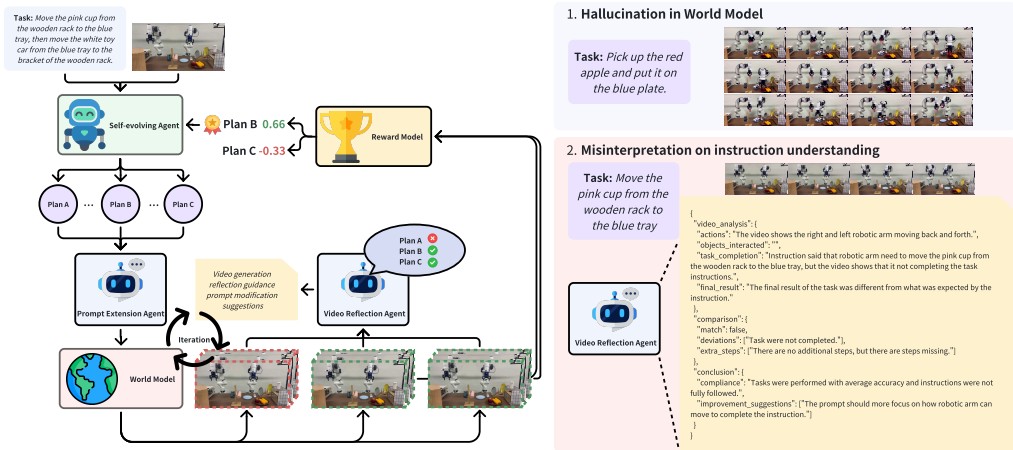

Figure 7: Pipeline of the experiment for the World Model as environmental feedback in Self-evolving Agent

Table 4: Inference Efficiency of SEEA-R1 on ALFWorld.

| Model | Success (%) | Avg. Steps | Total Tokens (Test) |
|---|---|---|---|
| Qwen2.5-VL-7B | 11.57 | 28.10 | 10,701,255 |
| **SEEA-R1** | **36.19** | **23.37** | **6,554,090** |

# E  Exploration of World Model as environmental feedback in Self-evolving Agent

In a self-evolving agent, environmental feedback is critical and can take many forms—such as a simulated environment, a world model, or the real world. Recent research increasingly trains world models on large-scale data so that they can emulate reality. To test whether updated world models are sufficiently capable of serving as environmental feedback, we intend to train a self-evolving agent not only in a simulator but also within a world-model environment. For short, we use kling, a sota world model, as a world simulator in our experiment setting.

During this data-collection phase we found that when the agent relies on a world model for feedback, obtaining a single accurate and complete trajectory is extremely difficult. This suggests that current world models are still far from being fully reliable world simulators and have a long way to go.

We have identified the following aspects that require improvement in world models:

**1. Frequent visual hallucinations in predicted videos.** Hallucinations in the video-based world model are clichéd problems, and there have been many previous studies on how to solve them. In our data collection phase, even when we supply detailed prompts—and likewise with simpler prompts—to the sota world model, we often encounter hallucinations. Typical failures include physically impossible dynamics, objects that suddenly warp or vanish, or the spontaneous appearance of extra robot arms and other artifacts. Repeated tests confirm that the current generation of world models cannot reliably prevent such hallucinations; therefore, the resulting videos frequently fail to meet our requirements and make the model unsuitable as a dependable world simulator.

**2. Limited understanding of task instructions.** Misinterpreting an instruction is distinct from hallucination: the video looks coherent, yet it fails to follow the given command. We observe two main failure modes. (i) Generalisation gaps: when the model encounters an unfamiliar instruction or action, it often produces a static scene or executes the wrong motion—without hallucinating, but still violating the instruction. (ii) Planning errors: if the upstream planner issues a physically unsound command (e.g., "place the plate on the very edge of the table"), a faithful simulator should show the plate toppling. Today's world models cannot reproduce such plausible but incorrect plans, revealing a deeper weakness in instruction comprehension.

**3. Inefficient video generation.** Large-scale visual diffusion models typically require 10–30 minutes to produce a 5–10-second clip. This lengthy turnaround severely slows data collection; if the output is unusable, we must restart the process and wait again. Consequently, training a self-learning agent with a world model demands substantially more time and compute than training in a conventional simulator.

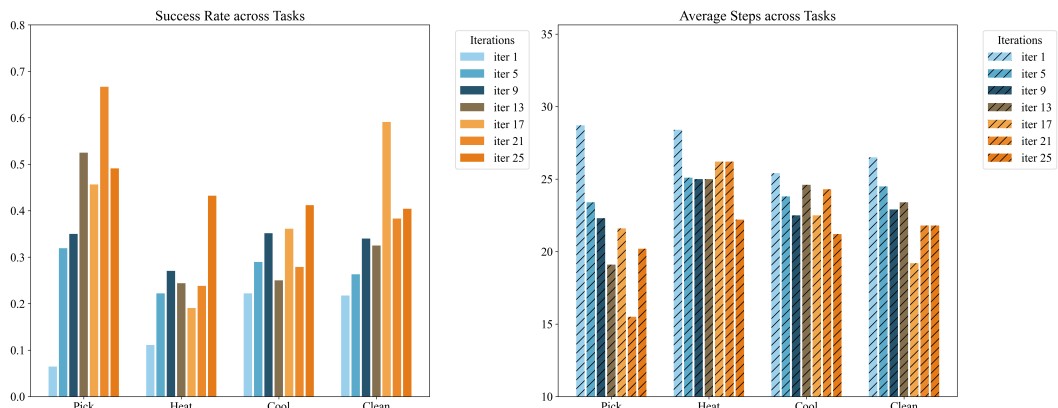

Figure 8: Performance comparison of SEEA-R1 across different tasks in ALFWorld over training iterations. Left: Success rate across tasks. Right: Average number of steps taken to complete tasks.

## F   Dataset Details

**ALFWorld**: The ALFWorld dataset is structured into a training set comprising 3321 games and a test set, further partitioned into test-seen (140 games) and test-unseen (134 games) splits. This distinction is crucial for assessing out-of-distribution (OOD) generalization, as unseen tasks introduce novel environments and instructions. For our experiments, we specifically evaluate our model on the test-unseen split. This focus on unseen scenarios is motivated by the need to rigorously assess the agent's capacity for effective long-horizon planning and generalization to novel embodied tasks, which is critical for real-world deployment. ALFWorld's embodied setup, sparse rewards, and requirement for OOD generalization collectively make it a rigorous benchmark for evaluating agents intended for complex real-world task-solving.

## G   Experiments

### G.1   Evaluation on the Genenal Multi-Modal Benchmark

MMBench is a multimodal benchmark that subdivides reasoning and perception capabilities into six Level-2 dimensions: Logic Reasoning (LR), Attribute Reasoning (AR), Relation Reasoning (RR) for Reasoning, and Fine-Grained Perception-Single Instance (FP-S), Fine-Grained Perception-Cross Instance (FP-C), and Coarse Perception (CP) for Perception. The overall average score is derived from the average of scores for all 20 Level-3 capabilities, and Level-2 capability dimension scores are the average of all Level-3 capability scores within that dimension. Ground truth labels for the test set are not publicly available. The evaluation results are reported in Table 5.

To comprehensively evaluate the capabilities of Multimodal Large Language Models (MM-LLMs), we report performance across a suite of diverse benchmarks in Table 6, each targeting distinct aspects of multimodal intelligence. MMStar assesses general multimodal reasoning and understanding. MMMU (Massive Multi-discipline Multimodal Understanding) focuses on complex, expert-level multimodal reasoning across various academic disciplines. HallusionBench specifically targets the detection and mitigation of hallucinations in MM-LLMs, particularly when dealing with potentially misleading visual information. AI2D (AI2 Diagrams) evaluates the models' ability to understand and reason about scientific diagrams, involving both visual perception and logical inference. Lastly, OCRBench measures the models' optical character recognition capabilities, evaluating their effectiveness in extracting and understanding text embedded within images. This comprehensive evaluation

Table 5: Performance comparison of various Multimodal Large Language Models on the MMBench benchmark.

| Model | Overall | LR | AR | RR | FP-S | FP-C | CP |
|-------|---------|------|------|------|------|------|------|
| Gemini-2.5-Pro | **88.5** | **89.1** | 87.8 | **83.7** | **94.2** | **94** | 83.7 |
| GPT-4o | 86.8 | 87.5 | **91.4** | 83.3 | 84.9 | 91 | **85.4** |
| InternVL3-8B | 82.3 | 71.2 | 84.1 | 82.9 | 88.2 | 81.3 | 80.9 |
| Qwen2.5-VL-7B | 82.4 | 71.7 | 84.9 | 80.2 | 89.8 | 80.1 | 81.3 |
| LLaVA-OneVision-7B | 80.9 | 67.9 | 88.2 | 82.2 | 84.9 | 73.8 | 81.9 |
| **SEEA-R1** | **81.4** | **66.9** | **82.9** | **81.0** | **90.3** | **75.4** | **81.7** |

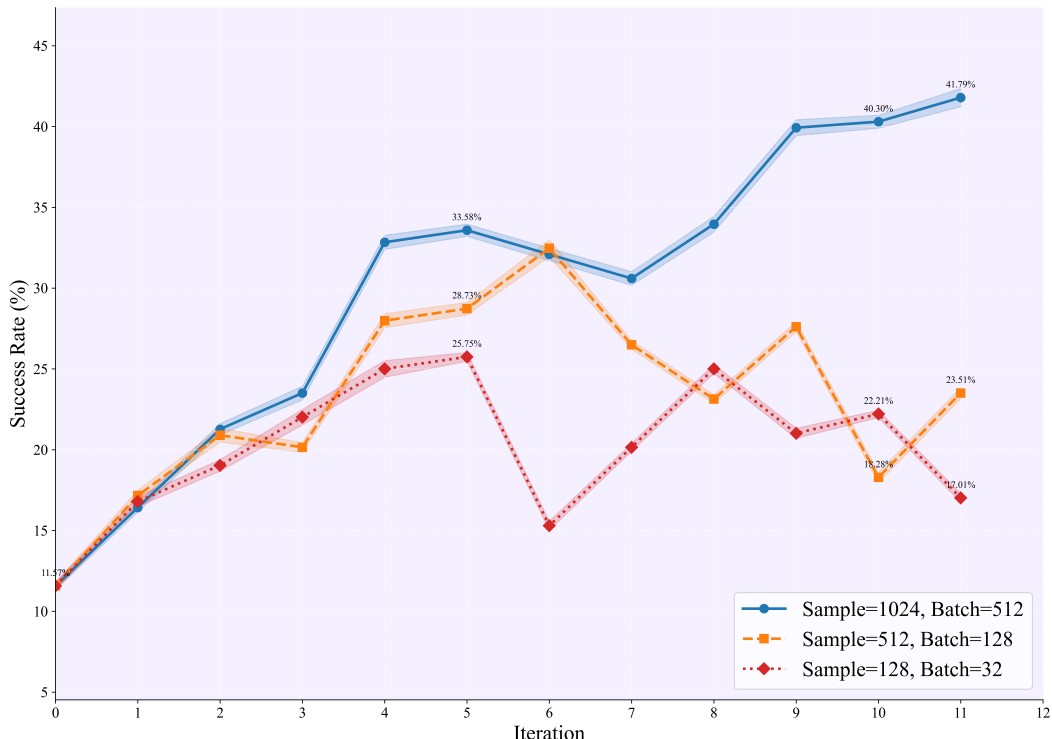

Figure 9: Performance comparison across different sample and batch sizes.

suite provides a holistic view of each model's strengths and limitations in real-world multimodal applications.

## G.2 Impact of Sample Size and Batch Size on Training Stability

We investigate how training configurations affect model performance. As illustrated in Figure 9, larger sample and batch sizes significantly influence the final success rate on the ALFWorld unseen test set. Specifically, the largest configuration (Sample=1024, Batch=512) consistently demonstrates a clear performance advantage, achieving a final success rate of 41.79%. In stark contrast, while the medium setting (Sample=512, Batch=128) reached a peak success rate of 32.46% during its intermediate iterations, and the smallest configuration (Sample=128, Batch=32) similarly achieved its highest accuracy of 25.75% at an earlier iteration, their overall final performance was notably lower and exhibited less stable convergence compared to the largest configuration. These results collectively indicate that increasing the sample and batch sizes contributes to more stable and effective policy updates in GRPO-based training, likely due to reduced variance in gradient estimation, ultimately leading to superior and more consistent performance on unseen tasks.

Table 6: Performance comparison of various Multimodal Large Language Models on diverse multi-modal benchmarks.

| Model | MMStar | MMMU | HallusionBench | AI2D | OCRBench |
|---|---|---|---|---|---|
| Gemini-2.5-Pro | 73.6 | **74.7** | 64.1 | **89.5** | 862 |
| GPT-4o | 70.2 | 72.9 | 57 | 86.3 | 822 |
| SenseNova-V6-Pro | **73.7** | 70.4 | **67.1** | 89.2 | **895** |
| InternVL3-8B | 68.7 | 62.2 | 49 | 85.1 | 884 |
| Qwen2.5-VL-7B | 64.1 | 58 | 51.9 | 84.3 | 888 |
| LLaVA-OneVision-7B | 56.7 | 46.8 | 47.5 | 82.8 | 697 |
| **SEEA-R1** | **60.2** | **48.5** | **66.4** | **79.4** | **766** |

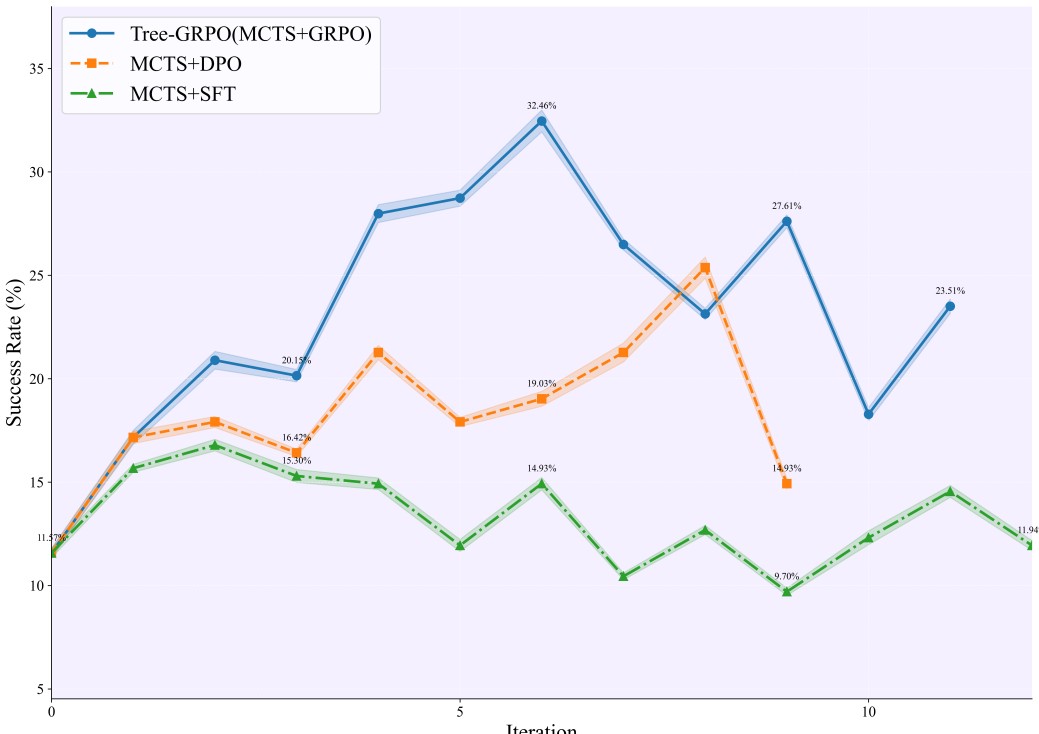

Figure 10: Performance comparison of Tree-GRPO against MCTS+DPO and MCTS+SFT across training iterations.

## G.3 Comparison with Different Algorithms

We investigate the comparative performance of Tree-GRPO against two established baseline methods: MCTS integrated with DPO (MCTS+DPO) and MCTS with SFT (MCTS+SFT). As illustrated in Figure 10, Tree-GRPO consistently outperforms both baselines across training iterations on the ALFWorld unseen test set. Tree-GRPO reached a peak success rate of 32.46%, which is significantly higher than the 25.37% peak achieved by MCTS+DPO and the 16.79% peak of MCTS+SFT.

This superior performance can be largely attributed to GRPO's on-policy nature, which facilitates more accurate and stable policy improvement through direct, interaction-aligned updates. In contrast, DPO is fundamentally an off-policy algorithm; it shares practical limitations with SFT, as both methods heavily rely on static, pre-collected data and lack the dynamic optimization benefits derived from online rollouts. Consequently, the off-policy characteristic of DPO contributes to its observed performance ceiling, while the stagnation and eventual degradation in the performance of MCTS+SFT clearly reflect the inherent limitations of purely supervised learning in complex interactive environments. The on-policy nature of GRPO, especially when combined with tree-based

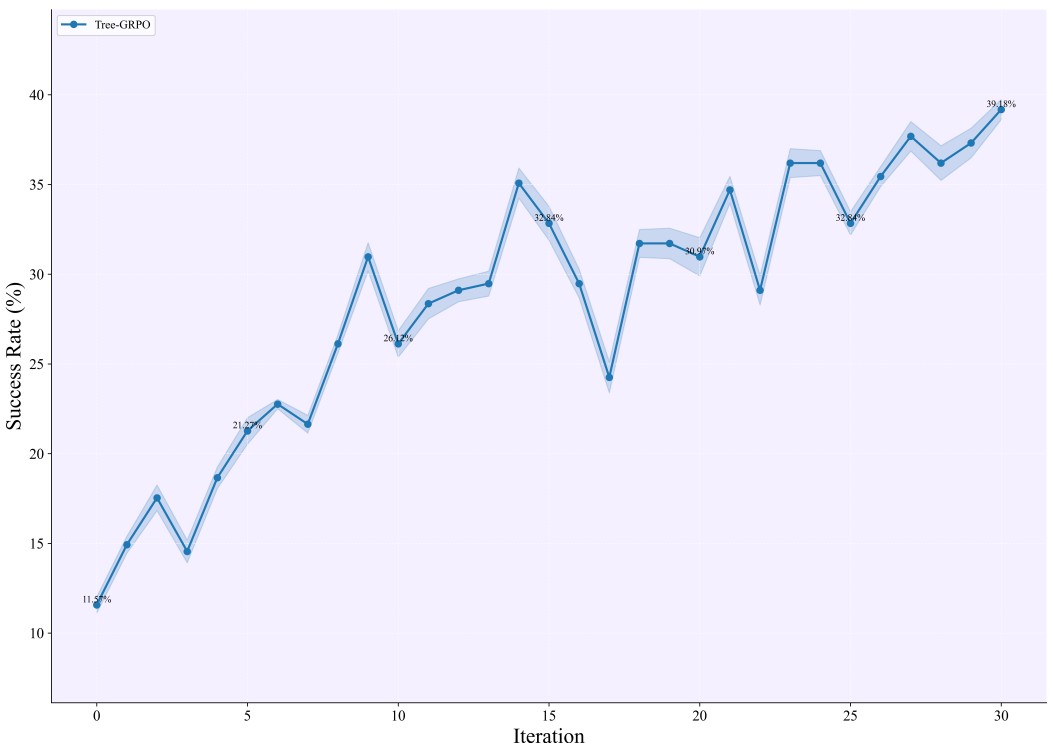

Figure 11: Learning curve of Tree-GRPO over 30 training iterations.

exploration, enables more stable and effective updates, leading to superior sample efficiency and enhanced robustness in reinforcement learning.

## G.4 Long-Term Performance of Tree-GRPO

To thoroughly assess the long-term effectiveness of Tree-GRPO, we extended the training duration of the model to 30 iterations. As illustrated in Figure 11, the success rate on the ALFWorld unseen test set demonstrates a consistent and substantial increase, rising from an initial 11.57% to a robust final success rate of 39.48%. While minor fluctuations are observed, which are indicative of exploration-induced variance inherent in reinforcement learning, the overall upward momentum remains undeniably consistent. This sustained improvement over an extended training horizon unequivocally confirms the scalability and strong convergence potential of Tree-GRPO. Such reliable long-term performance makes Tree-GRPO particularly well-suited for practical deployment in complex, long-horizon decision-making tasks where consistent improvement and robustness are paramount.

## G.5 Impact of Iterative Self-Evolution on MCTS Performance and Data Quality

This experiment investigates how iterative self-evolution affects Monte Carlo Tree Search (MCTS) performance and the quality of trajectories it generates. We hypothesize that refining the MCTS's underlying model through successive self-evolution iterations leads to progressively higher-quality sampled trajectories, as measured by increased average rewards.

The results are presented in Figure 12. A clear positive trend is evident: later self-evolution iterations (warmer colors, e.g., `iter_12`) consistently achieve higher average cumulative maximum rewards compared to earlier iterations (cooler colors, e.g., `iter_1`) across various MCTS search step counts. For instance, at 15 MCTS search steps, `iter_12` attains an average reward of approximately 0.8, whereas `iter_1` reaches about 0.7. This performance gap tends to widen with an increasing number of search steps, implying that models from later iterations benefit more substantially from deeper MCTS searches.

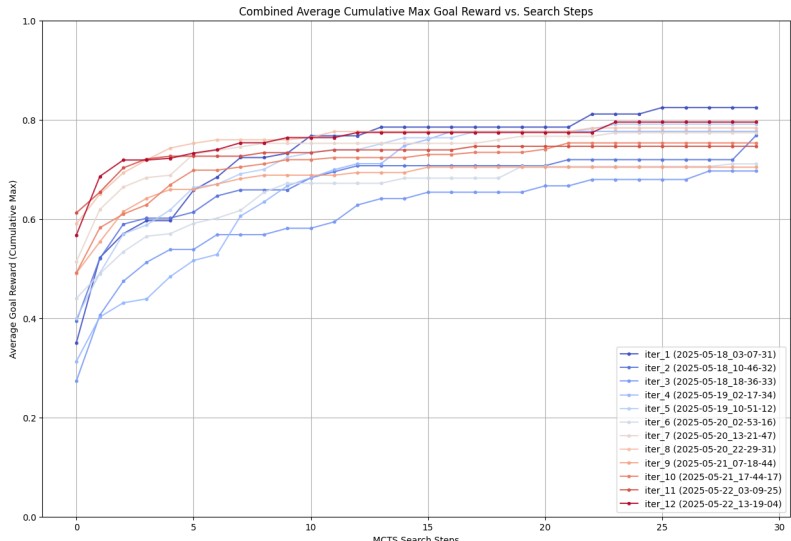

Figure 12: Iterative Self-Evolution Enhances MCTS Performance and Sampled Trajectory Quality. Each line represents a distinct self-evolution iteration (from `iter_1` to `iter_12`), plotting the average cumulative maximum goal reward against MCTS search steps. The color gradient (blues for early iterations, reds for later ones) highlights the progression.

This observed upward trend in average reward across self-evolution iterations strongly suggests that the models become more proficient at guiding MCTS towards successful outcomes. Consequently, trajectories sampled by MCTS using these more advanced models are of increasingly higher quality. This improvement in sampled data is crucial, as it provides a more effective training signal for subsequent model refinement, thereby validating the efficacy of the self-evolutionary approach in enhancing both agent performance and data generation.

### G.6 Evaluating Data Quality from Iterative MCTS + GRPO via Supervised Fine-Tuning

To assess the evolution of data quality throughout the Iterative MCTS + GRPO training process, we conducted an auxiliary experiment. We aimed to determine if data collected in later iterations progressively improves and if this improvement translates to enhanced performance when a separate, pre-trained model (Qwen2.5-VL-7B-Instruct) is fine-tuned using this data.

For this experiment, data generated by the policy model after each iteration of Iterative MCTS + GRPO was collected. These samples were then filtered, retaining only those with an Advantage value greater than zero, which we considered as potentially high-quality instances. Subsequently, this filtered data from each respective iteration was used for Supervised Fine-Tuning (SFT) of the Qwen2.5-VL-7B-Instruct model. We hypothesized that if data quality improves with more iterations, the SFT performance of Qwen2.5-VL-7B-Instruct would exhibit a corresponding upward trend when trained on data from later iterations.

However, as depicted by the blue bars in Figure 13, the SFT performance of Qwen2.5-VL-7B-Instruct did not show a monotonically increasing trend with the iteration number of the source data. Instead, the success rate fluctuated. For instance, SFT on data from iteration 3 yielded a success rate of approximately 0.247, but this dipped with data from iteration 4 (approx. 0.164) and iteration 5 (approx. 0.179). While performance peaked again with data from iteration 7 (approx. 0.268), it subsequently declined for data from later iterations (8, 9, and 10).

This observation—that SFT performance on a fixed baseline model does not consistently improve with the iteration number of the data-generating policy—suggests a nuanced aspect of iterative reinforcement learning. While the Iterative MCTS + GRPO process itself (orange line in Figure 13) demonstrates clear improvement in its own success rate, the data generated at each step may become increasingly specialized or "tailored" to the current state and capabilities of the policy model that produced it. Consequently, this data, while optimal for the self-improvement of the generating policy, may not be universally or increasingly beneficial for fine-tuning a different, general-purpose model

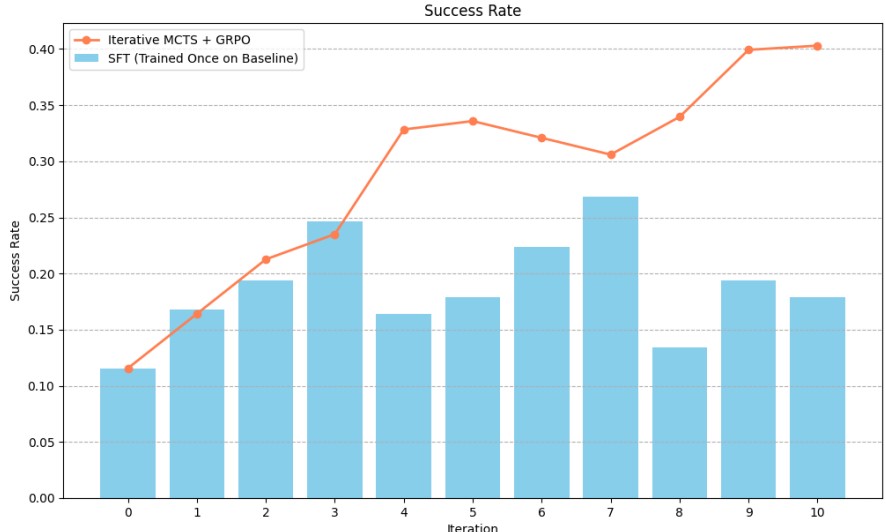

Figure 13: Success Rate Comparison: Iterative MCTS + GRPO vs. SFT on Generated Data. The orange line tracks the success rate of the Iterative MCTS + GRPO policy over its training iterations. The blue bars represent the success rate of the Qwen2.5-VL-7B-Instruct model after SFT using data collected from the corresponding Iterative MCTS + GRPO iteration (filtered for Advantage > 0).

Table 7: Performance comparison on the text-only scenario of ALFWorld Benchmark. **ITO** stands for *Inference-Time Optimization*. MPO (Meta Plan Optimization) [23] provides the agent with the meta plan as context in inference time to improve performance.

| Model | w/o Training | ITO | ALFWorld | | Average |
| --- | --- | --- | --- | --- | --- |
| | | | Seen | Unseen | |
| GPT-4o* [43] | ✗ | - | 78.6 | 83.6 | 81.1 |
| Qwen2.5-7B* [55] | ✗ | | 71.4 | 75.4 | 73.4 |
| Llama-3.1-8B* [56] | ✗ | - | 22.9 | 28.4 | 25.7 |
| Llama-3.1-70B* [56] | ✗ | - | 78.6 | 73.9 | 76.3 |
| GPT-4o* | ✗ | MPO | **89.3** | **93.3** | **91.3** |
| Llama-3.1-8B* | ✗ | MPO | 50.0 | 52.2 | 51.1 |
| Llama-3.1-8B + SFT* [57] | ✓ | - | 79.3 | 71.6 | 75.5 |
| Llama-3.1-8B + ETO* [58] | ✓ | - | 77.1 | 76.4 | 76.8 |
| Llama-3.1-8B + KnowAgent* [59] | ✓ | - | 80.0 | 74.9 | 77.5 |
| Llama-3.1-8B + WKM* [60] | ✓ | - | 77.1 | 78.2 | 77.7 |
| Llama-3.1-8B + SFT* | ✓ | MPO | 80.7 | **81.3** | 81.0 |
| Llama-3.1-8B + ETO* | ✓ | MPO | **85.0** | 79.1 | **82.1** |
| **SEEA-R1 (w/ GT-reward)** | ✓ | - | **85.3** | **85.1** | **85.2** |

like Qwen2.5-VL-7B-Instruct. This suggests that the data collected by the evolving policy becomes more "self-serving" and less generalizable for enhancing distinct models as training progresses

## H    Case Studies on ALFWorld Tasks

We present representative execution trajectories across various task categories in the ALFWorld benchmark. Each case highlights the SEEA-R1 agent's ability to understand language instructions, plan goal-directed behaviors, and interact with the environment effectively.

### H.1 ALFWorld: Pick and Place Task

Figure 14 illustrates a typical Pick and Place task, where the agent is instructed to place a pencil into a shelf. The agent must first locate and pick up the correct object, navigate to the target receptacle, and complete the placement action. This requires not only spatial reasoning but also precise object manipulation.

### H.2 ALFWorld: Multi-Object Placement Task

A more complex variant of the pick and place paradigm is shown in Figure 14, where the agent is asked to place two CDs into a safe. This scenario introduces increased goal complexity, requiring multi-object tracking, inventory management, and the satisfaction of compound task conditions.

### H.3 ALFWorld: Look Task

As shown in Figure 14, the Look task evaluates the agent's perceptual capabilities. The instruction "Look at CD under the desklamp" requires SEEA-R1 to locate the correct spatial reference (under the desklamp), identify the target object, and perform an observation action. This task emphasizes the importance of grounded language understanding and fine-grained spatial perception.

Table 8: Accuracy Comparison Between Frozen and Training MGRM

| Model | Success Accuracy | Continue Accuracy | Failure Accuracy | Overall Accuracy |
|---|---|---|---|---|
| Frozen MGRM | 60.00% (9/15) | 36.42% (460/1263) | 81.25% (39/48) | 51.59% (65/126) |
| Training MGRM | 75.00% (9/12) | 96.92% (1478/1525) | 90.00% (27/30) | 91.67% (77/84) |

### H.4 ALFWorld: Heat and Cool Tasks

Figures 14 and 15 illustrate representative instances of Heat tasks in the ALFWorld benchmark. In these scenarios, the agent is required to manipulate the thermal state of target objects—such as heating a mug prior to placing it in a coffee machine, or ensuring that a tomato is sufficiently warmed before disposal. Successfully completing these tasks demands the agent to not only identify the correct object and comprehend the temporal order of subgoals (e.g., "heat before use"), but also to correctly interact with relevant environmental appliances (e.g., microwaves or stoves). These tasks pose unique challenges in reasoning about object affordances, causal relationships, and multi-step transformations, as the agent must infer the implicit preconditions necessary for goal satisfaction and execute a valid sequence of actions accordingly.

### H.5 ALFWorld: Clean Task
In Figure 15, the Clean task involves transforming the object's state (e.g., cleaning an egg before microwaving it). This category tests the agent's ability to recognize when preprocessing steps are required to satisfy high-level goals.

### H.6 ALFWorld: Composite Pick & Look Task
Figure 15 presents a composite task that blends object manipulation with perception. The agent must pick up an item and inspect its surroundings, requiring coordination of multiple sub-skills within a single instruction. This reflects real-world complexity and cross-modal reasoning demands.

### H.7 Visualization of Policy and Reward Model Interactions
Figure 16 provides qualitative examples of the policy and reward models interacting within the environment. Each panel shows the visual state, the reward model's reasoning ('RM: <think>...'), and the policy model's chosen action ('Actor: Action:...'), illustrating their step-by-step interplay.

## I Qualitative Analysis of Reasoning Traces

This section provides qualitative examples of how the agent's reasoning traces evolve throughout the training process, focusing on the task "find two CDs and put them in the safe" to illustrate improvements in reasoning capability.

### I.0.1 Baseline Model (Qwen-2.5-VL-7B-Instruct)

The baseline model's multi-round chain of thought exhibits limited adaptive reasoning, primarily relying on exhaustive search with repetitive failed actions:

---

**Baseline (Qwen-2.5-VL-7B-Instruct) Reasoning Traces**

Thought: To find the CDs, I should start checking the shelves and drawers since they are likely places where CDs might be stored. I'll begin with shelf 1.
Action: go to shelf 1
...
Thought: There are no CDs on shelf 3 either. I should check the next shelf, which is shelf 4.
Action: go to shelf 4
Observation: [image]
...
Thought: Drawer 3 is closed. I need to open it to check inside.
Action: open drawer 3
Observation: Nothing happens.
... (more than ten consecutive failed attempts)
Thought: Drawer 3 is still closed. I need to open it again to check inside.
Action: open drawer 3
Observation: Nothing happens.

---

The baseline initially identifies plausible CD locations (shelves, drawers) and performs basic exploration but rigidly adheres to exhaustive search logic. It repeatedly executes unsuccessful actions (e.g., "open drawer 3" over ten times) without adjusting strategy, reflecting a lack of reflective reasoning about failure causes.

### I.0.2 SEEA-R1 (Iteration 5)

After 5 training iterations, the model begins to exhibit adaptive reasoning, attempting alternative strategies when facing failures:

---

**SEEA-R1 (Iteration 5) Reasoning Traces**

Thought: To find the CDs, I should start checking the shelves and drawers since they are likely places where CDs might be stored. Shelf 1 seems like a good place to start.
Action: go to shelf 1
...
Thought: Safe 1 is open. I should put the CD inside.
Action: put cd 1 in/on safe 1
Observation: Nothing happens.
Thought: Putting the CD in the safe didn't work. I should try closing the safe first.
Action: close safe 1
Observation: Nothing happens.
Thought: Closing the safe didn't help either. I should try opening it again.
Action: open safe 1
Observation: [image]
Thought: Now that the safe is open, I should put the CD inside.
Action: put cd 1 in/on safe 1
...

---

SEEA-R1 (Iteration 5) avoids mindless repetition of failed actions. When "put cd 1 in safe 1" fails, it reasons about potential causes (e.g., safe state) and tests alternatives (close → open). Though initial understanding of the safe's state is flawed, it eventually succeeds through strategic adjustment, demonstrating emerging reflective reasoning.

### I.0.3 SEEA-R1 (Iteration 10)

After 10 training iterations, the model exhibits structured planning and targeted corrective reasoning:

---

**SEEA-R1 (Iteration 10) Reasoning Traces**

Thought: To solve the task, I need to find and take the first CD, then put it in the safe, then find and take the second CD, then put it in the safe. First, I need to find the first CD. A CD is more likely to appear in drawer, desk, shelf, and laundryhamper. I can check one by one, starting with drawer 1.
Action: go to drawer 1
...
Action: put cd 1 in/on safe 1
Observation: Nothing happens.
Thought: The action to put the first CD in the safe did not work, so I need to try opening the safe first.
Action: open safe 1
Observation: [image]
Action: put cd 1 in/on safe 1
...

---

SEEA-R1 (Iteration 10) shows significant improvements in planning and execution: (1) Its initial thought chain outlines a clear step-by-step plan (find → take → place CD1; repeat for CD2), with subsequent actions aligning closely with this plan. (2) When encountering failures (e.g., CD not placed), it immediately engages in reflective reasoning to identify fixes (e.g., "open the safe first"). (3) Unlike earlier stages, it minimizes redundant thoughts, focusing on actionable logic—evolving naturally toward efficient reasoning without explicit length penalties.

This progression confirms enhanced coherence and adaptability in the agent's internal thought processes.

## J Detailed Real-World Tasks

Table 9 lists all real-world tasks used in our experiments, organized by scenario. Each scenario contains 12 distinct tasks.

## K Reward Model Accuracy Analysis

The frozen reward model refers to directly using the untrained Qwen2.5-VL-7B-Instruct model to judge task states, and the accuracy of this untrained reward model in judging task states is only **51.59%**. Therefore, the reward signals are highly unstable, as it can only provide correct reward signals in half of the cases. For example, when Embodied Agents are still in the process of completing the task, the actual task completion status is "Continue", but the reward model directly judges the task as "Failure", thus misjudging the actual situation and providing an incorrect reward signal.

In Table 8, the reward model trained with GRPO improves the prediction accuracy to **91.67%**, enabling it to provide more accurate reward signals to Embodied Agents. The feedback from the frozen MGRM was primarily noisy rather than consistently rewarding incorrect actions, given its low accuracy of 51.59%—near random for binary judgments of task states (e.g., "Continue" vs. "Failure"). However, this noise included critical misjudgments that disrupted the agent's learning process. Specifically, the frozen model frequently misclassified ongoing task states (where the agent should "Continue") as "Failure," prematurely signaling task termination. Over time, such erroneous feedback led the embodied agent to learn flawed behavioral patterns: it began to halt task execution prematurely, even when progress was valid, as it adapted to the unreliable reward signals. In short, while the frozen MGRM did not systematically reward incorrect actions, its noisy feedback—particularly misclassifications of ongoing progress as failure—introduced persistent biases that distorted the agent's learned behaviors.

| Home Living Room | | Children's Room | |
|---|---|---|---|
| 1 | Put the lemon on the sofa | 1 | Put the toy on the bedside table |
| 2 | Put the beverage on the TV cabinet | 2 | Put the toy on the bed |
| 3 | Put two fruits on the cabinet | 3 | Put the pillow on the bed |
| 4 | Put two fruits on the sofa | 4 | Close the bedside drawer |
| 5 | Put the beverage on the cabinet | 5 | Put the toy back on the shelf |
| 6 | Put the beverage on the sofa | 6 | Put the book on the bed |
| 7 | Put the apple in the plate | 7 | Put the book on the bookshelf |
| 8 | Throw the cup into the trash bin | 8 | Turn on the bedside lamp |
| 9 | Put the pear on the coffee table | 9 | Open the bedside drawer |
| 10 | Put the apple and pear on the coffee table | 10 | Put the book in the drawer |
| 11 | Put the apple in the refrigerator | 11 | Take the book out of the drawer |
| 12 | Tidy up items on the sofa | 12 | Put the toys on the bed into the drawer |
| **Guest Room** | | **Kitchen** | |
| 1 | Put the book on the sofa | 1 | Put the pot on the gas stove |
| 2 | Put the book back on the bookshelf | 2 | Put the cherry tomatoes in the plate |
| 3 | Put the beverage on the sofa | 3 | Put the apple in the refrigerator |
| 4 | Throw the cup into the trash bin | 4 | Put the bowl in the sink |
| 5 | Put the apple in the plate | 5 | Wipe the countertop |
| 6 | Put the apple and banana on the coffee table | 6 | Put the empty plate in the cabinet |
| 7 | Throw the paper ball and cup into the trash bin | 7 | Put the sesame paste back into the refrigerator |
| 8 | Take the fruits on the sofa to the coffee table | 8 | Take the soybean paste to the stove |
| 9 | Put the apple and mango on the plate | 9 | Put the chili in the refrigerator |
| 10 | Put the pillow on the floor back on the sofa | 10 | Arrange the seasonings on the countertop neatly |
| 11 | Wipe the coffee table with a towel | 11 | Throw the trash on the countertop into the trash bin |
| 12 | Tidy up the pillows on the sofa | 12 | Put the tableware on the shelf |
| **Tea Room** | | **Apartment** | |
| 1 | Put the book on the chair | 1 | Put the apple in the refrigerator |
| 2 | Put the book back on the bookshelf | 2 | Throw the beverage bottle into the trash bin |
| 3 | Put the beverage bottle on the bookshelf | 3 | Take the beverage from the refrigerator to the kitchen counter |
| 4 | Put the bowl on the table | 4 | Wipe the coffee table with a towel |
| 5 | Put the cushion on the chair | 5 | Take the beverage from the refrigerator to the table |
| 6 | Tidy up the tea sets on the tea table | 6 | Throw the paper ball and cup into the trash bin |
| 7 | Throw the trash on the table into the trash bin | 7 | Put the beverage and apple in the refrigerator |
| 8 | Put the water cup and beverage on the tea table | 8 | Put the clothes in the washing machine |
| 9 | Take a book and put it on the tea table | 9 | Put the apple and banana in the refrigerator |
| 10 | Wipe the tea table clean with a towel | 10 | Tidy up the pillows on the sofa |
| 11 | Put the cups on the tea table onto the dining table | 11 | Put the shoes on the shoe rack |
| 12 | Take the fruits on the bookshelf to the tea table | 12 | Put the apple in the refrigerator and the book on the sofa |

Table 9: Detailed real-world tasks by scenario. Each scenario includes 12 tasks evaluating different manipulation capabilities.

# L    Training Cost and Efficiency Analysis

This section provides detailed analysis of the training costs and efficiency of our proposed methods, including experimental setup, computational requirements, and performance comparisons.

## L.1    Experimental Environment

All experiments were conducted on a high-performance computing cluster equipped with 8 NVIDIA A100 80GB GPUs. We used the MS-Swift framework for distributed model training, which provided efficient scaling across multiple GPUs. For inference performance evaluation, we employed the vLLM library [61] to ensure high-throughput and low-latency model serving.

## L.2 Training Efficiency Comparison

Table 10 presents a comprehensive comparison of the training efficiency between different configurations of our SEEA-R1 model. All time metrics in the table represent average values per training iteration; specifically, they include average sampling time per iteration, policy training time per iteration, reward model training time per iteration, and total training time per iteration. All time units are standardized to minutes for consistency.

Table 10: Training Efficiency Comparison of SEEA-R1 Under Different Reward Configurations (Per Training Iteration).

| Method | MGRM Training Method | Average Sampling Time | Policy Training Time | Reward Model Training Time | Total Time |
|---|---|---|---|---|---|
| SEEA-R1 (w/o GT-reward) | TTRL | 173.5 mins | 4.0 mins | 244.0 mins | 421.5 mins |
| SEEA-R1 (w/ GT-reward) | - | 190.1 mins | 4.0 mins | - | 194.1 mins |

## L.3 Key Observations

From the training efficiency analysis, we can draw several important conclusions:

1. **Self-supervised Training Overhead**: The SEEA-R1 model with self-supervised MGRM training (w/o GT-reward) requires significantly more total training time (248 minutes) compared to the supervised configuration, primarily due to the additional reward model training time (244 minutes).

2. **Sampling Efficiency**: Despite the longer total training time, the self-supervised configuration demonstrates more efficient sampling, with an average sampling time of 173.50 minutes compared to 190.12 minutes for the supervised configuration.

3. **Policy Training Consistency**: The policy training time remains consistent at 4 minutes across both configurations, indicating that the core policy learning process is not significantly affected by the reward configuration.

4. **Cost-Effectiveness Trade-off**: The self-supervised approach, while computationally more expensive, offers the advantage of not requiring ground truth rewards, making it more practical for real-world scenarios where labeled data is scarce or unavailable.

These findings highlight the computational considerations involved in deploying our proposed methods and provide insights into the trade-offs between training efficiency and the need for ground truth supervision.

# M  Limitations

Despite its strong performance and generalization, SEEA-R1, like current embodied AI, does not yet fully address the complexities of operating in highly dynamic and unpredictable real-world environments. This reflects a broader, significant challenge for the field, indicating a substantial path forward for future research to bridge the gap to truly autonomous real-world agents. Future work might explore scaling experiments involving more diverse environments, larger sample size, and bigger models, aiming to enhance SEEA-R1's capacity for dynamic environments.

# N Prompts

## N.1 Prompt for the ALFWorld Dataset

---

**Instruction Prompt for ALFWorld**

Interact with a household to solve a task. Imagine you are an intelligent agent in a household environment and your target is to perform actions to complete the task goal. At the beginning of your interactions, you will be given the detailed description of the current environment and your goal to accomplish.

For each of your turn, you will be given the observation of the last turn. You should choose from two actions: "Thought" or "Action". If you choose "Thought", you should first think about the current condition and plan for your future actions, and then output your action in this turn. Your output must strictly follow this format:"Thought: your thoughts.\n Action: your next action"; If you choose "Action", you should directly output the action in this turn. Your output must strictly follow this format:"Action: your next action".

The available actions are:
    1. go to {recep}
    2. take {obj} from{recep}
    3. put {obj} in/on{recep}
    4. open{recep}
    5. close{recep}
    6. use {obj}{recep}
    7. clean {obj} with{recep}
    8. heat {obj} with{recep}
    9. cool {obj} with{recep}
where {obj} and{recep} correspond to objects and receptacles.

After your each turn, the environment will give you immediate feedback based on which you plan your next few steps. if the envrionment output "Nothing happened", that means the previous action is invalid and you should try more options.

Reminder:
1. The action must be chosen from the given available actions. Any actions except provided available actions will be regarded as illegal.
2. Think when necessary, try to act directly more in the process.

- - -
Here is an example.
{example}
- - -

Now, it's your turn and here is the task.
{task_instruction}

---

## N.2 Prompt for the EmbodiedEval Dataset

> **Instruction Prompt for EmbodiedEval**
>
> You are an intelligent vision-language embodied agent skilled at solving tasks and answering questions in a 3D environment. Your job is to efficiently complete a specified task by choosing the optimal action at each timestep from a set of available actions. You are given a series of ego-centric images, and a history of previous actions with optional feedback (success/failure or human response). Each image shows what you see at a particular step in the action history, along with an extra image showing your current view.
>
> Current task:
> {task}
>
> Action history (action -> feedback):
> {action_history}
>
> Visual history:
> {image_history}
> Current view:
> {image}
>
> For the current step, your available options are listed as "[Option Number]. Content" as follows:
> {options}
>
> Choose your action from the above options by replying with "Thought: Your reasoning.\nChoice: [Option Number] (e.g. [1])".
>
> Note:
> - If the task needs more information of the scene, navigate wisely to the required targets (objects, places, or people).
> - Avoid repeated actions like useless forward motion and circling.
> - You can only interact with objects or humans (e.g. pick/place/open/close/handover) if they are within your view and very close to you.
> - You can only hold one object at a time. Put down any held object before picking up another.
> - Tasks containing "I" or "me" are requested by a person in the scene.
> - Reflect on why previous actions fail to avoid repeating mistakes and ajdust your current action.
> - You have a limited number of {max_steps} steps to complete the task.

### N.3 Prompt for the real-world experiment.

> **Instruction Prompt for real-world experiment.**
>
> Interact with a household to solve a task. You are an intelligent agent in a real-world household environment and your target is to perform actions to complete the task goal. At the beginning of your interactions, you will be given the goal to accomplish.
>
> For each of your turn, you will be given the observation of the last turn. You should choose from two actions: "Thought" or "Action". If you choose "Thought", you should first think about the current condition and plan for your future actions, and then output your action in this turn. Your output must strictly follow this format:"Thought: your thoughts. Action: your next action"; If you choose "Action", you should directly output the action in this turn. Your output must strictly follow this format:"Action: your next action".
>
> The available actions are:
>     1. look around
>     2. turn left
>     3. turn right
>     4. turn around
>     5. move forward {number} steps
>     6. move backward {number} steps
>     7. go to{recep}
>     8. take {obj} from{recep}
>     9. put {obj} in/on{recep}
>     10. open{recep}
>     11. close{recep}
>     12. use {obj}
>     13. clean {obj} with{recep}
>     14. heat {obj} with{recep}
>     15. done
> where {obj},{recep} and {number} correspond to objects, receptacles and the number.
>
> After your each turn, the environment will give you immediate feedback based on which you plan your next few steps. if the envrionment output "Nothing happens", that means the previous action is invalid and you should try more options.
>
> Reminder: 1. The action must be chosen from the given available actions. Any actions except provided available actions will be regarded as illegal.
> 2. Think when necessary, try to act directly more in the process.

### N.4 Prompt for the Multi-modal Generative Reward Model.

> **Instruction Prompt for MGRM**
>
> A conversation between User and Assistant. The user provides the task and the current state, and the Assistant evaluates the state and determine if the current state is success, failure, or continue. Failure means the user cannot achieve success through further actions, while continue means the user can still achieve success through additional actions. The assistant first thinks about the reasoning process in the mind and then provides the user with the answer. The reasoning process and answer are enclosed within <think> </think> and <answer> </answer> tags, respectively, i.e., <think> reasoning process here </think> <answer> Success/Failure/Continue </answer>.

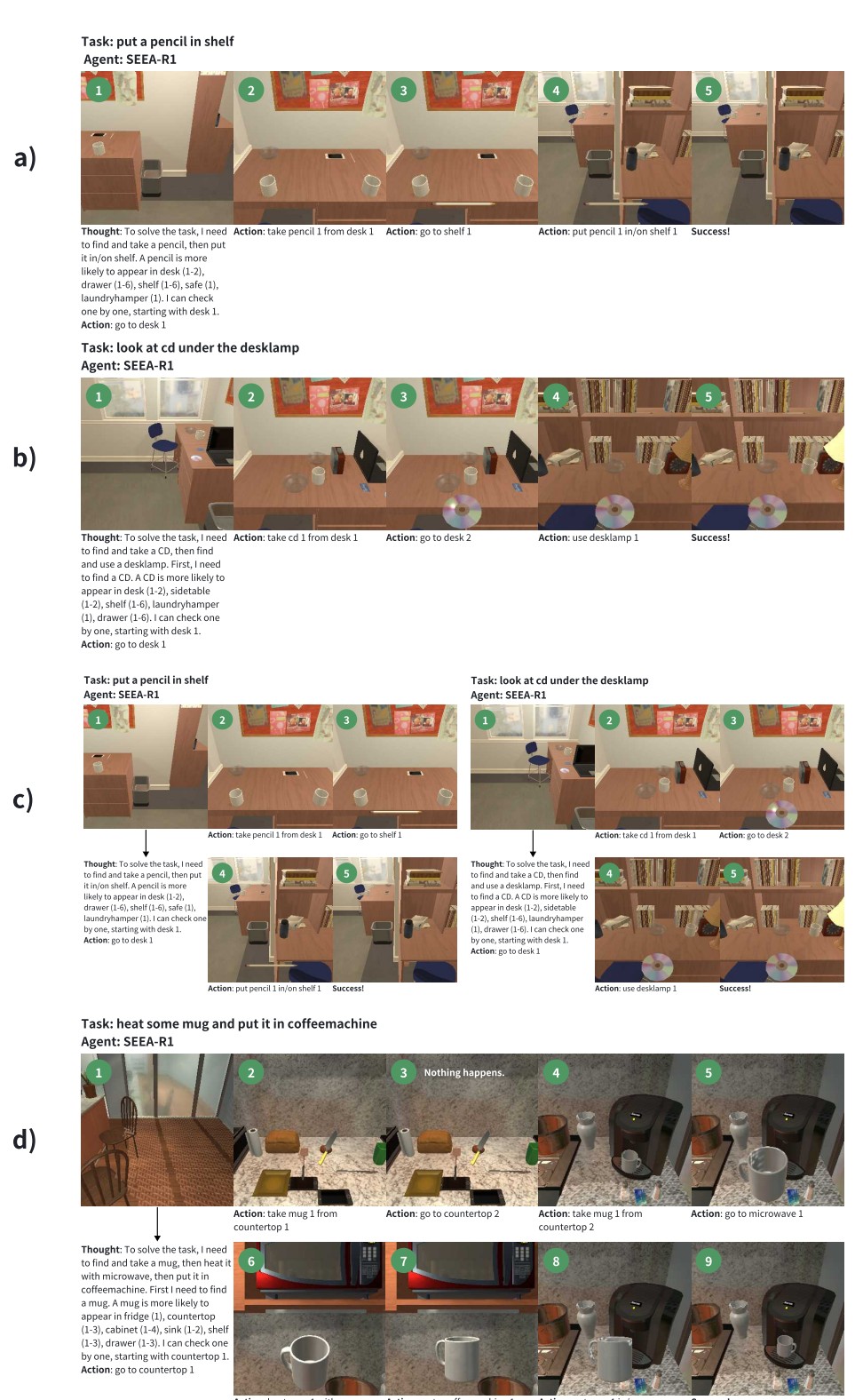

Figure 14: **ALFWorld Task Demonstrations (a–d).** (a) *Put a pencil in shelf* — step-by-step object manipulation. (b) *Put two CDs in safe* — compound goal handling. (c) *Look at object* — spatial and property-based reasoning. (d) *Heat mug and place in coffeemachine* — multi-step planning.

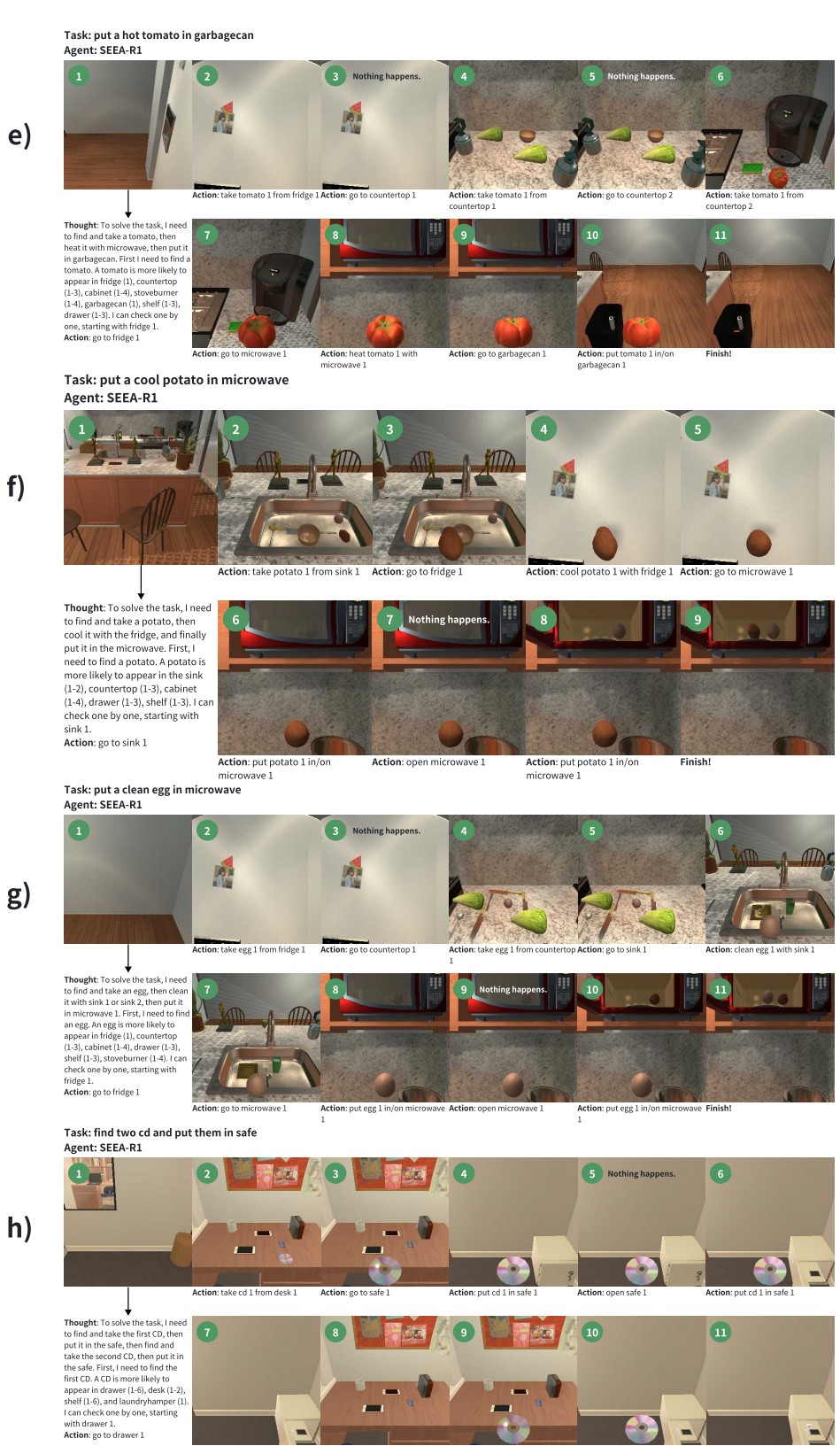

Figure 15: **ALFWorld Task Demonstrations (e–h).** (e) *Put hot tomato in trash* — handling state change before disposal. (f) *Cool an object* — inverse state manipulation. (g) *Clean the egg and microwave* — sequential object preparation. (h) *Pick and Look* — combined manipulation and inspection.

Task: put two cloth in shelf
Agent: SEEA-R1

Actor: None

Actor: Thought: To solve the task, I need to find and take the first cloth, then put it in shelf, then find and take the second cloth, then put it in shelf. First I need to find the first cloth. A cloth is more likely to appear in bathtubbasin (1), bathtub (1), toilet (1), garbagecan (1), toiletpaperhanger (1), shelf (1–3). I can check one by one, starting with bathtubbasin 1.
Action: go to bathtubbasin 1

Actor: Action: take cloth 1 from bathtubbasin 1

Actor: Action: go to shelf 1

Actor: Action: put cloth 1 in/on shelf 1

RM: None
Pred: Continue
GT: Continue

RM: <think>Since there are no visible cloths in the bathtub basin, I should move on to the next location to search for a cloth.</think>
<answer>Continue</answer>
Pred: Continue
GT: Continue

RM: <think>Now that the first cloth has been taken from the bathtubbasin, the next step is to put it in a shelf. Since the shelves are visible in the environment, the logical next action would be to go to one of the shelves.</think>
Pred: Continue
GT: Continue

RM: <think>Now that the first cloth has been taken from the bathtubbasin, the next step is to put it on the shelf. The observation shows that the shelf is empty except for the cloth already being held by the actor.</think>
<answer>Continue</answer>
Pred: Continue
GT: Continue

RM: <think>Since the task requires putting two cloths in the shelves, and the first cloth has already been taken and placed on shelf 1, the next step is to find and take the second cloth. The current observation does not show any other cloth in the room, so the next logical action would be to check the remaining shelves or other areas where a cloth might be located.</think>
<answer>Continue</answ...

Actor: Action: go to toilet 1

Actor: Action: take cloth 2 from toilet 1

Actor: Action: go to shelf 1

Actor: Action: put cloth 2 in/on shelf 1

Actor: Action: go to bathtub 1

RM: <think>Since the cloth was not found in the toilet, I should check other locations like the bathtubbasin, bathtub, garbagecan, or toiletpaperhanger for the second cloth. The next logical step would be to check the bathtubbasin again or another location that might contain a cloth.</think>
<answer>Continue</answer>
Pred: Continue
GT: Continue

RM: <think>Since the cloth was not taken from the toilet, it might be necessary to check other locations such as the bathtub or the garbagecan for the second cloth. However, given the current observation, it seems that the second cloth was not found in any of the locations checked so far.</think>
<answer>Continue</answer>
Pred: Continue
GT: Continue

RM: <think>So far, the task requires finding and putting two cloths into a shelf. The first cloth was found in the bathtubbasin and placed on the shelf. However, attempting to take another cloth from the toilet did not succeed, and placing the second cloth on the shelf also did not work. It seems that the second cloth might not be present in the toilet or there could be an issue with the interaction mechanics.</thi...

RM: <think>So far, the task involves finding and placing two cloths on a shelf. The first cloth has been found and placed on the shelf successfully. However, the second cloth could not be taken from the toilet or bathtub. It might be necessary to check other locations for the second cloth. The next step should involve checking another location where a cloth might be present.</think>
<answer>Continue</answ...

RM: <think>Since the cloth was taken from the bathtub but not placed on the shelf, it seems there might be an issue with the action command. However, the observation shows that the cloth is still in hand, indicating that it has been taken successfully. The next step should involve placing the cloth on the shelf.</think>
<answer>Continue</answer>
Pred: Continue
GT: Continue

Actor: Action: go to shelf 2

Actor: Action: put cloth 1 in/on shelf 2

Actor: Action: go to bathtubbasin 1

Actor: Action: take cloth 1 from bathtubbasin 1

Actor: Action: go to shelf 3

RM: <think>Since the task is to put two cloths in the shelf, and we have already taken the first cloth and placed it on the shelf, we need to take the second cloth. However, the action to take the second cloth from the toilet did not succeed as there was no cloth found. We should check other locations where a cloth might be present, such as the bathtub or any other location that has not been checked yet.</thi...

RM: <think>Since the task requires putting two cloths into the shelf, and the first cloth has already been placed on the shelf, the next step is to take another cloth. The observation shows that there is no cloth available in the bathtubbasin 1. Therefore, we might need to check other locations like the bathtub 1, toilet 1, or garbagecan 1 for the second cloth. However, since taking the cloth from the bathtubbasin 1 did not work,...

RM: <think>So far, the task involves finding and placing two cloths into the shelves. The first cloth has been placed successfully on shelf 1. The second cloth was taken from the toilet but not placed correctly, and then taken again from the bathtub but also not placed correctly. It seems there might be a problem with the placement action or the availability of the cloth in the correct location. Let's try another approach to ensure ...

RM: <think>So far, the first cloth has been taken from the bathtubbasin and placed on the shelf. The second cloth was taken from the toilet but could not be placed on the shelf. It seems that the second cloth might have been taken again from the bathtubbasin, which did not work either. The next step should involve trying another location for the second cloth.</think>
<answer>Continue</answer>
Pred: Continue

RM: <think>So far, the task has been partially completed. The first cloth was taken from the bathtubbasin and placed on the shelf. However, the second cloth could not be taken from the bathtubbasin or any other location. It seems that the cloth might not be present in the locations checked so far. The task requires finding and placing the second cloth, and since the second cloth could not be found, the task may not...

RM: <think>Since the cloth has not been placed on any shelf yet, I should try placing it on shelf 1 again as it might be possible that the previous attempt did not succeed due to some reason. If that does not work, I will try the other shelves.</think>
<answer>Continue</answer>
Pred: Continue
GT: Continue

RM: <think>So far, the actor has taken the first cloth from the bathtubbasin and placed it on shelves 1, 2, and 3. However, the task requires placing two cloths on the shelves. The actor has already placed one cloth on each of the shelves 1, 2, and 3. The next step should be to take another cloth and place it on one of these shelves. Since the actor has already tried taking a cloth from the bathtubbasin without success, the n...

RM: <think>So far, the task involves finding and putting two cloths into shelves. The first cloth has been found and placed in shelf 1, shelf 2, and shelf 3. However, the second cloth has not been found yet. The current observation shows that the cloth is not in the bathtubbasin 1. Since the cloth was previously taken from bathtubbasin 1 but not successfully placed in any shelf, it might be necessary to recheck the bathtubba...

RM: <think>So far, the actor has taken a cloth from the bathtub basin and tried to put it in shelves 1, 2, and 3 without success. The cloth is still in the actor's hand. The next logical step would be to try putting the cloth in another shelf that might not have been checked yet.</think>
<answer>Continue</answer>
Pred: Continue
GT: Continue

RM: <think>Since the cloth has not been placed on any shelf yet, it seems like there might be an issue with interacting with the shelves. It's possible that the action to place the cloth on the shelf is not functioning correctly, or the cloth might not be available in the bathtub basin. However, since the cloth is already taken, we should try to interact with the shelves again.</think>
<answer>Continue</answ...

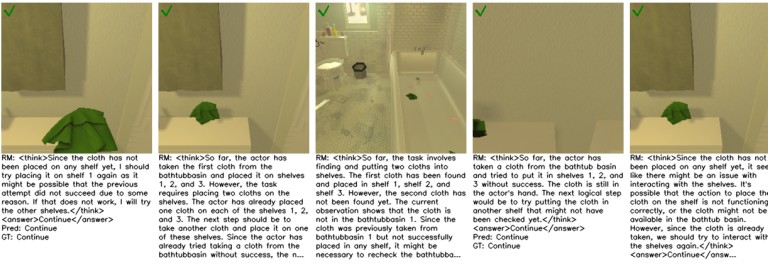

Figure 16: Policy and reward model interactions. Shows state, reward model thoughts, and actor actions step-by-step.

