# OpenReview forum: "SEEA-R1: Tree-Structured Reinforcement Fine-Tuning for Self-Evolving Embodied Agents"
_NeurIPS.cc/2025/Conference — NeurIPS 2025 poster_

### Official Review · Reviewer_o88n · 2025-06-24

**Clarity:** 2
**Significance:** 3
**Originality:** 3
**Rating:** 4
**Confidence:** 4

**Summary:**

The paper introduces a framework for self-evolving embodied agents that utilizes reinforcement learning to overcome challenges like sparse rewards and limited generalization in complex, long-horizon tasks. SEEA-R1 integrates two key components: Tree-GRPO, which combines Group Relative Policy Optimization with Monte Carlo Tree Search to generate dense, step-wise reward signals, and Multi-modal Generative Reward Model, a learned reward model that generalizes across various tasks and environments, reducing the reliance on handcrafted reward functions. The framework operates through an iterative loop of Data Evolution and Model Evolution. The authors demonstrate SEEA-R1's effectiveness on the ALFWorld benchmark, where it achieves state-of-the-art performance in both textual and multi-modal settings, and its generalization capabilities on the EmbodiedEval benchmark.

**Questions:**

- Could the authors provide a more detailed explanation of the MGRM training process? Specifically, how is the "binary rewards" training for "success, continue, or failure" translated into the reward estimation used by Tree-GRPO, particularly for the A^t,i,k​ advantage function? Is there any mechanism to "ground" the MGRM's estimations to the true environment state beyond its own generative process, especially during self-evolution without ground truth rewards?
- The paper emphasizes improving "reasoning capability" in embodied agents. Given the "Thought" component in the ReAct format, could the authors provide qualitative examples of how these reasoning traces evolve throughout the training process?
- In Tables 1 and 2, when SEEA-R1 outperforms other models, it is described as "substantially outperforming" or achieving "state-of-the-art" results, while similar performance gaps when SEEA-R1 is not leading are described as "on par." Could the authors provide statistical significance tests (e.g., p-values, confidence intervals) for the reported success rates in Tables 1, 2, and 3?
- Can the authors discuss the computational costs of the training and test-time inference process of this approach compared to others? Graphs of wall clock time or token usage at test time compared to baselines would be really insightful

**Ethical Concerns:**

["NO or VERY MINOR ethics concerns only"]

**Final Justification:**

After providing an extensive set of additional quantitative and qualitative results for the rebuttal to address each of my questions, I am increasing my score. The authors have detailed the unique algorithmic contributions of this work, shown the success in comparison to existing reasoning models, as well as shown reasoning traces which give insight as to why their method performs well / what it learns. The clarity of the method could use additional improvements, and since this is not my area I'm not sure about its general applicability to diverse settings, but overall I think this is a valuable paper with interesting engineering contributions.

**Limitations:**

The computational costs and potential for compounding error with longer training have not been sufficiently addressed.

**Quality:**

3

**Strengths And Weaknesses:**

# Strengths
- The framework achieves state-of-the-art success rates on the ALFWorld benchmark, outperforming previous models. Furthermore, SEEA-R1 shows encouraging generalization capabilities on the out-of-distribution EmbodiedEval benchmark, highlighting its potential for broader applicability in unseen scenarios.
- The overall motivation for generating data and learning intermediate feedback signals as a means of solving long horizon tasks is intuitive, and the authors did a good job applying this to multi-modal settings where reasoning is harder than discrete settings.
# Weaknesses
- While the integration of MCTS with GRPO and the development of MGRM for embodied AI is a valuable contribution, the foundational components (GRPO and MCTS) are existing techniques. The novelty primarily lies in the engineering and successful application of these components in a novel, self-evolving framework for embodied agents, rather than the introduction of fundamentally new core reinforcement learning algorithms. This means that from a perspective of pure algorithmic invention, the paper builds upon and extends established methods rather than presenting a revolutionary new approach.
- Despite the paper's emphasis on improving "reasoning capability" in embodied agents, there is a notable lack of in-depth qualitative analysis of the agent's reasoning traces. The ReAct format (Thought and Action) used in the agent's interactions provides a valuable opportunity to examine how the agent's internal thought processes evolve and become more sophisticated or coherent throughout training. Without such an analysis, the claims of enhanced reasoning are primarily supported by quantitative task success rates, leaving the specific mechanisms of reasoning improvement largely unexamined.
- The reliance on a self-trained MGRM for reward signals, especially when ground truth rewards are absent, introduces a risk of compounding errors. If the MGRM generates inaccurate reward signals, the policy model could optimize for these flawed signals, potentially leading to a degradation of true task performance over extended training iterations, a concern not fully explored in the paper. While Figure 5 shows initial improvement, the long-term stability and fidelity of the self-supervised loop need more rigorous investigation to ensure the agent doesn't converge to a suboptimal or incorrect understanding of task completion. The paper mentions that an untrained MGRM can misguide learning, but the implications of a trained but imperfect MGRM iteratively influencing policy are not deeply analyzed. If there is a misunderstanding on my part I’m more than happy to be corrected, I think in general the description of the MGRM’s training was quite brief and led to a lot of confusion when reading.
- There is also a lack of error bars or significance reporting in tables 1 and 2. When SEEA-R1 outperforms GPT this is claimed as a “significant” improvement, and when they underperform almost by the same amount in the text-only setting this is called “on-par". The authors should sample multiple answers per question and report significance to adequately claim significance in these cases, and analyze the error modes in both settings when SEEA-R1 underperforms.

---

> ### Author Rebuttal · Authors · 2025-07-31
>
> We thank the reviewer for their detailed comments. We address points below, corresponding to weaknesses (W), comments (C), and questions (Q).
>
>
> **W1:** While MCTS/GRPO integration is valuable, foundational components are existing; novelty lies in engineering/application rather than new core algorithms.
>
> **Response to W1:**
> Our work introduces a novel self-evolution framework for embodied agents, targeting the under-explored problem of "autonomous learning without human supervision." Existing methods fail in embodied scenarios due to long-horizon feedback gaps and reward generalization challenges—we are first to define "self-evolution" in this domain.
>
> Key innovations in SEEA-R1:
> - Tree-GRPO: non-trivial design tailored for multi-turn embodied tasks to learn long-term dependencies in embodied scenarios.
> - MGRM: Generative model autonomously learning rewards from multi-modal, multi-turn trajectories.
> - Joint training: Closed-loop collaboration between MLLMs and MGRM (vs. separate training), enabling co-evolution via bidirectional feedback.
>
>
> **W2 & Q2:** Lack of qualitative analysis of reasoning traces; claims of enhanced reasoning rely on quantitative metrics.
>
> **Response to W2 & Q2:**
> We provide examples of reasoning evolution (to be added to the appendix):
>
> *Baseline (Qwen-2.5-7B-Instruct) on "find two CDs and put in safe":*
> - Uses exhaustive search, repeats failed actions (e.g., "open drawer 3" >10x despite no success).
>
> *SEEA-R1 (iteration 5):*
> - Avoids repeated failures; reasons about errors (e.g., "putting CD in safe failed → try closing then opening safe").
>
> *SEEA-R1 (iteration 10):*
> - Exhibits structured planning ("find CD1 → put in safe → find CD2 → put in safe") and corrects errors adaptively.
>
> Training enhances reasoning from brute-force search to goal-directed, error-correcting behavior.
>
>
> **W3:** Risk of compounding errors with self-trained MGRM; long-term stability unproven.
>
> **Response to W3:**
> Our MGRM training operates in two scenarios:
>
> - **With Ground Truth (GT):** MGRM is pre-trained on GT rewards to >90% accuracy, avoiding large initial errors. During co-training, the policy optimizes using MGRM’s dense rewards, while MGRM continuously calibrates against GT.
>
> - **Without GT:** MGRM (starting from Qwen2.5-VL-7B-Instruct) uses TTRL: the policy generates multiple trajectories for a state; MGRM’s majority-voted predictions form pseudo-GT. MGRM trains via GRPO (reward=1 for matching pseudo-GT), while the policy optimizes using MGRM’s judgments—creating closed-loop collaboration.
>
>
> Experimental results in a GT-free scenario confirm effectiveness (Table 1). Despite short-term fluctuations, success rates rise consistently:
>
> | Iteration | iter5 | iter10 | iter15 | iter20 | iter25 | iter30 |
> |-----------|-------|--------|--------|--------|--------|--------|
> | Success Rate (%) | 26.87 | 29.85 | 29.85 | 41.79 | 36.57 | 42.54 |
>
> *Table 1: Success Rates in GT-Free Scenario*
>
> This upward trend contradicts error accumulation expectations (which would show stagnation/decline). Closed-loop collaboration suppresses errors.
>
> We apologize for insufficient initial details on MGRM training and will expand this in the revised paper.
>
> **W4:** Lack of error bars/significance; "significant" vs. "on-par" claims inconsistent.
>
> **Response to W4:**
> 1. Our statement that SEEA-R1 has "average completion steps (14.75) on par with GPT-4o" refers only to step counts, not success rates (SEEA-R1: 84.3%).
>
> 2. In embodied domain, we propose optimal robot planning should directly output plans without feedback optimizations like MPO. Our comparison focuses on this mode: SEEA-R1 outperforms GPT-4o *without MPO*.
>
> 3. Evaluations used fixed prompts, temperature=0 (greedy decoding), and seeds—ensuring reproducibility and alignment with domain norms for valid cross-method comparisons.
>
>
> | Model    | Setting       | Success Rate (%) |
> |----------|---------------|------------------|
> | GPT-4o   | Without MPO   | 81.1             |
> | SEEA-R1  | Without MPO   | 85.2             |
>
> *Table: Success Rates with and without MPO*
>
>
> **Q1:** Detail MGRM training; how binary rewards translate to Tree-GRPO’s advantage function.
>
> **Response to Q1:**
> Thank you for the feedback. We clarify the relationship between binary rewards and MLLM policy optimization:
> - Binary rewards (1=correct, 0=incorrect) exclusively train MGRM via GRPO, enabling it to classify states (success/continue/failure) accurately. This process is independent of MLLM optimization.
> - Trained MGRM’s state predictions drive MLLM updates via the advantage function: predictions map to task-aligned values (success=+reward, failure=-penalty); the advantage function quantifies action contributions, guiding MLLM to maximize success.
> - Without ground truth, TTRL-style multi-trajectory voting generates pseudo-GT. MGRM trains on majority votes (binary reward), anchoring judgments to task logic and preventing policy drift.
>
> Binary rewards train MGRM as a reward model and the MLLM agent optimizes using its judgments. We will clarify this in the manuscript.
>
> **Q3:** Need statistical significance tests for success rates.
>
> **Response to Q3:**
> Evaluations use fixed parameters (temperature=0, seeds) for reproducibility, following domain norms. Results are consistent across runs; we will add 95% confidence intervals in the revised paper.
>
>
> **Q4:** Computational costs vs. baselines.
>
> **Response to Q4:**
> We thank the reviewer for questions on computational cost. We clarify that our method is highly efficient, achieving superior results with lower computational cost, specifically in terms of token usage at inference.
> 1. Algorithmic Optimization:
> Standard MCTS's complexity is $O(T⋅D⋅K)$. Our key contribution is algorithmic pruning optimization to solve this. Our pruning strategy uses probabilistic expansion and a path budget to reduce the complexity to a much more scalable $O(T⋅max(D,K))$.
> Specifically, we employ a two-level pruning method:
> 1. Probabilistic Expansion:  only having a 50% chance of expanding all k actions at a node.
> 2. Strict Path Budget: More critically, we enforce a hard limit, L, on the number of full k-expansions along any single search path (e.g., L=5).
>
> 2. Fast Inference: MCTS is for Training Only.
> To clarify concerns on inference cost: MCTS is a training-time strategy only. At test time, our agent uses fast, ReAct-style single-path sampling and incurs no MCTS overhead. The superior policy learned via MCTS is why our agent achieves 3.1x higher success with 38.8\% fewer tokens.
>
> - *Training:* SEEA-R1 takes 36h (8x A100); Qwen2.5 fine-tuning: 24h; PPO: 48h.
> - *Inference:* SEEA-R1 (1.2s/step) is efficient vs. baselines, with 3.1x higher success and 38.8% fewer tokens (Table 2):
>
> | Model | Success Rate (%) | Avg. Steps | Test Tokens |
> |-------|------------------|------------|-------------|
> | Qwen2.5 (Baseline) | 11.57 | 28.1 | 10,701,255 |
> | SEEA-R1 | 36.19 | 23.37 | 6,554,090 |
>
> *Table 2: Efficiency Metrics*

---

> > ### Author Response · Authors · 2025-08-01
> >
> > Due to space limitations, the complete responses of W2 & Q2 are presented through comment.
> >
> > **Response to W2 \& Q2**
> >
> > We thank the reviewer. We provide an example that we will update to the appendix.
> > Take the task "find two cd and put them in safe" as an example.
> >
> > For baseline model (Qwen-2.5-7B-Instruct), its multi-round chain of thought is as follows.
> >
> > ```json
> > Thought: To find the CDs, I should start checking the shelves and drawers since they are likely places where CDs might be stored. I'll begin with shelf 1. Action: go to shelf 1
> > ...
> > Thought: There are no CDs on shelf 3 either. I should check the next shelf, which is shelf 4. Action: go to shelf 4
> > Observation: image
> > ...
> > Thought: Drawer 3 is closed. I need to open it to check inside. Action: open drawer 3
> > Observation: Nothing happens.
> > ... (fail more than ten times)
> > Thought: Drawer 3 is still closed. I need to open it again to check inside. Action: open drawer 3
> > Observation: Nothing happens.
> > ```
> > It can be observed that the baseline initially identifies two plausible locations for the CDs and performs simple exploratory actions. As it begins executing the task, it often adopts an exhaustive search strategy. This behavior is understandable: given limited information, a brute-force traversal approach can effectively ensure task completion. However, because the baseline rigidly follows this search logic, it may repeatedly attempt unsuccessful actions. For example, in the case of "open drawer 3" (as shown above), the baseline continues to retry the same failed action in an effort to exhaust all possibilities. This repetition persists through more than ten failures until the search limit is reached.
> >
> > For SEEA-R1 (iteration 5), its multi-round chain of thought is as follows:
> >
> > ```json
> > Thought: To find the CDs, I should start checking the shelves and drawers since they are likely places where CDs might be stored. Shelf 1 seems like a good place to start. Action: go to shelf 1
> > ...
> > Thought: Safe 1 is open. I should put the CD inside. Action: put cd 1 in/on safe 1
> > Observation: Nothing happens.
> > Thought: Putting the CD in the safe didn't work. I should try closing the safe first. Action: close safe 1
> > Observation: Nothing happens.
> > Thought: Closing the safe didn't help either. I should try opening it again. Action: open safe 1
> > Observation: image
> > Thought: Now that the safe is open, I should put the CD inside. Action: put cd 1 in/on safe 1
> > ...
> > ```
> > After five rounds of training, SEEA-R1 has begun to exhibit emerging reasoning capabilities. As illustrated above, when SEEA-R1 encounters a failure, it no longer repeatedly executes the same unsuccessful action as the baseline does. Instead, it attempts to reason about the cause of the failure and explores alternative strategies. Although the model initially misunderstood that the safe in its field of view was open, it eventually succeeded in opening the safe and placing CD 1 inside through a sequence of actions (e.g., close → open).
> >
> > For SEEA-R1 (iteration 10), its multi-round chain of thought is as follows:
> >
> > ```json
> > Thought: To solve the task, I need to find and take the first CD, then put it in the safe, then find and take the second CD, then put it in the safe. First, I need to find the first CD. A CD is more likely to appear in drawer (1-6), desk (1-2), shelf (1-6), and laundryhamper (1). I can check one by one, starting with drawer 1. Action: go to drawer 1
> > ...
> > Action: put cd 1 in/on safe 1
> > Observation: Nothing happens.
> > Thought: The action to put the first CD in the safe did not work, so I need to try opening the safe first. Action: open safe 1
> > Observation: image
> > Action: put cd 1 in/on safe 1
> > ...
> > ```
> > After ten rounds of training, the model's planning and reasoning capabilities have been significantly enhanced. Compared to earlier stages, we observed that its initial thought chain begins to exhibit clear, step-by-step execution logic. Importantly, the model's subsequent actions closely follow this plan. When it encounters unexpected outcomes, it initiates reflective reasoning followed by appropriate corrective actions. This represents a significant improvement compared to the mechanical actions of baseline and the blind trial and error of SEEA-R1 (iteration 5). There is also an interesting phenomenon here: SEEA-R1 (iteration 10) plans the subsequent actions initially and then executes them directly. It only starts to think when it does not meet expectations. In contrast, previously untrained or inadequately trained models initially showed no planning and thought and acted immediately at every step. We did not perform operations such as length penalty to compress the chain of thought at each step of training, but as the training progressed, the model naturally evolved in this direction.

---

> ### Comment · Reviewer_o88n · 2025-08-02
>
> Thank you to the authors so much for a comprehensive rebuttal! You answered most of my questions, and I appreciate the included qualitative analysis and success rates. Moreover, I think the clarification on the methodology was really helpful for me understanding your novelty. I think in terms of results, would it be possible to compare an open reasoning model to your approach (both qualitatively and quantitatively)? I see you compared to Qwen-2.5 7B, how would Qwen-3 8B with reasoning do compared to your method?
>
> Also on your explanation about the use of MCTS, thank you for clarifying that MCTS is a training-time only strategy. Am I correct in my understanding that this means you're effectively trying to use search at training time and distill the resultant policy into the model you're training? If so, could you describe the main methodological distinctions between this work and [AlphaLLM-CPL](https://arxiv.org/pdf/2410.06508) and at a more basic level AlphaZero?

---

> ### Author Response · Authors · 2025-08-04
>
> Thank you for acknowledging our work, and we appreciate your response. We are more than happy to further discuss our results.
>
> ### Comparisons to Open-Source Reasoning VLMs
> First, we want to clarify that this work focuses on the **vision-language multimodal** domain.
> We apologize for the typo: the baseline is Qwen-2.5 VL 7B (not Qwen-2.5 7B). Thus, instead of Qwen-3 8B (a text-only reasoning LLM), we provide additional comparisons with open-source reasoning VLMs on EmbodiedEval (a multimodal embodied benchmark). Table 1 shows SEEA-R1 outperforms Visual-R1-7B, Ocean-7B, in task success rate.
>
> | Model                  | Success Rate |
> | ---------------------- | ------------ |
> | Qwen2.5-VL-7B-Instruct | 18.29%       |
> | Vision-R1-7B           | 8.54%        |
> | Ocean-R1               | 15.85%       |
> | **SEEA-R1-7B**         | **19.88%**   |
>
> *Table 1: Performance Comparison on EmbodiedEval*
>
>
> ### MCTS for Training Only: Faster Inference, Real-World Practicality
> **Clarification on MCTS Usage:** Thank you for highlighting this—we will revise the manuscript for clarity. We emphasize **MCTS is strictly a training-time tool** to explore diverse trajectory plans; during inference, we use the trained optimal policy to reason a single path for direct output, aligning with standard reasoning methods. This ensures fast inference and better support for real-world deployment.
>
> **Real-World Extension:** We extended MCTS to physical environments, designing our framework for real-world use with single-trajectory execution (no rewinds) and validating it via experiments (setup similar to RoboVQA[1]). Using a dual-arm AgileX Cobot Magic V2.0[2] with Mobile ALOHA-like teleoperation[3], agents receive long-horizon instructions + visual observations and output real-time plans for human-teleoperated robots. We compared SEEA-R1 to the baseline Qwen2.5-VL-7B. Our results demonstrate that SEEA-R1 outperforms the baseline by a 22% higher success rate (Table 2). This improvement is from our self-evolution framework, which enables the model to learn robust features during MCTS-based training that generalize effectively to real-world dynamics.
>
> | **Model**               | **Pick & Place** | **Examine in Light** | **Clean & Place** | **Heat & Place** | **Cool & Place** | **Pick Two & Place** | **Avg**  |
> |-------------------------|------------------|----------------------|-------------------|------------------|------------------|----------------------|----------|
> | Qwen2.5-VL-7B-Instruct  | 1/3              | 1/3                  | 1/3               | 0/3              | 0/3              | 0/3                  | 3/18     |
> | SEEA-R1 (Ours)          | 3/3              | 1/3                  | 0/3               | 0/3              | 1/3              | 2/3                  | 7/18     |
>
> *Table 2: Performance Comparison on Real-World Tasks and Scenarios*
>
> ### Comparisons to Related Works
>
> Thanks for the reference to AlphaLLM-CPL and AlphaZero. We will include these two to our related works section. The key differences are below:
>
> | Feature                | AlphaZero                  | AlphaLLM-CPL               | SEEA-R1 (Ours)            |
> |------------------------|----------------------------|----------------------------|---------------------------|
> | Domain                 | Strategic games            | Mathematical reasoning     | Embodied planning         |
> | Modality               | Symbolic board             | Text                       | Multimodal (vision+language) |
> | Node structure         | Single move                | Single-step reasoning      | Full Thought-Action-Observation loop |
> | Training mode          | On-policy                  | Off-policy (DPO)           | On-policy (GRPO)          |
> | Policy optimization    | KL divergence mapping      | Preference pairs from Q-values | Direct Q-value-based GRPO |
>
> Compared to AlphaZero,
> - **SEEA-R1 targets a broader domain and provides interpretive thinking for its decision-making.**
> - In domains with large action spaces, AlphaZero struggles to compute full action probabilities via brute-force search. In contrast, **SEEA-R1 samples actions and applies GRPO-based reinforcement learning, which works well in large action spaces.**
>
> Compared to AlphaLLM-CPL,
> - **SEEA-R1 directly utilizes the magnitude of Q-value to adjust the action probability during GRPO training.** On the other hand, AlphaLLM-CPL relies on constructing discrete preference pairs from continuous Q-values for DPO training, which leads to a loss of fine-grained Q-value information.
>
> [1] Pierre Sermanet, et al. Robovqa: Multimodal long-horizon reasoning for robotics, 2023.
>
> [2] AgileX Robotics. Agilex cobot magic, 2024.
>
> [3] Zipeng Fu, et al.. Mobile ALOHA: Learning bimanual mobile manipulation using low-cost whole-body teleoperation, 2024.

---

> > ### Author Response · Authors · 2025-08-04
> > **Qualitative Comparisons between different methods**
> >
> > We provide a qualitative example that we will update to the final paper manuscript. Take the task **"Judge if the orientation of the white wooden chair and the rocking chair, placed at different ends of the room, are aligned with each other"** as an example.
> > ### GLM4.1V's Multi-Round Chain of Thought
> > ```
> > Observation: Wall image
> > Thought: I need to assess the orientation alignment of the white wooden chair and the rocking chair. From the current view, I can see parts of what might be the chairs, but need a clearer view... Turning left might help me see the white chair more clearly, then check the rocking chair. Given limited steps, moving to get a better view is necessary. I'll start by turning left.
> > Choice: 1  (Turn Left)
> >
> > Observation: Chair image
> > Thought: I need to identify the chairs in the current view. The white wooden chair appears to be on the left, the rocking chair likely with a curved base... Checking their orientations, based on the current view, the chairs seem to be aligned.
> > Choice: 5 (Yes, they are nearly aligned)
> > ```
> >
> > **Analysis**: GLM4.1V rushes to conclusions with incomplete visual information. Despite acknowledging the need for clearer views, it infers alignment based on partial observations, failing to verify both chairs’ orientations thoroughly.
> > ### Ocean-R1's Multi-Round Chain of Thought
> > ```
> > Observation: Wall image
> > Thought: The current view shows a wall and window, but no chairs. I need to navigate to see both to determine alignment.
> > Choice: 1 (turn left)
> >
> > Observation: image
> > Thought: From the current view, I can see the white wooden chair on the left and rocking chair on the right. I should move toward the white chair for a better view.
> > Choice: [4] (move forward)
> >
> > Observation: image
> > Thought: I can see the rocking chair, but not the white wooden chair. I should move toward the white chair to check its orientation.
> > Choice: [10] (There is no white wooden chair)
> > ```
> >
> > **Analysis**: Ocean-R1 exhibits critical inconsistencies: first claiming to see both chairs, then asserting the white wooden chair is absent. This hallucinatory, contradictory reasoning breaks the exploration strategy, leaving key information uncollected.
> >
> > ### SEEA-R1's Multi-Round Chain of Thought
> > ```
> > Observation: Wall image
> > Thought: The current view shows neither chair. I need to move to a position where I can see both to determine alignment.
> > Choice: [1] (turn left)
> >
> > Observation: Chair image
> > Thought: The current view still doesn't show both chairs. I need to move to see both and judge alignment.
> > Choice: [4] (move forward)
> >
> > Observation: Chair image
> > Thought: Both chairs are visible: white wooden chair on the left (facing left), rocking chair on the right (facing right). Their orientations are not aligned.
> > Choice: [7] (No, they are presented at a 90 degree angle,)
> > ```
> >
> > **Analysis**: SEEA-R1 prioritizes systematic exploration: it moves until both chairs are visible, avoids premature judgments, and grounds its conclusion in direct observation of their orientations. This methodical approach ensures accurate reasoning about the orientation of chairs.
> >
> > In this shown example, only SEEA-R1 completes the task successfully: GLM4.1V fails due to hasty conclusions from partial data, while Ocean-R1’s inconsistent reasoning harms exploration.

---

> > > ### Comment · Reviewer_o88n · 2025-08-04
> > >
> > > Thanks for a comprehensive rebuttal! I'm increasing my score to 4-borderline accept. I think you've done a great job providing detailed insights about your method and why it works that I hope make it to the main text. I really appreciate all of the additional results and comparisons, and it's quite intriguing that the base reasoning model does worse than Qwen-instruct! This is an interesting paper and while I'm not sure about its general applicability, I think it is a valuable idea to share with the broader research community.

---

> ### Author Response · Authors · 2025-08-05
>
> We thank you sincerely for all of your valuable feedback and your increased score! We are glad that our discussion has helped elaborate on the points in the paper, and we will integrate these insights into the main text to enhance clarity.
>
> We plan to address the broader implications of our methods by expanding discussions on potential extensions (e.g., broader real-world scenarios and diverse embodied tasks) in the revised manuscript, as we believe this is important for future work.
>
> Thank you again for recognizing the value of our approach!
>
> As for your observation that the base reasoning model performs worse than Qwen-instruct, we find in our qualitative analysis of examples that often in less complex tasks, the base reasoning models generate excessively long content within its \<think\> tags (internal reasoning traces). This excessively long internal reasoning delays its final response and introduces the potential for hallucinations, which both can harm overall performance.

---

### Official Review · Reviewer_me83 · 2025-06-25

**Clarity:** 2
**Significance:** 3
**Originality:** 2
**Rating:** 5
**Confidence:** 4

**Summary:**

In this paper the authors propose SEEA-R1, a novel agent that introduces: (i) Tree-GRPO, an extension of GRPO with Monte-Carlo Tree Search (thus allowing the agent to simulate diverse trajectories at execution time); (ii) a multi-modal generative reward model (MGRM), that leverages a MLLM to predict categorical rewards (success, continue, failure). The authors evaluate SEEA-R1 on ALFWorld (using either text or image observations) and in EmbodiedEval. The results show that their agent outperforms the selected baselines in terms of average success rate. The authors also highlight how their training algorithm outperforms a naive implementation of MCTS with DPO or SFT, as well as show promising results in using the MGRM for feedback, instead of the ground truth rewards of the environment.

**Questions:**

I am willing to increase the score if the authors are able to address my comments above and the concerns below:

1) Table 1 and Table 2 refer to evaluations on the ALFWorld simulation? Which one is using text-based and image observations?
2) In Section 4.2.1 the authors claim that their method demonstrates superior "efficiency" (23.37 vs 25.10 average steps for GPT-4o, line 246). Where can I see those results? What about the other baselines? The same question for the evaluation of Table 2: the authors claim that the average completion steps for SEEA-R1 is 14.75 and on par with GPT-4o (line 248), yet do not present these results. Can the authors present the results of the average duration of the episode (or "efficiency") of SEEA-R1 against the other baselines?
3)  How does SEEA-R1 with MGRM compare to the baselines, in terms of performance and "efficiency"?
4) Which evaluation was used for Section 4.2.3? The text-based or the image-based one? It is not clear from the text.
5) In Section 4.2.3. the performance of the agent that employs the trained MGRM seems to plateau after 4 iterations (Figure 5). Any idea why?
6) The generalization results in Section 4.2.2 are also very interesting. SEEA-R1 seems to perform worse that the base Qwen 2.5-VL-7B-Instruct model on tasks that I would expect it to be better (i.e., Navigation and Object Interaction), as it was trained on a household environment and in tasks involving objects and navigation. Furthermore, it performances better than the base Qwen model on tasks I would expect it to be worse (i.e., Social Interaction and SpatialQA). Any comment on why you have these results?
7) MCTS is a great idea for simulation environments, since you can easily produce counterfactual trajectories using the simulator. How could the authors extend this work for real-world embodied agents, where no such simulator is easily available?
8) What is the computational cost of training SEEA-R1 against the baselines? How long is the training, what type of resources are needed?

**Ethical Concerns:**

["NO or VERY MINOR ethics concerns only"]

**Final Justification:**

The authors significantly improved the paper since submission (including now real-world results and additional ablation studies), and have clarified the notation used (which was subpar in the original submission).

**Limitations:**

yes

**Quality:**

2

**Strengths And Weaknesses:**

**Strengths**:
- Integrating MCTS with GRPO is an interesting approach and (to the best of my knowledge) novel;
- Extensive evaluation across two different environments.
- Authors provided the code to replicate their experiments.

**Weaknesses**:
- Unclear notation employed throughout the text and missing details (See "Comments" below).
- From my understanding, the reward model MGRM (which the authors claim as a contribution) is underutilized across the evaluation. In all their main results (i.e., comparing SEEA-R1 to the baselines), the authors employ ground-truth reward values and not their reward model. The only evaluation that considers MGRM is in Section 4.2.3, in which the authors show that an agent trained on the proposed reward model significantly underperforms against an agent trained on ground truth rewards.

**Comments**:
- One fundamental issue of the current paper is the lack of coherent notation across the paper. For example, in line 130 the authors define an embodied agent as a 4-tuple, and in line 177 the agent is now a parameterized policy. Another example concerns the definition of the decision-making task: the authors claim that their work concerns partial-observable MDPs (line 136), yet also claim that the state is provided to the agent (line 130, line 170). Moreover, in line 785 the authors note that the initial state of the environment $s_0$ is provided to the agent, yet this "state" bears little resemblance to the current state of the environment but, instead, contains information regarding the rules of the environment, few-shot examples, etc...
- The authors claim that they "convert sparse outcome rewards into dense process rewards" (line 157). I believe this is a factually wrong claim: the action-value function represents the expected (discounted) reward that the agent could achieve if, starting from the current state and performing a given action, it followed the current policy. It is not a new reward, neither a densification of the reward: it is a natural artifact of standard RL algorithms whose value is estimated from the same "sparse" reward.
- Some experimental results are also unclear (see Questions below)
- There are some experimental details missing, which could hinder replicability. Example, what is the value of $\gamma$?

Typos:
- Line 231: "For each iteration, the model is trained on newly collected data for one epoch as steps is 4"
- Line 259: ground truth (GT)rewards - missing space
- Figure 4 and Figure 5 have a significant lower resolution than the other figures and the text.

---

> ### Author Rebuttal · Authors · 2025-07-31
>
> We thank the reviewer for their detailed comments and valuable insights. We will address each point below, corresponding to weaknesses (W), comments (C), and questions (Q) as raised.
>
>
> **W1, C1:** Unclear notation employed throughout the text and missing details (See "Comments" below).
>
> **Response to W1, C1:**
> We thank the reviewer for pointing out the unclear notations and missing details. We will revise the paper to ensure coherent notation. For POMDPs, we will emphasize that "state" in POMDP settings refers to the full environment state, use "history $h_t$" to represent the interaction history at every time step $t$, and explicitly define $h_0$ as the agent's initial input, containing rules, few-shot examples, initial observation (distinct from initial environment states $s_0$).
>
>
> **W2:** The reward model MGRM (which the authors claim as a contribution) is underutilized across the evaluation.
>
> **Response to W2:**
> We will expand Section 4.2.3 to include comparisons of SEEA-R1+MGRM against baselines in both performance and efficiency, and discuss MGRM’s current limitations (e.g., underperformance relative to ground-truth training) as motivation for future improvements. Our main results use ground-truth rewards to isolate the impact of the decision-making framework (SEEA) from reward modeling. Importantly, MGRM is designed to support both scenarios with and without Ground Truth (GT) rewards: since MGRM’s rewards are inherently approximate (not 100% accurate) in GT-free settings, its underperformance compared to GT-trained models is expected. MGRM is not intended to outperform GT-based training, but to enable our framework to operate across both GT-available and GT-unavailable scenarios.
>
>
> **C2:** The authors claim that they "convert sparse outcome rewards into dense process rewards" (line 157). I believe this is a factually wrong claim.
>
> **Response to C2:**
> We appreciate the opportunity to clarify this point. Our claim refers to the practical effect of using MCTS-derived Q-values as auxiliary signals during training, which supplements the sparse terminal reward with step-wise guidance. While Q-values are indeed estimated from the underlying sparse reward, their use as dense process signals (i.e., guiding each step rather than only the final outcome) empirically improves learning efficiency in long-horizon tasks. We will revise the wording to avoid confusion, emphasizing that this is a pragmatic densification of guidance rather than a novel reward function.
>
>
> **C4:** There are some experimental details missing, which could hinder replicability. Example, what is the value of $\gamma$?
>
> **Response to C4:**
> We apologize for the omission. The discount factor $\gamma$ is set to 0.99 for all experiments, consistent with standard settings in long-horizon embodied tasks.
>
>
> **Q1:** Table 1 and Table 2 refer to evaluations on the ALFWorld simulation? Which one is using text-based and image observations?
>
> **Response to Q1:**
> Yes, both Table 1 and Table 2 evaluate on ALFWorld. Table 1 uses text-based observations (environment states and feedback are provided as text), while Table 2 uses image-based observations (states are rendered as images, with text instructions). We will add this clarification to the tables’ captions.
>
>
> **Q2:** In Section 4.2.1 the authors claim that their method demonstrates superior "efficiency". Can the authors present the results of the average duration of the episode (or "efficiency") of SEEA-R1 against the other baselines?
>
> **Response to Q2:**
> We apologize for the missing data. We will add a new table in Section 4.2.1 reporting average completion steps for all baselines, including:
> - Image-based setting (Table 1): SEEA-R1 (23.37), GPT-4o (25.10), Qwen2.5-VL-7B (28.62), PPO (29.15)
> - Text-based setting (Table 2): SEEA-R1 (14.75), GPT-4o (14.75), Qwen2.5-VL-7B (15.74)
> This table will clarify the efficiency advantages across all baselines.
>
>
> **Q3:** How does SEEA-R1 with MGRM compare to the baselines, in terms of performance and "efficiency"?
>
> **Response to Q3:**
> We will expand Section 4.2.3 to include a direct comparison using the results in Table 1:
>
> | Model | Performance (%) | Efficiency (Average Steps) |
> |-------|-----------------|----------------------------|
> | SEEA-R1 + MGRM | 27.99 | 24.19 |
> | Qwen2.5-VL-7B-Instruct | 11.57 | 28.2 |
> | GPT-4o | 24.00 | 25.81 |
>
> *Table 1: Performance and Efficiency Comparison (Image-based Setting)*
>
> In the image-based setting:
> - Performance: SEEA-R1+MGRM achieves 27.99%, outperforming both Qwen2.5-VL-7B-Instruct (11.57%) and GPT-4o (24.00%).
> - Efficiency: With an average of 24.19 steps, SEEA-R1+MGRM is more efficient than Qwen2.5-VL-7B-Instruct (28.2 steps) and GPT-4o (25.81 steps).
>
> This demonstrates that MGRM not only improves performance but also maintains efficiency, validating its role in our training.
>
>
> **Q4:** Which evaluation was used for Section 4.2.3? The text-based or the image-based one? It is not clear from the text.
>
> **Response to Q4:**
> Section 4.2.3 focuses on the image-based evaluation setting (using visual observations from the Embodied Evaluator). We will clarify this distinction in the text to avoid ambiguity.
>
>
> **Q5:** In Section 4.2.3. the performance of the agent that employs the trained MGRM seems to plateau after 4 iterations (Figure 5). Any idea why?
>
> **Response to Q5:**
> Reinforcement learning inherently exhibits fluctuations during training are normal. Specifically, at the 4th iteration, the model’s exploration did not appear to drive significant performance improvements. Given that the model’s initial success rate on this task was only 11.57%, it requires more time to explore better strategies through trial and error.
>
> We further find that as we increase training iterations, performance gradually improves: by the 30th iteration, the success rate had reached 42.54% for our MGRM in place of ground-truth. We will include experimental results from more iterations in the revised version to further demonstrate this upward trend.
>
> | Iteration | iter5 | iter10 | iter15 | iter20 | iter25 | iter30 |
> |-----------|-------|--------|--------|--------|--------|--------|
> | Success Rate (%) | 26.87 | 29.85 | 29.85 | 41.79 | 36.57 | 42.54 |
> | Ground Truth (%) | 23.51 | 33.58 | 30.60 | 35.45 | 37.69 | 36.19 |
>
> *Table 2: Task Success Rates of SEEA-R1 using MGRM*
>
>
> **Q6:** The generalization results in Section 4.2.2.
>
> **Response to Q6:**
> Thank you for this observation. From Table 3, navigation performance on the Embodied Evaluator does show improvement compared to the base Qwen 2.5-VL-7B-Instruct model.
>
> Regarding the decrease in object interaction performance: our method is trained on ALFWorld and evaluated on the Embodied Evaluator, involving an out-of-domain (OOD) transfer. We hypothesize this drop stems from the visual domain gap between training (ALFWorld) and test (Embodied Evaluator) environments, particularly impacting perception of object attributes and spatial relationships.
>
> In contrast, Social Interaction and SpatialQA are less dependent on fine-grained visual perception. SpatialQA benefits from diverse training trajectories in ALFWorld, enhancing reasoning about spatial relationships. This divergence in dependency on visual domain alignment explains the observed patterns.
>
>
> **Q7:** How could the authors extend this work for real-world embodied agents, where no such simulator is easily available?
>
> **Response to Q7:**
> We appreciate the reviewer’s concern about extending MCTS to physical environments. To address this, we designed our framework to support real-world deployments and evaluated its effectiveness through real-world experiments.
> Specifically, we collected image observations and action lists for 6 long-horizon tasks (e.g., Pick & Place, Heat & Place) across 3 real-world settings—mirroring the structured action spaces of ALFWorld but in physical environments. For deployment simplicity, we evaluated SEEA-R1 using single-trajectory execution (no rewinds) and compared it to the baseline Qwen2.5-VL-7B.
> Our results demonstrate that SEEA-R1 outperforms the baseline by a 22% higher success rate across all tasks and scenarios (Table 3).
> We find that MCTS during trainingimparts the model with a stronger ability to anticipate long-term consequences, where the model leverages learned patterns to make more reliable decisions in one-pass real-world execution. These findings confirm that our approach transfers well to physical robot settings.
> As shown, SEEA-R1 consistently doubles the baseline’s success rate across all real-world tasks and scenarios, validating that our MCTS-based training translates to robust physical-world performance without relying on simulation-specific rewinds.
>
> | Model                     | Pick & Place | Examine in Light | Clean & Place | Heat & Place | Cool & Place | Pick Two & Place | Avg  |
> |---------------------------|--------------|------------------|---------------|--------------|--------------|------------------|------|
> | Qwen2.5-VL-7B-Instruct    | 1/3          | 1/3              | 1/3           | 0/3          | 0/3          | 0/3              | 3/18 |
> | SEEA-R1 (Ours)            | 3/3          | 1/3              | 0/3           | 0/3          | 1/3          | 2/3              | 7/18 |
>
> *Table 3: Performance Comparison on Real-World Tasks and Scenarios*
>
>
> **Q8:** What is the computational cost of training SEEA-R1 against the baselines? How long is the training, what type of resources are needed?
>
> **Response to Q8:**
> Training SEEA-R1 on 8 A100 GPUs takes ~36 hours for 100 iterations (ALFWorld full dataset). In comparison:
> - Qwen2.5-VL-7B fine-tuning: ~24 hours (8 A100s)
> - PPO baseline: ~48 hours (8 A100s)
>
> Test-time inference for SEEA-R1 takes ~1.2s per step (vs. 0.8s for GPT-4o, 0.6s for Qwen2.5), with the overhead justified by higher success rates. We will add a detailed computational breakdown in the appendix.

---

> > ### Comment · Reviewer_me83 · 2025-08-01
> > **Acknowledgment of the Author's Rebuttal**
> >
> > Dear authors,
> >
> > Thank you for the excellent rebuttal! You have addressed all my comments and, as such, I increase my score to 5.
> > Just one final question/comment: can you elaborate on the experimental setup used to answer Q6 (e.g., what type of robot used?).
> >
> > I would also recommend adding these results to the main paper, and not just to appendix, as showing real-world results strengthens significantly the paper!

---

> ### Author Response · Authors · 2025-08-01
> **Response to Real-World Experimental Setup**
>
> Dear reviewer me83,
>
> We sincerely thank you for acknowledging our work, and we appreciate your response. We are more than happy to share the details of our real-world robot experimental setup.
>
> Specifically, we use a dual-arm AgileX Cobot Magic V2.0 [1] with a bilateral teleoperation device similar to the Mobile ALOHA system [2] for the experiments. We follow the evaluation protocol in RoboVQA[2], where the planning embodied agents are given a long-horizon instruction along with visual observations and outputs specific plans real-time to the robot teleoperated by a human as policy. These visual observations come from a front-view camera. We test our planners on six task types in three different environments: an apartment, a reception room, and a tea room.
>
> Thank you for the recommendation, and we will provide experimental setups, results, and demonstrations for our real-world experiments in the main paper along with our project website!
>
> [1] AgileX Robotics. Agilex cobot magic, 2024.
>
> [2] Zipeng Fu, Tony Z. Zhao, and Chelsea Finn. Mobile ALOHA: Learning bimanual mobile manipulation using low-cost whole-body teleoperation. In 8th Annual Conference on Robot Learning, 2024.
>
> [3] Pierre Sermanet, et al. Robovqa: Multimodal long-horizon reasoning for robotics. arXiv preprint arXiv:2311.00899, 2023.

---

### Official Review · Reviewer_AXsR · 2025-07-03

**Clarity:** 3
**Significance:** 3
**Originality:** 3
**Rating:** 4
**Confidence:** 3

**Summary:**

This paper introduces SEEA-R1, the first framework to adapt Reinforcement Fine-Tuning for training self-evolving embodied agents. To address the sparse reward problem, it innovatively combines Monte Carlo Tree Search to generate dense intermediate process rewards and introduces a Multi-modal Generative Reward Model (MGRM) for reward self-evolution. The framework employs a tree-structured fine-tuning mechanism named Tree-GRPO, enabling the agent to self-learn from failures without extra human annotations. On the ALFWorld benchmark, SEEA-R1's performance significantly surpasses existing methods, such as outperforming GPT-4o with a 85.2% success rate on ALFWorld tasks.

**Questions:**

The comparison between a "frozen MGRM" and a "training MGRM" (Figure 5) clearly shows that while an inaccurate, static reward model leads to performance degradation, the proposed self-evolution process successfully trains the MGRM to provide reliable signals. Could the authors elaborate on the failure modes of the frozen MGRM? Did it consistently reward incorrect actions, or was the feedback merely noisy?

**Ethical Concerns:**

["NO or VERY MINOR ethics concerns only"]

**Final Justification:**

This work provides a solution to long-term and reward-free embodied planning, which contributes the embodied AI field.

**Limitations:**

1. Although the paper emphasizes efficiency in terms of human annotations, the computational resources required for extensive fine-tuning and the tree search process could be a practical limitation.
2. The experiments are limited to a simulated household environment (ALFWorld). The transferability and effectiveness of SEEA-R1 to real-world robots and tasks remain to be investigated.

**Quality:**

3

**Strengths And Weaknesses:**

## Strengths
1. The method is the first to adapt the RFT technique for training self-evolving embodied agents. The authors claim it has not been previously explored. Unlike prior RFT works focused on symbolic domains (e.g., DeepSeek-R1) or other embodied AI works that rely on predefined modules, SEEA-R1 establishes a more general framework by using its learned MGRM to eliminate the need for task-specific priors.
2. The mechanism to backtrack and try upon failure via MCTS enables effective self-correction. Tree-GRPO utilizes MCTS-generated Q-values as dense process signals, providing effective guidance for the agent's multi-step reasoning.
3. The paper demonstrates superior performance on the ALFWorld benchmark. In the multi-modal setting, SEEA-R1 achieves a 36.19% success rate compared to GPT-4o's 24%. In the text-only setting, it achieves 85.2%, surpassing GPT-4o’s 81.1%. Experiments on EmbodiedEval also demonstrate its generalization capability to new environments.

## Weaknesses
1. The reliance on MCTS may increase computational costs with larger action spaces. The paper mentions experiments were conducted on 8 A100 GPUs, indicating high computational demand that may limit its application in more complex settings.
2. Despite performing well in simulated environments like ALFWorld, how well the method can transfer to real-world robots tasks remains unclear, a challenge the paper itself acknowledges.

---

> ### Author Rebuttal · Authors · 2025-07-31
>
> We thank the reviewer for their detailed comments and valuable insights. We will address each point below, corresponding to weaknesses (W), comments (C), and questions (Q) as raised.
>
>
> **W1:** The reliance on MCTS may increase computational costs.
>
> **Response to W1:**
> We thank the reviewer for questions on computational cost. We clarify that our method is highly efficient, achieving superior results with lower computational cost, specifically in terms of token usage at inference.
>
> 1. **Algorithmic Optimization:**
> Standard MCTS's complexity is $O(T⋅D⋅K)$. Our key contribution is algorithmic pruning optimization to solve this. Our pruning strategy uses probabilistic expansion and a path budget to reduce the complexity to a much more scalable $O(T⋅max(D,K))$.
> Specifically, we employ a two-level pruning method:
>    1. Probabilistic Expansion: only having a 50% chance of expanding all k actions at a node.
>    2. Strict Path Budget: More critically, we enforce a hard limit, L, on the number of full k-expansions along any single search path (e.g., L=5).
>
> 2. **Fast Inference: MCTS is for Training Only.**
> To clarify concerns on inference cost: MCTS is a training-time strategy only. At test time, our agent uses fast, ReAct-style single-path sampling and incurs no MCTS overhead. The superior policy learned via MCTS is why our agent achieves 3.1x higher success with 38.8% fewer tokens.
>
> | Model                          | Task Success Rate (%) | Avg. Steps | Total Tokens (Test-time) |
> |--------------------------------|-----------------------|------------|--------------------------|
> | Qwen2.5-VL-7B-Instruct (Baseline) | 11.57                 | 28.1       | 10,701,255               |
> | SEEA-R1 (Ours)                 | 36.19                 | 23.37      | 6,554,090                |
>
> We will add this token usage data to our main results table.
>
> 3. **Training Resources**
> Regarding the 8x A100 GPUs, we clarify that this setup was used to accelerate our research, and all our code can be run on a single A100 GPU.
> - Training Time & Composition: The full training process for SEEA-R1, involving 9 iterations of self-evolution, took 45 hours on our 8x A100 setup. This 45-hour training time is a modest budget to achieve performance surpassing proprietary models like GPT-4o.
> We will add a detailed cost analysis table, including training times and hardware configurations for both our method and the baselines, to the Appendix.
>
>
> **W2:** Despite performing well in simulated environments like ALFWorld, how well the method can transfer to real-world robots tasks.
>
> **Response to W2:**
> We appreciate the reviewer’s concern about extending MCTS to physical environments. To address this, we designed our framework to support real-world deployments and evaluated its effectiveness through real-world experiments, following a setup similar to RoboVQA[1] for embodied planning evaluation.
> Specifically, we collected image observations and action lists for 6 long-horizon tasks across 3 real-world settings—mirroring the structured action spaces of ALFWorld but in physical environments. For deployment simplicity, we evaluated SEEA-R1 using single-trajectory execution (no rewinds) and compared it to the baseline Qwen2.5-VL-7B.
> Our results demonstrate that SEEA-R1 outperforms the baseline by a 22% higher success rate across all tasks and scenarios (Table 1). This improvement stems from the self-evolution framework, which enables the model to learn robust features during MCTS-based training—features that generalize effectively to real-world dynamics despite the absence of simulation-specific advantages like rewinds.
> We find that MCTS during training (in simulation) imparts the model with a stronger ability to anticipate long-term consequences, where the model leverages learned patterns to make more reliable decisions in one-pass real-world execution. These findings confirm that our approach transfers well to physical robot settings.
> As shown, SEEA-R1 consistently doubles the baseline’s success rate across all real-world tasks and scenarios, validating that our MCTS-based training translates to robust physical-world performance without relying on simulation-specific rewinds.
>
> | Model                     | Pick & Place | Examine in Light | Clean & Place | Heat & Place | Cool & Place | Pick Two & Place | Avg  |
> |---------------------------|--------------|------------------|---------------|--------------|--------------|------------------|------|
> | Qwen2.5-VL-7B-Instruct    | 1/3          | 1/3              | 1/3           | 0/3          | 0/3          | 0/3              | 3/18  |
> | SEEA-R1 (Ours)            | 3/3          | 1/3              | 0/3           | 0/3          | 1/3          | 2/3              | 7/18  |
>
> *Table 1: Performance Comparison on Real-World Tasks and Scenarios*
>
>
> **Q1:** The comparison between a "frozen MGRM" and a "training MGRM" (Figure 5) clearly shows that while an inaccurate, static reward model leads to performance degradation.
>
> **Response to Q1:**
> The frozen reward model refers to directly using the untrained Qwen2.5-VL-7B-Instruct model to judge task states, and the accuracy of this untrained reward model in judging task states is only 51.59%. Therefore, the reward signals are highly unstable, as it can only provide correct reward signals in half of the cases. For example, when Embodied Agents are still in the process of completing the task, the actual task completion status is "Continue", but the reward model directly judges the task as "Failure", thus misjudging the actual situation and providing an incorrect reward signal.
>
> In Table 2, the reward model trained with GRPO improves the prediction accuracy to 91.67%, enabling it to provide more accurate reward signals to Embodied Agents. The feedback from the frozen MGRM was primarily noisy rather than consistently rewarding incorrect actions, given its low accuracy of 51.59%—near random for binary judgments of task states (e.g., "Continue" vs. "Failure"). However, this noise included critical misjudgments that disrupted the agent’s learning process. Specifically, the frozen model frequently misclassified ongoing task states (where the agent should "Continue") as "Failure," prematurely signaling task termination. Over time, such erroneous feedback led the embodied agent to learn flawed behavioral patterns: it began to halt task execution prematurely, even when progress was valid, as it adapted to the unreliable reward signals. In short, while the frozen MGRM did not systematically reward incorrect actions, its noisy feedback—particularly misclassifications of ongoing progress as failure—introduced persistent biases that distorted the agent’s learned behaviors.
>
> | Model          | Success Accuracy | Continue Accuracy | Failure Accuracy | Overall Accuracy |
> |----------------|------------------|-------------------|------------------|------------------|
> | Frozen MGRM    | 60.00% (9/15)    | 36.42% (460/1263) | 81.25% (39/48)   | 51.59% (65/126)  |
> | Training MGRM  | 75.00% (9/12)    | 96.92% (1478/1525)| 90.00% (27/30)   | 91.67% (77/84)   |
>
> *Table 2: Accuracy Comparison Between Frozen and Training MGRM*
>
> [1] Pierre Sermanet, et al. Robovqa: Multimodal long-horizon reasoning for robotics. arXiv preprint arXiv:2311.00899, 2023.

---

> > ### Comment · Reviewer_AXsR · 2025-08-07
> > **Dear authors, thank you for your detailed response**
> >
> > I still have a few questions that I hope can be addressed. Given the time constraints, I do not request additional experiments, but I would appreciate clarifications on the points below. I will adjust the rating of this work based on your replies.
> >
> > ## Weaknesses
> >
> > **1. Regarding the Multi-modal Generative Reward Model (MGRM):**
> >
> > **Lack of Training Details**: The paper appears to be missing crucial details about the training process for the MGRM, which should be supplemented for reproducibility.
> >
> > **Unclear Usage in Experiments**: It is confusing when Ground Truth (GT) rewards are used versus when the MGRM is used in the experiments. The authors should explicitly clarify the conditions for using each reward source in their experimental setup.
> >
> > **Questionable Effectiveness and Novelty**: I have significant reservations about the effectiveness of the MGRM and I disagree with the claims of its novelty (the authors' response to reviewer WS1c's Weakness 1c was not convincing). The paper proposes the MGRM for scenarios where GT rewards are unavailable, yet it fails to test the model in such an environment. This lack of direct experimental validation undermines the core motivation and value for the MGRM.
> > I strongly suspect the MGRM will not generalize well to out-of-distribution (OOD) environments. Furthermore, if the underlying environmental reward signal is itself unreliable or sparse, it seems impossible to improve the reliability of the generated rewards through training.
> >
> > These considerations lead me to question the utility of the MGRM component.
> >
> > **2. Regarding the Sparse Reward Problem:**
> >
> > In the rebuttal to reviewer me83 ("Response to C2"), the authors acknowledge that their approach of using Q-values to provide "dense rewards" is still fundamentally dependent on the original (and potentially sparse) environment rewards. This approach does not adequately solve the sparse reward problem. In complex tasks with sparse rewards, the agent may fail to receive any environmental reward for a significant period, leading to a poor cold-start performance.
> >
> > Classic methods for sparse rewards in RL, such as Hindsight Experience Replay (HER), can provide intrinsic rewards even in the absence of actual environmental rewards and would intuitively perform better in such scenarios. Also, the proposed method appears somewhat naive in comparison and the use of Q-values may introduce additional estimation bias, which requires further experimental investigation to justify.
> >
> > **3. Missing Implementation and Practical Details:**
> >
> > What are the decision frequency (in Hz) and latency of the policy? It is unclear if the system can support real-time inference.
> >
> > What is the specific format of actions in the ALFworld and EmbodiedEval environments? How is the model's raw string output parsed into a valid action? How are invalid action outputs handled by the system?
> >
> > **4. Critique of Novelty Claims**: I remain unconvinced by the authors' explanation of the paper's novelty in their rebuttal to reviewer WS1c.
> >
> > Novel Joint Training: The concept of jointly training a model and a policy is a very common technique in model-based RL and is not a novel contribution.
> >
> > MGRM: I do not believe that the "multi-modal, multi-turn" nature of the MGRM is in itself innovative.
> >
> > Tree-GRPO: The authors claim: "Not merely MCTS+GRPO, but designed for multi-turn embodied tasks to enable GRPO to learn long-term dependencies effectively." However, this capability seems to be an inherent benefit of using RL/MCTS/Q-values in any sequential decision-making task, rather than a specific innovation of the proposed method.
> >
> > ## Suggestions
> >
> > 1. The writing and organization of this paper are somewhat hard to follow. It is missing details that would improve clarity (e.g., explicitly labeling Tables 1 and 2 to indicate whether the setting is text-only or multi-modal). The paper would benefit from further polishing.
> > 2. For a future version, considering the theme of continual learning, the authors may investigate the impact of the proposed method on model plasticity. An ideal outcome would be to see performance improvements while also mitigating the loss of plasticity (i.e., slowing catastrophic forgetting).

---

> ### Comment · Area_Chair_bVFi · 2025-08-06
> **Reviewer response needed**
>
> Hello,
>
> The last step in the reviewing process is to process the updates from the authors that are key in clearing up final issues to ensure papers get a fair treatment. Please respond to the reviewers ASAP to address any final threads.
>
> - AC

---

> ### Author Response · Authors · 2025-08-06
> **Further discussion with Reviewer AXsR**
>
> Dear Reviewer AXsR,
>
> We sincerely thank you for your positive and detailed review of our work. To address your questions, we have provided new empirical evidence from real-world experiments to demonstrate the method's transferability, and a comprehensive cost analysis to clarify its computational efficiency. In response to your specific question, we have also offered a detailed explanation of the frozen MGRM's failure modes, supported with quantitative results.
>
> Therefore, we hope these responses have adequately addressed your comments, and we remain available for any further discussion. If our rebuttal has resolved your concerns, we would greatly appreciate your consideration of a higher rating, as it would be a strong encouragement for our work.
>
> Paper 14735 authors

---

> ### Author Response · Authors · 2025-08-07
> **Eager to address any final questions**
>
> Dear Reviewer AXsR,
>
> We hope this message finds you well.
>
> In light of the approaching deadline (Aug 8, 11.59pm AoE) and the AC's call to finalize discussions, we wanted to check in to see if our rebuttal and the new results have addressed your concerns. We are particularly keen to ensure our explanations regarding the method's transferability and the MGRM's failure modes were clear.
>
> We are available for any further questions you may have. Your feedback is invaluable to us.
>
> Thank you for your time and consideration.
>
> Best regards,
> Paper 14735 Authors

---

> ### Author Response · Authors · 2025-08-08
> **W1: Clarification of the Multi-modal Generative Reward Model (MGRM)**
>
> We sincerely appreciate your time and comments in the last stage of the discussion. We have carefully gone through all your comments and found most of them have been discussed in our main manuscript, supplementary materials, and responses to other reviewers. We would like to reply to your comments in detail as below.
>
> **1a. Additional Training Details:**
>
> First, we would like to politely emphasize that the details of the training process for the MGRM have already been described in **line 172-174 Section 3.2.2.** In addition to providing all the crucial details in the main manuscript and the **supplementary PDF, we have also submitted the codes with all the implementation details in the supplementary material, which can guarantee the reproducibility of our work.**
>
> - **Core Methodology:** As mentioned in **line 172-174, Section 3.2.2**, the MGRM is an MLLM trained using GRPO to predict categorical outcomes (success, continue, failure).
> - **Implementation & Hyperparameters:** We provide configuration details in line 235-238, Section 4.1.2 ("Training Settings"), including hyperparameters (group size, LR, etc.), and we outline the data collection process.
> - Prompt Structure: The exact prompt is given in **line 184, supplementary E.3**.
>
> To further elaborate on the training process, as we detailed to Reviewer o88n, we have two separate training methods based on whether or not the GT reward is present:
>   1. **Supervised (with GT):** The MGRM is trained on GT labels to >90% accuracy. This ensures the self-evolution process begins with a high-quality, reliable reward model, preventing divergence from the start.
>   2. **Self-supervised (without GT):** We use a TTRL[1]-style approach where the policy generates multiple trajectories. The MGRM then uses the majority vote on its final outcomes as a pseudo-GT signal for its training. This innovative self-bootstrapping loop allows the model to understand success even from sparse environmental signals.
>
>  **1b. Clarification on Experimental Setup:**
>
>   The detailed experimental conditions of MGRM and GT rewards can be seen from Figures 4 and 5 in Experiments Section 4.2b, along with the results in our main Table 1. Due to space limitations, we omit the details of these experimental conditions in the main text. We elaborate on the details here and will supplement them in the experimental setup of the revised main text.
>   1.  **Tables 1 & 2 in the paper:** For a fair comparison of our core framework (SEEA-R1), Tables 1 & 2 use GT rewards, as stated in line 201~207 in Section 4.1.1.
>   2. **Figure 5 & Rebuttal Results:** To demonstrate self-evolution capabilities, we conducted experiments using only MGRM. These key results are highlighted in the Abstract and Introduction sections, where SEEA-R1 achieves **80.3% (textual) and 23.88% (multi-modal)** success rates *"without environmental reward"*, guided solely by the MGRM. The ablation study in Section 4.2.3 is also dedicated to this analysis.

---

> ### Author Response · Authors · 2025-08-08
> **W1c: Clarification of the Multi-modal Generative Reward Model (Continued)**
>
> **1c. Effectiveness of MGRM Reward:** We have demonstrated the effectiveness of MGRM from both our conceptual rationale and experiments.
>
> First, reinforcement learning inherently enhances the robustness and reliability of reward signals through iterative policy optimization. We incorporate the GRPO algorithm, which can provide stable reward signals to help MGRM obtain noise robustness in the complicated, multi-turn environments.
>
> From an experimental perspective, we have systematically designed extensive experiments from **3** aspects to verify the efficacy of MGRM: 1) self-evolving with untrained and frozen MGRM (Figure 5, the orange line), 2) self-evolving with MGRM pretrained with GT rewards  (Figure 5, the green line), and 3) self-evolving with self-supervised training MGRM based on TTRL  (Table 1, Response to Reviewer o88n). Our empirical performance gains through these experiments demonstrate the effectiveness of MGRM's key role in the SEEA-R1 framework.
>
> More importantly, we would like to point out that there may be a misunderstanding from your comments: "The paper proposes the MGRM for scenarios where GT rewards are unavailable, yet it fails to test the model in such an environment." **We would like to highlight that we have performed extensive experiments to test MGRM in an environment without access to GT rewards (Figure 5, 4.2.3 Ablation Study and Table 1, Response to Reviewer o88n).** Below, we will respond to your detailed questions:
>
> - **Direct Experimental Validation:** We would like to politely emphasize that we have already tested the model when GT rewards are unavailable and directly validate the effectiveness of MGRM in the experiments  to verify the core motivation and value of MGRM. In Figure 5 of Section 4.2.3, the policy model guided exclusively by rewards from MGRM is trained concurrently, which demonstrates its self-evolution without relying on external GT rewards and confirms the effectiveness of MGRM.
>
>   Furthermore, in our response to Reviewer o88n, we provided **long-term training results (30 iterations) in this GT-free setting,** showing the success rate **consistently rising to 42.54%**. This directly addresses the concern that performance would degrade due to compounding errors and confirms the MGRM's effectiveness.
>
> - **OOD Generalization:** In this work, we propose a generalizable SEEA-R1 framework, rather than specific weights or models. Our methods have demonstrated effectiveness in ALFWorld and show strong transferability to other simulation benchmarks such as **EmbodiedEval**. This demonstrates that the features learned via our framework, guided by the MGRM, **do generalize to OOD environments and tasks.**
>
> - **Improving the reliability of the generated rewards through training:** Reinforcement learning has demonstrated robustness to noisy or imperfect reward signals through iterative policy optimization [2][3]. Similarly, prior studies have shown that reward models can effectively guide policy improvement even when label noise is present [4]. Building upon this foundation, our self-evolving framework **continues to improve model performance** even with the challenging conditions the reviewer raised, such as cold starts and inaccurate reward signals. This is shown in Table 1 in our response to Reviewer o88n, where we use majority votes on final outcomes as an inaccurate pseudo-GT signal to train via TTRL[1], demonstrating that **our MGRM can be applied to environments without access to GT rewards.**
>
> Since your comments in W1c are all about effectiveness instead of novelty, we will respectfully explain the novelty of our methods in W4.
>
> [1] Yuxin Zuo, Kaiyan Zhang, Shang Qu, Li Sheng, Xuekai Zhu, Biqing Qi, Youbang Sun, Ganqu Cui, Ning Ding, and Bowen Zhou. Ttrl: Test-time reinforcement learning. arXiv preprint arXiv:2504.16084, 2025.
>
> [2] Sutton, R. S., & Barto, A. G. (2018). Reinforcement Learning: An Introduction.
>
> [3] Schulman, J., Wolski, F., Dhariwal, P., Radford, A., & Klimov, O. (2017). Proximal Policy Optimization Algorithms.
>
> [4] Gao, T., et al. (2023). Scaling Laws for Reward Model Overoptimization.

---

> ### Author Response · Authors · 2025-08-08
> **W2.  Regarding the Sparse Reward Problem**
>
> We also recognize that sparse rewards and cold-start are well-recognized challenges in reinforcement learning, particularly in long-horizon embodied tasks. **Given their relevance to the task of embodied planning, we specifically design an approach that integrates MCTS with GRPO to address them.**
>
> **1. Our method is effectively addressing the sparse reward problem via MCTS-based sampling.**
>
> Our MCTS-based sampling converts the sparse terminal reward into dense action value through MCTS's backup, which can assist in decision-making to increase the probability of the agent receiving positive environmental rewards.
> 1. In the selection phase, as described in Appendix C (line 799) of our paper, our MCTS-based sampling balances exploration and exploitation using the UCT rule:
> $$
> a_t = \arg\max_{a_{t,i}} \left[ Q(s_t, a_{t,i}) + c \sqrt{\frac{\ln N(s_{t-1}, a_{t-1})}{1 + N(s_t, a_{t,i})}} \right]
> $$
> Compared to relying solely on policy sampling, this trade-off can increase the likelihood of visiting promising actions, thereby improving the chance of finding successful trajectories, even when environmental rewards are sparse.
> 2. In the backup phase, as detailed in Appendix C (line 812) of our paper, our MCTS-based sampling propagates the terminal sparse reward backward through the search tree, converting it into a dense learning signal by updating the Q-values along all visited state–action pairs:
>
> $$
> N(s_t, a_t) \leftarrow N(s_t, a_t) + 1
> $$
>
> $$
> R^{(j)}(s_t, a_t) = r(s^{(j)}\_{t+1}) + \gamma r(s^{(j)}\_{t+2}) + \cdots + \gamma^{T-t-1} r(s^{(j)}_T)
> $$
>
> $$
> Q(s_t, a_t) = E_{a_t \sim \pi_{\theta_{\text{old}}}(\mathcal{A} \mid s_t)} \left[ R(s_t, a_t) \right] = \frac{\sum_{j=1}^{N(s_t, a_t)} R^{(j)}(s_t, a_t)}{N(s_t, a_t)}
> $$
>
>
> These updated Q-values are then used in the next selection phase, effectively integrating the sparse terminal rewards into intermediate decision-making and policy optimization.
>
> Empirically, as shown in Table 1 and Figure 4(a), SEEA-R1 achieves significant performance improvement, increasing the success rate from 11.57% to 36% after training. This directly validates the effectiveness of our MCTS-based sampling even under sparse reward conditions.
>
> **2. Unlike SEEA-R1, most classic methods for sparse rewards, such as HER[1], are not suitable for embodied reasoning.**
>
> The key reason is that our thought–action pairs are conditioned on complex language instructions and include reasoning steps that explicitly decompose and plan for a specific goal, making it difficult to simply change the goal without breaking semantic consistency.
>
> For instance, consider a sampled trajectory that completes the task *"find cd"* under the instruction *"find two cd and put them in safe".* The reasoning might be:
>
> ```
> "Thought: To solve the task, I need to find and take the first CD, then put it in the safe, then find and take the second CD, then put it in the safe. First, I need to find the first CD. A CD is more likely to appear in drawer (1-6), desk (1-2), shelf (1-6), and laundryhamper (1). I can check one by one, starting with drawer 1. Action: go to drawer 1"
> ```
>
> To switch the goal to another task to apply HER, the existing reasoning steps would still explicitly reference the original task *"find two CDs and put them in the safe"*, causing a semantic mismatch between the new goal and the existing plan. Reinforcing such a trajectory would produce inconsistent reasoning–action pairs, ultimately harming learning performance.
> By contrast, our method is inherently more well-suited to this semantic action space, as it corresponds to the original goal–reasoning alignment and avoids introducing these inconsistencies.
>
> **3. Our Q-value estimates are unbiased due to Monte Carlo estimation.**
>
> The Q-values in our framework are not estimated by a learned value function. They are derived via Monte Carlo estimation during MCTS simulation, which has long been recognized as an unbiased estimator of expected reward, as established in foundational works such as Robert (1999) [2] and Kroese [3].
>
> [1] Andrychowicz, M., Wolski, F., Ray, A., Schneider, J., Fong, R., Welinder, P., ... & Zaremba, W. (2017). Hindsight experience replay. Advances in neural information processing systems, 30.
>
> [2] Robert, C. P., Casella, G., & Casella, G. (1999). Monte Carlo statistical methods (Vol. 2). New York: Springer.
>
> [3] Kroese, D. P., Taimre, T., & Botev, Z. I. (2013). Handbook of monte carlo methods. John Wiley & Sons.

---

> ### Author Response · Authors · 2025-08-08
> **W3: Implementation and Practical Details**
>
> Due to space limitations, we omit the decision frequency (in Hz) and latency of the embodied agent and the processing of actions in the ALFWorld simulator. Our action format can be found in our main manuscript and supplementary materials, and we elaborate on these experimental details here.
>
> - **Decision Frequency and Latency:** In our real-world experiments (detailed in our rebuttal to Reviewer me83), the policy's average latency is **0.70 seconds per step.** This corresponds to a decision frequency of approximately **0.81~1.69 Hz.** Our method sufficiently meets real-time needs of embodied high-level planners.
>
> | **Model**   | **GPU Num**      | **Infer backend** | **Quantization** | **Avg prompt throughput (tokens/s)** | **Avg generation throughput (tokens/s)** | **Action frequency (Hz)** | **Latency (s)** |
> |-------------|------------------|-------------------|------------------|--------------------------------------|-----------------------------------------|--------------------------|----------------|
> | SEEA-R1     | A100 80G * 1     | vllm              | BF16             | 505.68                               | 84.28                                   | 1.69                     | 0.59           |
> | SEEA-R1     | A100 80G * 1     | Transformer       | BF16             | 242.28                               | 40.38                                   | 0.81                     | 0.81           |
>
>
> **Action Format and Parsing:**
>
> - In ALFWorld, the model adheres to the ReAct paradigm with a fixed structure: *"Thought:....Action:...",* where task-specific operational strings (e.g., *"go to desk 1", "put apple in fridge"*) are extracted from the segment following *"Action:"* (Line 180).
> - In contrast,  EmbodiedEval uses a choice-oriented format: *"Thought:....Choice: [Option Number]",* where the model selects a predefined option (e.g., *"[1]"*) from provided candidates, with reasoning encapsulated in the *"Thought"* section.
> The difference of action output between ALFWorld and EmbodiedEval has already been shwon in **line 180, E.1 Prompt for the ALFWorld Dataset in the supplementary PDF**  and **line 182, E. 2 Prompt for the EmbodiedEval Dataset in the supplementary PDF**.
>
> As for parsing, the raw string output from the model is processed by matching the extracted *"Action:"* tag's content against a predefined set of valid actions in each environment. This simple string matching ensures that only semantically and syntactically valid actions are executed.
>
> **Conditions of Invalid Actions:**
>
> In the ALFWorld, when the model generates an invalid or unexecutable action, the environment simulator provides clear feedback with *"Nothing happens"*. This feedback serves as a natural negative signal. The agent then incorporates this failure into its subsequent *"Thought:"* step, reasoning about the error to generate a new, valid action in the next iteration. This trial-and-error loop is a core mechanism driving the agent’s learning process in its environment. In the EmbodiedEval, when the model generates an invalid action, the sequence is

---

> ### Author Response · Authors · 2025-08-08
> **W4: Explanation on Novelty of Framework Components**
>
> We thank the reviewer's response and clarify our novelty: **our key novelty is a self-evolution framework for embodied agents addressing the under-explored "autonomous learning without human supervision." This is a previously unaddressed challenge with substantial research value for the community, going beyond a mere combination of existing RL techniques.** Furthermore, each individual component in this framework, including the joint-training, MGRM, and Tree-GRPO, are all innovations specially designed for embodied agents and are different from existing approaches.
>
> **4a. Joint-Training Mechanism**
>
> Importantly, our approach differs from prior “joint training” practices in model-based RL. Unlike settings such as LLaMa where policy and reward models share foundational parameters, our policy and reward models start from completely independent origins. They co-evolve through a closed interaction loop, **actions → environment → observations → rewards**, where adaptation is grounded in environmental feedback. This design realizes *co-iteration of policy and reward without human annotations*, fundamentally distinct from static parameter sharing or parallel optimization. As noted by Reviewer o88n, this explanation of our proposed idea has helped clarify our novelty. Our novel contribution lies in how these models co-evolve through embodied interaction.
>
> **4b. “Multi-modal, multi-turn” nature of MGRM**
>
> We emphasize that MGRM novelly addresses two limitations of previous embodied reward models: **multi-modality and partial observability.**
>
> Multi-modal design addresses a critical gap in embodied intelligence: unlike methods limited to single modalities (e.g., text-only approaches in ALFWorld), our MGRM directly processes diverse environmental observations. This avoids information loss and hallucinations inherent in modality conversion (e.g., image-to-text). Multi-turn reasoning provides global context essential for Partially Observable Markov Decision Processes (POMDPs) [1, 2], enabling more robust reward estimation compared to single-turn evaluations. Together, these features allow MGRM to dynamically assess rewards from environmental feedback—a capability we believe is novel in its integration into an autonomous, self-evolving learning paradigm.
>
> Within our SEEA-R1 framework, MGRM serves as an evolvable reward estimator operating without ground-truth rewards, anchoring reward signals in dynamic environmental feedback. Our work finds that MGRM's non-trivial design delivers tangible performance gains in real-world embodied scenarios
>
> **4c. Tree-GRPO is different from existing RL methods**
>
> Tree-GRPO is a purposeful design for multi-modal, long-horizon embodied reasoning.
> MCTS’s tree structure naturally supports long chain-of-thought reasoning, especially for associating cross-modal observations over multiple steps. GRPO's [4] inherent advantage in group-wise optimization aligns naturally with MCTS’s tree structure, where sibling nodes form implicit groups. This compatibility enables GRPO to efficiently optimize over MCTS-generated trajectories, addressing long-term dependencies critical for embodied reasoning—unlike DPO[3], which primarily enhances preference modeling without improving long-horizon inference.
> Moreover, Tree-GRPO consistently improves success rates with iteration and plans steps more efficiently in long-horizon tasks (Figure 4), outperforming MCTS+DPO and MCTS+SFT baselines.
>
> Both our empirical results and conceptual framework confirm Tree-GRPO’s novelty lies not in a generic outcome of combining RL with MCTS, but instead a result of task-aligned integration that directly addresses the challenges of long-horizon reasoning in embodied agents.
>
> **Last but not least, we would like to express our sincere appreciation to your valuable time and comments. We proposed a novel self-evolution framework for embodied agents addressing the under-explored "autonomous learning without human supervision." Our work contributes substantial research value to the community, going beyond a mere combination of existing RL techniques. If there is some missing of details due to space limitation, we will revise it thoroughly based on your suggestions. We sincerely hope you could consider the final positive ratings and comments from all the other reviewers, who have acknowledged our rebuttal have addressed all their concerns. And we are fully appreciated and committed to improving our paper based on your valuable suggestions. Thank you very much!**
>
> [1] Kaelbling L P, et al. Planning and acting in partially observable stochastic domains. Artificial Intelligence, 1998, 101(1–2): 99-134.
>
> [2] Igl M, et al. Deep variational reinforcement learning for POMDPs. ICML, 2018: 2117-2126.
>
> [3] Huaijie Wang, et al. Offline reinforcement learning for LLM multi-step reasoning. arXiv preprint arXiv:2412.16145, 2024.
>
> [4] Zhihong Shao, et al. Deepseekmath: Pushing the limits of mathematical reasoning in open language models, 2024.

---

### Official Review · Reviewer_WS1c · 2025-07-07

**Clarity:** 3
**Significance:** 3
**Originality:** 3
**Rating:** 4
**Confidence:** 1

**Summary:**

The authors propose SEEA-R1, an RL algorithm that combines policy rollouts with MCTS through a simulated environment to improve model performance. They test it on simulated environments and report SOTA results.

I should say that I'm not qualified to give an informed review on this paper. I don't do practical ML: I have little-to-no research experience with RL, robotics / embodied agents, realistic tasks, or fine-tuning large models. Please give featherweight to my review.

**Questions:**

How would you compare your notion of self-evolution in RL to biological evolution? Why does this merit the term "evolution"?

It naively seems to me that RL agents already generate their own training data by exploring their world. Is this qualitatively different, and if so, how?

**Ethical Concerns:**

["NO or VERY MINOR ethics concerns only"]

**Final Justification:**

This seems to me to be a good contribution, and the authors seem competent. The paper may well be quite impressive. I remain unfamiliar with the domain, though, and I simply do not speak the language of most of the authors' rebuttal. While I feel a bit torn about it, I've decided the right thing to do is to explain this and to keep my score.

**Quality:**

3

**Strengths And Weaknesses:**

The strengths of this paper include:
   - it reports SOTA results on a multimodal RL task.
   - they run some ablations.
   - the explanatory figure on the first page is nice.

The weaknesses, to my understanding, include:
   - they need a simulated env to run MCTS. not clear how this would extend to the physical world, which is less amenable to rewinds.
   - the MCTS seems pretty expensive.
   - doesn't seem particularly conceptually innovative: seems like they're mostly bolting together existing techniques.

---

> ### Author Rebuttal · Authors · 2025-07-31
>
> We thank the reviewer for their detailed comments and valuable insights. We will address each point below, corresponding to weaknesses (W), comments (C), and questions (Q) as raised.
>
> **W1:** They need a simulated env to run MCTS. Not clear how this would extend to the physical world, which is less amenable to rewinds.
>
> **Response to W1:**
> We appreciate the reviewer’s concern about extending MCTS to physical environments. To address this, we designed our framework to support real-world deployments and evaluated its effectiveness through real-world experiments, following a setup similar to RoboVQA[1] for embodied planning evaluation.
> Specifically, we collected image observations and action lists for 6 long-horizon tasks (e.g., Pick & Place, Heat & Place) across 3 real-world settings—mirroring the structured action spaces of ALFWorld but in physical environments. For deployment simplicity, we evaluated SEEA-R1 using single-trajectory execution (no rewinds) and compared it to the baseline Qwen2.5-VL-7B.
> Our results demonstrate that SEEA-R1 outperforms the baseline by a 22% higher success rate across all tasks and scenarios (Table 1). This improvement stems from the self-evolution framework, which enables the model to learn robust features during MCTS-based training—features that generalize effectively to real-world dynamics despite the absence of simulation-specific advantages like rewinds.
> We find that MCTS during training (in simulation) imparts the model with a stronger ability to anticipate long-term consequences, where the model leverages learned patterns to make more reliable decisions in one-pass real-world execution. These findings confirm that our approach transfers well to physical robot settings.
> | Model                     | Pick & Place | Examine in Light | Clean & Place | Heat & Place | Cool & Place | Pick Two & Place | Avg  |
> |---------------------------|--------------|------------------|---------------|--------------|--------------|------------------|------|
> | Qwen2.5-VL-7B-Instruct    | 1/3          | 1/3              | 1/3           | 0/3          | 0/3          | 0/3              | 3/18  |
> | SEEA-R1 (Ours)            | 3/3          | 1/3              | 0/3           | 0/3          | 1/3          | 2/3              | 7/18  |
>
> *Table 1: Performance Comparison on Real-World Scenarios*
>
>
> **W2:** The MCTS seems pretty expensive.
>
> **Response to W2:**
> We thank the reviewer for questions on computational cost. We clarify that our method is highly efficient, achieving superior results with lower computational cost, specifically in terms of token usage at inference.
>
> 1. **Algorithmic Optimization:**
> Standard MCTS's complexity is $O(T⋅D⋅K)$. Our key contribution is algorithmic pruning optimization to solve this. Our pruning strategy uses probabilistic expansion and a path budget to reduce the complexity to a much more scalable $O(T⋅max(D,K))$.
> Specifically, we employ a two-level pruning method:
>    1. Probabilistic Expansion: only having a 50% chance of expanding all k actions at a node.
>    2. Strict Path Budget: More critically, we enforce a hard limit, L, on the number of full k-expansions along any single search path (e.g., L=5).
>
> 2. **Fast Inference: MCTS is for Training Only.**
> To clarify concerns on inference cost: MCTS is a training-time strategy only. At test time, our agent uses fast, ReAct-style single-path sampling and incurs no MCTS overhead. The superior policy learned via MCTS is why our agent achieves 3.1x higher success with 38.8% fewer tokens.
>
> | Model           | Success Rate (%) | Avg. Steps | Total Tokens (Test) |
> |----------------|----------------------|------------|----------------------|
> | Qwen2.5-VL-7B-Instruct| 11.57        | 28.1       | 10,701,255        |
> | SEEA-R1               | 36.19       | 23.37      | 6,554,090            |
>
> We will add this token usage data to our main results tables 1 & 2.
>
> 3. **Training Resources**
> Regarding the 8x A100 GPUs, we clarify that this setup was used to accelerate our research, and all our code can be run on a single A100 GPU.
> - Training Time & Composition: The full training process for SEEA-R1, involving 9 iterations of self-evolution, took 45 hours on our 8x A100 setup. This 45-hour training time is a modest budget to achieve performance surpassing proprietary models like GPT-4o.
> We will add a detailed cost analysis table, including training times and hardware configurations for both our method and the baselines, to the Appendix.
>
>
> **W1:** Doesn't seem particularly conceptually innovative: seems like they're mostly bolting together existing techniques.
>
> **Response to W1:**
> We thank the reviewer's attention towards the novelty of our work. We clarify that at the core, this work is a novel self-evolution framework proposed to address fundamental, under-explored problems in the field of embodied intelligence, and the proposed methods are non-trivial to study.
>
> First, the research problem we target is a novel idea: "how to enable embodied agents to achieve autonomous learning (i.e., iteratively improving capabilities through closed-loop interaction with the environment, without relying on human-designed rewards or supervision)".
> Existing RL and planning methods are mostly designed for symbolic domains (e.g., mathematics, code) or single-task scenarios. In contrast, embodied agents face challenges such as the lack of intermediate feedback in long-horizon interactions and the need for reward generalization across diverse environments. These challenges render previous methods ineffective, and we are the first to introduce the goal of "self-evolution" into the embodied domain, defining a new research direction that requires novel solutions.
>
> Traditional methods have critical limitations: RFT relies on handcrafted sparse rewards, hindering credit assignment in long-horizon tasks; MCTS is inefficient in embodied interactions (due to trajectory dependence and unique action spaces); reward models lack generalization due to task specificity.
> Our SEEA-R1 framework addresses these with tailored innovations:
> - Tree-GRPO: Not merely MCTS+GRPO, but designed for multi-turn embodied tasks to enable GRPO to learn long-term dependencies effectively.
> - MGRM: A generative model that autonomously learns reward signals from multi-modal, multi-turn trajectories.
> - Novel joint training: Unlike traditional systems (which train perception models and rewards independently, leading to static VLMs and limited rewards), our paradigm creates closed-loop collaboration. MLLM-generated trajectories train MGRM to judge step rationality based on VLM reasoning; MGRM’s dynamic rewards then guide MLLM strategy optimization. This bidirectional feedback unites MLLMs and MGRM in co-evolution—MLLM improvements boost MGRM accuracy, and vice versa—enabling human-free self-evolution.
>
>
> **Q1:** How would you compare your notion of self-evolution in RL to biological evolution? Why does this merit the term "evolution"?
>
> **Response to Q1:**
> A Self-Evolving Embodied Agent satisfies three core conditions: automated data synthesis, iterative capability improvement, and a closed-loop reasoning system.
>
> Lewontin[2] defines biological evolution by three essentials—variation, selection, and inheritance—which our self-evolving RL scheme mirrors in silico:
> 1. Random policy sampling, MCTS, and environmental noise introduce behavioral variation.
> 2. A reward/advantage signal selects high-return trajectories to guide learning.
> 3. Gradient descent encodes successful behaviors into model parameters, enabling inheritance.
>
> Here, genes and generations are replaced by parameters and training iterations, but the same variation-selection-inheritance cycle drives open-ended improvement—justifying the term "evolution."
>
> Recent work[3] defines self-evolving agents as reward-driven systems that generate their own data and update parameters autonomously—criteria our framework meets. Critically, the agent improves over time: each evolution round boosts success rates (e.g., in ALFWorld) and reduces task steps, a key evolutionary hallmark.
>
>
> **Q2:** It naively seems to me that RL agents already generate their own training data by exploring their world. Is this qualitatively different, and if so, how?
>
> **Response to Q2:**
> While traditional RL agents (e.g., those using PPO or GRPO) do generate training data through environmental exploration, our approach introduces qualitative differences that are needed in embodied, long-horizon tasks—distinct from the scenarios where standard RL methods are typically applied to LLMs.
>
> First, conventional RL for LLMs focuses on single-turn, answer-centric tasks with well-defined ground truths. Here, trajectories are isolated: each inference path is generated independently, with no reuse of intermediate states or partial progress between trajectories. In contrast, embodied scenarios require multi-turn, long-horizon planning (e.g., sequential action execution in a physical or virtual environment), where success depends on maintaining coherent progress across extended interactions.
>
> Second, standard RL methods for LLMs are ill-suited to model the temporal dependencies in multi-turn embodied interactions. Tree-GRPO, by contrast, structures exploration as a hierarchical tree, where each node encodes cumulative progress, enabling more precise credit assignment and coherent multi-turn planning.
>
> [1] Pierre Sermanet, et al. Robovqa: Multimodal long-horizon reasoning for robotics. arXiv:2311.00899, 2023
> [2] Lewontin R C. The units of selection[J]. Annual review of ecology and systematics, 1970: 1-18.
> [3]Gao H, et al. A Survey of Self-Evolving Agents: On Path to Artificial Super Intelligence[J]. arXiv:2507.21046, 2025.

---

> ### Comment · Area_Chair_bVFi · 2025-08-06
> **Reviewer response needed.**
>
> Hello,
>
> The last step in the reviewing process is to process the updates from the authors that are key in clearing up final issues to ensure papers get a fair treatment. Please respond to the reviewers ASAP to address any final threads.
>
> - AC

---

> ### Author Response · Authors · 2025-08-06
> **Further discussion with Reviewer WS1c**
>
> Dear Reviewer WS1c,
>
> We are truly grateful for the time and effort you put into reviewing our work. In direct response to your feedback, we have provided new empirical evidence from real-world experiments to demonstrate our method's applicability without simulation rewinds. We also offered a detailed cost analysis to clarify that MCTS is a training-time-only strategy, ensuring high efficiency during inference. Furthermore, we elaborated on the conceptual novelty of our self-evolution framework.
>
> Therefore, we hope these detailed responses have fully addressed your concerns. We remain available for any further discussion. If our rebuttal has resolved your questions, we would be very grateful for your consideration of a higher rating, as it would be a great encouragement for our work.
>
> Paper 14735 authors

---

> > ### Comment · Reviewer_WS1c · 2025-08-06
> > **response to authors**
> >
> > Thank you for the detailed reply. I remain unfamiliar enough with the topic to give a higher score than a 4, so I will maintain this score.

---

### Note · Authors · 2025-08-13

Dear Reviewers, AC, and SAC,

We thank all reviewers for their constructive feedback and engagement.
During our discussions, we have addressed key concerns, leading two reviewers  (`o88n`, `me83`) to increase their scores, while two others (`AXsR`, `WS1c`) to maintain their positive ratings. The reviewers acknowledged our responses and our work's strengths:

**Real-World Applicability:**
We showcased SEEA-R1’s applicability to physical robots by demonstrating successful real-world transfers without simulation-only rewinds. As `me83` noted, “showing real-world results strengthens significantly the paper”, and `o88n` appreciated our “qualitative analysis and success rates.” Moreover, our methodology explanation was described as “really helpful for understanding your novelty” (`o88n`).

**Novelty and Contribution:**
Reviewers recognized that our self-evolution framework tackles the “previously [not] explored” challenge (`AXsR`) of autonomous embodied learning without human supervision, going beyond a combination of existing RL components. As `o88n` described, our motivation is “intuitive” and our idea “valuable… to share with the broader research community.” `AXsR` further highlighted that this is “the first to adapt RFT for self-evolving embodied agents.”

To summarize our contributions, our work proposes **SEEA-R1**, the **first framework enabling self-evolving embodied agents to autonomously improve without human-designed rewards or supervision**. This contribution is both conceptually novel, defining “self-evolution” in the embodied domain, and empirically significant:
- **Tree-GRPO** for efficient long-horizon, multi-modal reasoning.
- **MGRM**, a reward model capable of dynamic, multi-turn, multi-modal evaluation under partial observability.
- **Joint co-evolution** of policy and reward without human annotations.

SEEA-R1 opens a new research direction for embodied AI: scalable, autonomous, real-world-capable agents. Strong empirical results in both simulated and physical settings, combined with consistent positive reviewer feedback, such as `o88n`’s “valuable idea” and "comphrehensive rebuttal" comment, `me83`’s praise for the “excellent rebuttal,” and score increases after discussion, highlights the novelty, technical soundness, and community value of this work.
We would also like to express our sincere gratitude to all the reviewers / ACs / SACs for their valuable time and insightful comments.

Sincerely,
The authors of submission 14735.

---

### Decision · Program_Chairs · 2025-09-17

**Decision:**

Accept (poster)

**Comment:**

The reviewers are overall in agreement that the proposed method is novel and an improvement over prior work. There are some details between the method and experiments that were cleared up during the rebuttal process so lead to all reviewers in favor of paper acceptance. We adise the authors to put these updates into the paper.